# Cortical parvalbumin neurons are responsible for homeostatic sleep rebound through CaMKII activation

Kazuhiro Kon[1,8], Koji L. Ode[1,2], Tomoyuki Mano [1,2,3,9], Hiroshi Fujishima[2,4], Riina R. Takahashi[1], Daisuke Tone[1,2], Chika Shimizu[2], Shinnosuke Shiono[1], Saori Yada [1,10], Kyoko Matsuzawa[2], Shota Y. Yoshida[1,2], Junko Yoshida Garçon[2,11], Mari Kaneko[5], Yuta Shinohara[2,12], Rikuhiro G. Yamada[2,4], Shoi Shi[1,2,10], Kazunari Miyamichi [6], Kenta Sumiyama [7,13], Hiroshi Kiyonari [5], Etsuo A. Susaki [1,2,14] & Hiroki R. Ueda [1,2,3,4] ✉

The homeostatic regulation of sleep is characterized by rebound sleep after prolonged wakefulness, but the molecular and cellular mechanisms underlying this regulation are still unknown. In this study, we show that $Ca^{2+}$/calmodulin-dependent protein kinase II (CaMKII)-dependent activity control of parvalbumin (PV)-expressing cortical neurons is involved in homeostatic regulation of sleep in male mice. Prolonged wakefulness enhances cortical PV-neuron activity. Chemogenetic suppression or activation of cortical PV neurons inhibits or induces rebound sleep, implying that rebound sleep is dependent on increased activity of cortical PV neurons. Furthermore, we discovered that CaMKII kinase activity boosts the activity of cortical PV neurons, and that kinase activity is important for homeostatic sleep rebound. Here, we propose that CaMKII-dependent PV-neuron activity represents negative feedback inhibition of cortical neural excitability, which serves as the distributive cortical circuits for sleep homeostatic regulation.

The sleep-wake cycle is homeostatically regulated[1]. The wakefulness history is recorded as sleep need, which is dissipated during the subsequent sleep. In general, a mechanism with negative feedback loops maintains homeostasis[2], but the core feedback system underlying sleep homeostasis is unknown.

Slow-wave activity (SWA), a sleep need indicator[1], represents synchronous activity among cortical neurons in which the firing pattern alternates between depolarized/burst (up) and hyperpolarized/silent (down) states[3]. It has been reported that wake-promoting neurons in the midbrain suppress SWA globally via bottom-up projections toward the isocortex, whereas sleep-promoting neurons in the hypothalamus can inhibit the wake-promoting neurons[4,5]. The mechanism for global cortical activity regulation implies the presence of brain-wide control of sleep homeostasis. Indeed, ablation of sleep-promoting neurons in the

hypothalamic preoptic area (POA) impairs rebound sleep after sleep deprivation (SD)[6]. However, the flip-flop switch between wake- and sleep-promoting neurons[4], resulting in an overall positive feedback system, is less suitable for homeostatic regulation.

Sleep homeostasis is locally regulated in addition to the brain-wide regulation noted by the flip-flop model[7]. In humans and rodents, the SWA is locally enhanced in cortical regions that have been more activated during the preceding wakefulness[8,9]. Prolonged wakefulness results in local sleep, in which cortical neurons locally become silent like in sleep, despite the animal displaying an awake electro-encephalogram (EEG) throughout the brain[10]. These findings suggested that a local and minimal unit of the cortical circuit records their homeostatic sleep need and that the negative feedback system for sleep homeostasis can be found in general cortical circuits.

Parvalbumin (PV)-expressing neurons, the most abundant GABAergic interneurons in the isocortex, contribute to feedforward and feedback inhibition in cortical microcircuits[11,12]. $Ca^{2+}$ activity in PV neurons is increased during both rapid eye movement (REM) and non-REM (NREM) sleep in the somatosensory cortex[13], whereas the activity is constantly higher in REM sleep and incremental during NREM sleep in the motor cortex[14]. Chemogenetic activation of PV neurons in the secondary motor (M2) cortex leads to an increase in NREM sleep duration, but a decrease in the SWA[15]. These findings suggested that the cortical PV neurons play a role in physiological sleep-wake regulation.

Moreover, cortical PV neurons mature throughout postnatal development in terms of distribution, electrophysiological properties, and gene expression[16–18], coinciding with developmental changes in sleep architecture and homeostatic sleep responses[19–23]. Developmental dysfunctions in cortical PV neurons are associated with autism spectrum disorder (ASD)[24,25], where abnormal sleep symptoms are observed in both patients and animal models[26–28], highlighting the potential connection between the maturation of cortical PV neurons and developmental changes of sleep. However, the role of cortical PV neurons in the regulation of sleep architecture and homeostatic sleep rebound remains poorly understood.

Here, we show that activity control of cortical PV neurons is involved in the regulation of sleep homeostasis. We discovered a potential link between reduced sleep stability and a decline in the density of PV neurons during development using long-term sleep recordings and whole-brain analyses of PV-neuron distribution in developing mice. The chemogenetic manipulation of cortical PV-neuron activity revealed that the activity is important for sleep stability. Surprisingly, cortical PV-neuron activity is also required for homeostatic sleep rebound. To induce rebound sleep, cortical PV neurons require $Ca^{2+}$/calmodulin-dependent protein kinase II (CaMKII), whose kinase activity is critical for sleep promotion[29,30]. Bidirectional stimulation of endogenous CaMKII kinase activity in cortical PV neurons demonstrated that CaMKII activity induces homeostatic sleep rebound by stimulating the activity of PV neurons. In summary, these findings support the hypothesis that CaMKII-dependent activity control of cortical PV neurons contributes to the regulation of sleep homeostasis. This mechanism could be a key negative feedback system for sleep homeostasis in general and distributive cortical circuits, explaining local sleep homeostasis regulation.

## Results

### Sleep architecture and sleep homeostasis change with development

We conducted long-term sleep phenotyping in developing mice to better understand the developmental trajectory of the sleep phenotype. The 2-month sleep phenotyping was performed in individual mice from weaning (P21) to adult (P84) stage using a respiration-based sleep phenotyping system, Snappy Sleep Stager (SSS) (Fig. 1a). The SSS system demonstrates high accuracy ($95.3 \pm 0.4\%$) when sleep/wake staging based on EEG/electromyogram (EMG) recordings is used as a reference[31]. We focused on $P_{WS}$ and $P_{SW}$, the transition probabilities from wakefulness to sleep and sleep to wakefulness, respectively, in addition to daily sleep duration (Fig. 1b). There was no significant change in daily sleep duration during the post-weaning development (Fig. 1c). However, both $P_{WS}$ and $P_{SW}$ increased continuously as the developmental stages progressed (Fig. 1d, e), indicating that sleep and wake states were gradually fragmented from weaning to adulthood. The light phase saw a greater increase in $P_{WS}$, while the dark phase saw a greater increase in $P_{SW}$ (Supplementary Fig. 1a–e). Consistent with this, the duration of sleep episodes in the dark phase became shorter than that in the light phase during the later developmental stages (Supplementary Fig. 1f).

Following that, we performed a 6-hour SD in individual mice aged 4, 8, and 12 weeks to validate sleep homeostasis in developing mice (Fig. 1f). The homeostatic rebound in sleep duration was clearly observed at 8 and 12 weeks old, but less so at 4 weeks old (Fig. 1g–i), indicating that the homeostatic regulatory system in sleep amount is immature at 4 weeks old. $P_{WS}$ tended to increase during the rebound phase at 8 and 12 weeks old, whereas $P_{SW}$ tended to decrease, implying that wake fragmentation (i.e., increased $P_{WS}$) and sleep consolidation (i.e., decreased $P_{SW}$) were induced in rebound sleep (Fig. 1j, k). Following the SD experiments, we administered MK-801, an NMDA receptor antagonist, which increases neural excitation and wake consolidation followed by rebound sleep[29,32] (Fig. 1f). This approach allowed us to measure the actual sleep loss induced by MK-801 intraperitoneal (i.p.) administration (Fig. 1l). The homeostatic rebound in sleep duration was clearer at 8 and 12 weeks old than at 4 weeks old after MK-801 administration (Fig. 1l–n), as observed in the SD. Similar to SD, increased $P_{WS}$ and decreased $P_{SW}$ were observed during the rebound phase at 8 and 12 weeks old (Fig. 1o, p). The homeostatic rebound in sleep duration was less pronounced at 4 weeks old, even when rebound sleep was calculated from individual rebound onset without limiting to the dark phase (Supplementary Fig. 2a–f). Together, long-term sleep phenotyping in developing mice revealed that the daily sleep-wake pattern and its homeostatic regulatory system mature with post-weaning development in mice.

### Whole-brain distribution of PV neurons changes with development

We focused on PV neurons as a candidate neural basis for the changes in order to better understand the mechanisms underlying the developmental changes in sleep. The mouse brains were collected at various stages of development after weaning, followed by CUBIC-based tissue clearing for a whole-brain analysis of PV-neuron distribution[33,34] (Fig. 2a). The averaged PV-positive (PV+) cell density in the whole brain was increased from P21 to P28 and gradually decreased with subsequent developmental stages (Fig. 2b, top panel). Similar trends were observed in many cerebrum brain regions, including the isocortex, olfactory bulb (OLF), hippocampus (HPF), cortical subplate (CTXsp), striatum (STR), and pallidum (PAL) (Fig. 2b, middle panel), while the density of PV+ cells decreased monotonically in all brain-stem regions, including the thalamus (TH), hypothalamus (HY), midbrain (MB), pons (P), and medulla (MY) (Fig. 2b, bottom panel). These changes could be attributed to PV+ cell loss or a decrease in PV expression in individual PV+ cells (Supplementary Fig. 3a, b).

A detailed comparison of PV-neuron distribution in the P28 and P84 brains revealed subregion-specific changes in PV+ cell density (Fig. 2c, d). Except for the visual (VIS) cortex, the PV+ cell density in P84 was lower than in P28 in almost all areas of isocortex (Fig. 2e). PV+ cell density in the somatomotor (MO) and anterior cingulate (ACA) cortex gradually decreased beginning at P28, whereas it increased or stabilized during the same period in the VIS cortex (Fig. 2f). The density of PV+ cells in some cortical regions, such as the somatosensory (SS) cortex, changed little during development (Fig. 2f).

In addition to the isocortex, subcortical and brain-stem regions showed region-specific changes in PV-neuron distribution. The PV+ cell density in the striatal caudoputamen (CP) increased from P21 to P28, decreased from P28 to P42, and then stabilized from P42 to P270 (Supplementary Fig. 3c). The PV+ cell density in the globus pallidus external segment (GPe) or reticular nucleus of the thalamus (RT) decreased gradually from P21 to P42 and remained constant after P42 (Supplementary Fig. 3c). PV+ cell density in the zona incerta (ZI) and inferior colliculus (IC) decreased steadily until P270 (Supplementary Fig. 3c). In summary, the decline of PV+ cell density was observed across the entire brain during development when the daily sleep-wake pattern and its homeostatic regulatory system matured.

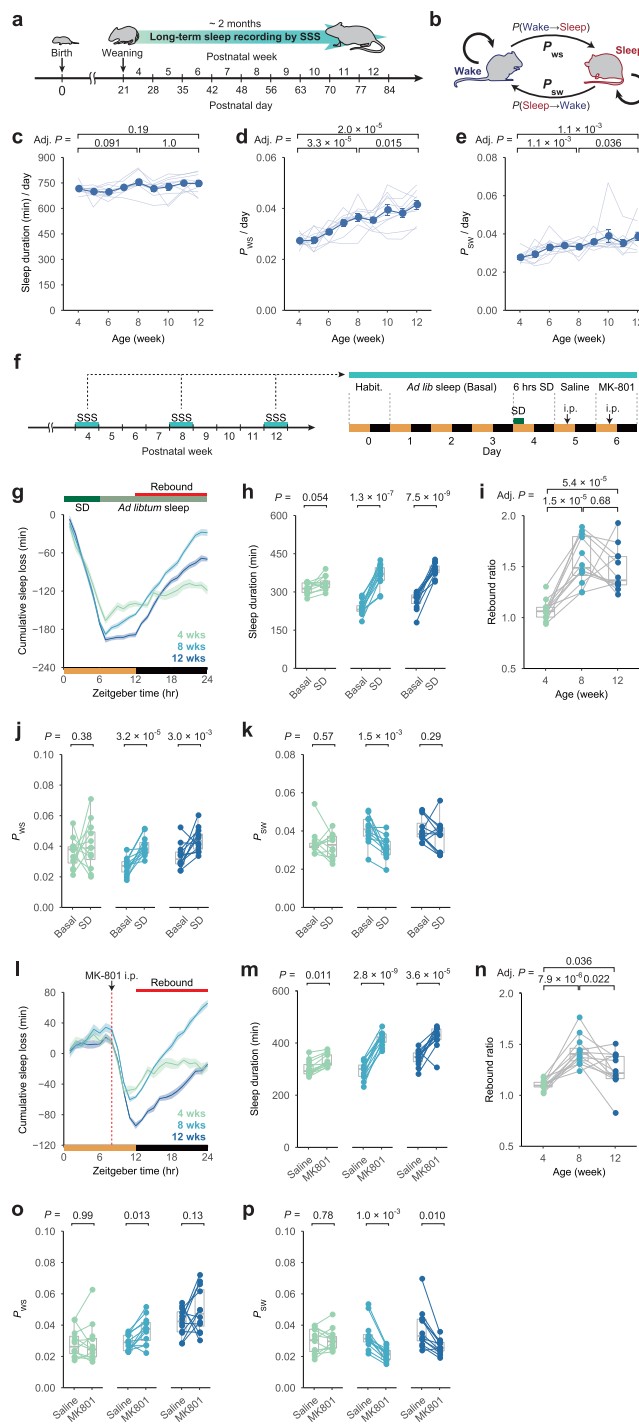

**Fig. 1 | Sleep architecture and sleep homeostasis are altered along with development. a** SSS-based long-term sleep phenotyping during post-weaning development (from P21 to P84). **b** Diagram of transition probabilities between sleep and wakefulness. When $P_{WS}$ or $P_{SW}$ is increased, the wake or sleep state becomes unstable. **c**–**e** Daily sleep duration (**c**), $P_{WS}$ (**d**), and $P_{SW}$ (**e**) in developing mice ($n = 10$). The blue points represent the average sleep parameters at each age. Two-sided Student's paired $t$-test with Bonferroni correction was used on mice aged 4, 8, and 12 weeks. **f** Schematic representation of sleep deprivation (SD) experiments performed on developing mice ($n = 12$) at 4, 8, and 12 weeks of age. Mice were freely behaving during the first 3 days, and the averaged value of sleep parameters is shown as Basal. On the fourth day, SD was performed for 6 h from ZT0 to ZT6. On the fifth (Saline) and sixth (MK-801) days, respectively, saline and 2 mg/kg MK-801 were administered at ZT8. **g** Cumulative sleep loss (SD − Basal) in mice aged 4, 8, and 12 weeks over 24 h. **h, j, k** Sleep duration (**h**), $P_{WS}$ (**j**), and $P_{SW}$ (**k**) on Basal and SD during ZT12−24 ($n = 12$). At each age, two-sided Welch's $t$-test was used. **i** Sleep rebound ratio (SD / Basal during ZT12−24). Individual mouse dots are connected by gray lines. Between the ages, two-sided Welch's $t$-test with Bonferroni correction was used. **l** Cumulative sleep loss (MK-801 − Saline) in mice aged 4, 8, and 12 weeks over 24 h. **m, o, p** Sleep duration (**m**), $P_{WS}$ (**o**), and $P_{SW}$ (**p**) on Saline and MK-801 during ZT12−24 ($n = 12$). At each age, two-sided Welch's $t$-test was used. **n** Sleep rebound ratio (MK-801 / Saline during ZT12−24). Individual mouse dots are connected by gray lines. Between the ages, two-sided Welch's $t$-test with Bonferroni correction was used. Line plots show mean ± SEM. In box plot, boxes show the median, 25th and 75th percentiles, and whiskers show minima to maxima excluding outliers. Source data are provided as a Source Data file.

reports[35] (Fig. 3b and Supplementary Fig. 4c). We also discovered that the SD increased the density of PV+ cell in the isocortex to some extent (Fig. 3c and Supplementary Fig. 4d). Furthermore, there was an increase in c-Fos+ cell density among the PV+ cell population (i.e., c-Fos+PV+ cells), particularly in the isocortex (Fig. 3d and Supplementary Fig. 4e).

Because the SD had the greatest effect on c-Fos+PV+ cell density in the isocortex (Fig. 3d), we looked into changes in c-Fos+PV+ cells in cortical subregions. The number of c-Fos+PV+ cells was higher in SD brains than in S brains in many subregions (Fig. 3e). The SD led to an increase in the population of PV+ cells expressing c-Fos (i.e., c-Fos+PV+ cells in c-Fos+ cells) or c-Fos+ cells expressing PV (i.e., c-Fos+PV+ cells in PV+ cells) (Fig. 3f and Supplementary Fig. 4f, g), indicating that the SD significantly increased the activity of PV+ cells. The same effect was seen with MK-801 administration; cortical c-Fos+PV+ cells were found to be increased in MK-801-treated brains (Fig. 3g, h). These findings indicate that prolonged wakefulness increases cortical PV-neuron activity, implying that these neurons are involved in the homeostatic regulation of sleep.

## Activation of cortical PV neurons induces a rebound-sleep-like state

The adeno-associated virus (AAV)-mediated targeting approach was used to test the influence of PV-neuron activity on sleep architecture and sleep homeostasis (Fig. 3i). PV neurons were targeted using E11, a PV-neuron selective enhancer derived from the *Pvalb* gene region[36]. The E11 enhancer and transgene(s) were packaged in the AAV-PHP.eB vector (Fig. 3i), which can be systemically injected to deliver the vector to the entire brain[37]. As previously reported[36], nuclear-localized mCherry signals under the E11 enhancer (E11-H2B-mCherry) delivered by the AAV-PHP.eB vector were primarily detected in cerebral regions including the isocortex (Fig. 3j, k). Colocalization analysis of E11-H2B-mCherry and anti-PV immunostaining signals revealed that more than 80% of E11-labeled cells in the isocortex were PV+ cells, whereas E11-labeled cells in other regions were less selective for PV+ cells (Fig. 3l and Supplementary Figs. 5a−c). These results demonstrated that the E11 enhancer can, to some extent, selectively target cortical PV neurons. The neural population targeted by this system was termed as E11 neurons after that.

## Cortical PV neurons are activated upon the prolonged wakefulness

Although the PV+ cell density may not directly reflect their inhibitory function, previous studies indicated that a lower number of PV neurons is associated with PV-neuron hypoactivity and reduced inhibition onto excitatory neurons in ASD models[24,25], whose sleep homeostasis is impaired[28]. Thus, we next investigated the potential link between PV-neuron activity and homeostatic regulation of sleep. After a 6-hour SD (ZT6), whole-mouse brains were collected for CUBIC-based clearing and immunostaining of PV and c-Fos proteins[33,34] (Fig. 3a and Supplementary Fig. 4a, b). A comparison of control (*ad libitum* sleep: S) and sleep-deprived (SD) brains revealed that SD increased c-Fos-positive (c-Fos+) cell density in many brain regions, consistent with previous

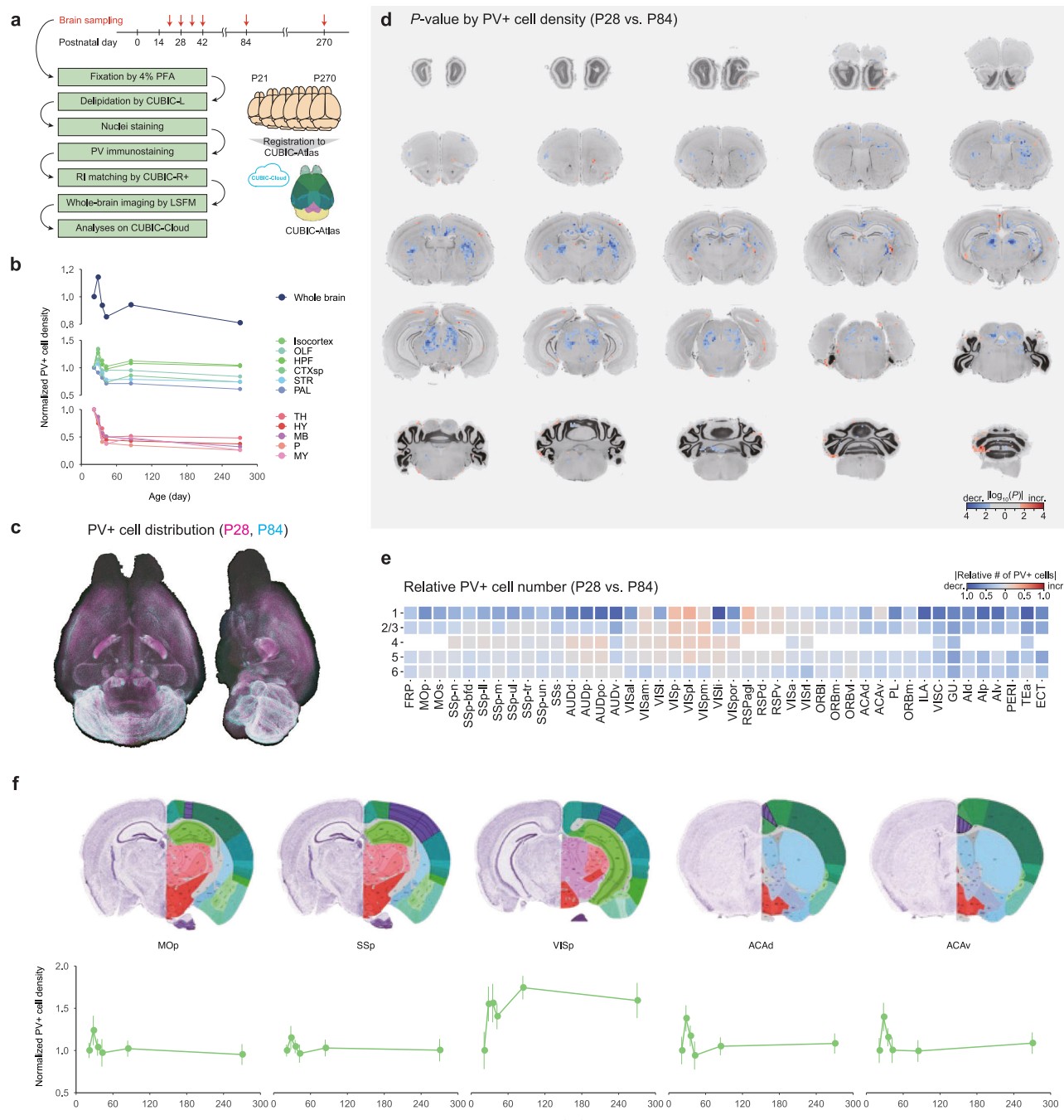

**Fig. 2 | PV-neuron distribution is altered along with development. a** Whole-brain analysis of PV+ cell distribution during post-weaning development. P21, P28, P35, P42, P84, and P270 mouse brains were collected (*n* = 6 for each age). Each set of whole-brain images was uploaded to the CUBIC-Cloud and registered to the CUBIC-Atlas brain. Image of CUBIC-Atlas brain was from the CUBIC-Cloud. **b** The mean of PV+ cell density in the whole brain (top), cerebral regions (middle), and brain-stem regions (bottom) across post-weaning development (*n* = 6 for each age). Each point represents the mean value of each age normalized by P21. **c** Representative whole-brain views of PV+ cell distribution from the dorsal and lateral sides at P28 and P84 ages. **d** A virtual cortical slice depicts a voxel-wise *P*-value heat map of PV+ cell density in comparison to P28 and P84 brains. P28 and P84 brains were tested using two-sided Welch's *t*-test. Red and blue colors represent a significant increase and decrease in P84 brains compared to P28 brains, respectively. Gray was assigned to regions with no significance (*P* > 0.05). **e** A region-wise heat map of relative cell number depicting changes in PV+ cell number in the isocortex. Red and blue colors represent a significant increase and decrease in P84 brains compared to P28 brains, respectively. **f** Mean PV+ cell density across post-weaning development in selected cortical regions (*n* = 6 for each age). Each point represents the mean value of each age normalized by that of P21. Images of coronal brain slice were from the Allen Brain Atlas. Line plots show mean ± SEM. The Allen Brain Atlas ontology is used to define brain region acronyms. Source data are provided as a Source Data file.

To see if E11-neuron activity influenced sleep/wake stability, we expressed a tetanus-toxin light-chain (TeLC) protein under the E11 enhancer to block SNARE-mediated synaptic transmission. TeLC expression under the E11 enhancer resulted in a negligible change in sleep duration but a significant increase in $P_{WS}$ and $P_{SW}$ (Supplementary Fig. 5d–g), demonstrating that chronic silencing of E11 neurons resulted in sleep-wake state fragmentation. In line with this, mice expressing TeLC (E11-TeLC mice) had shorter sleep

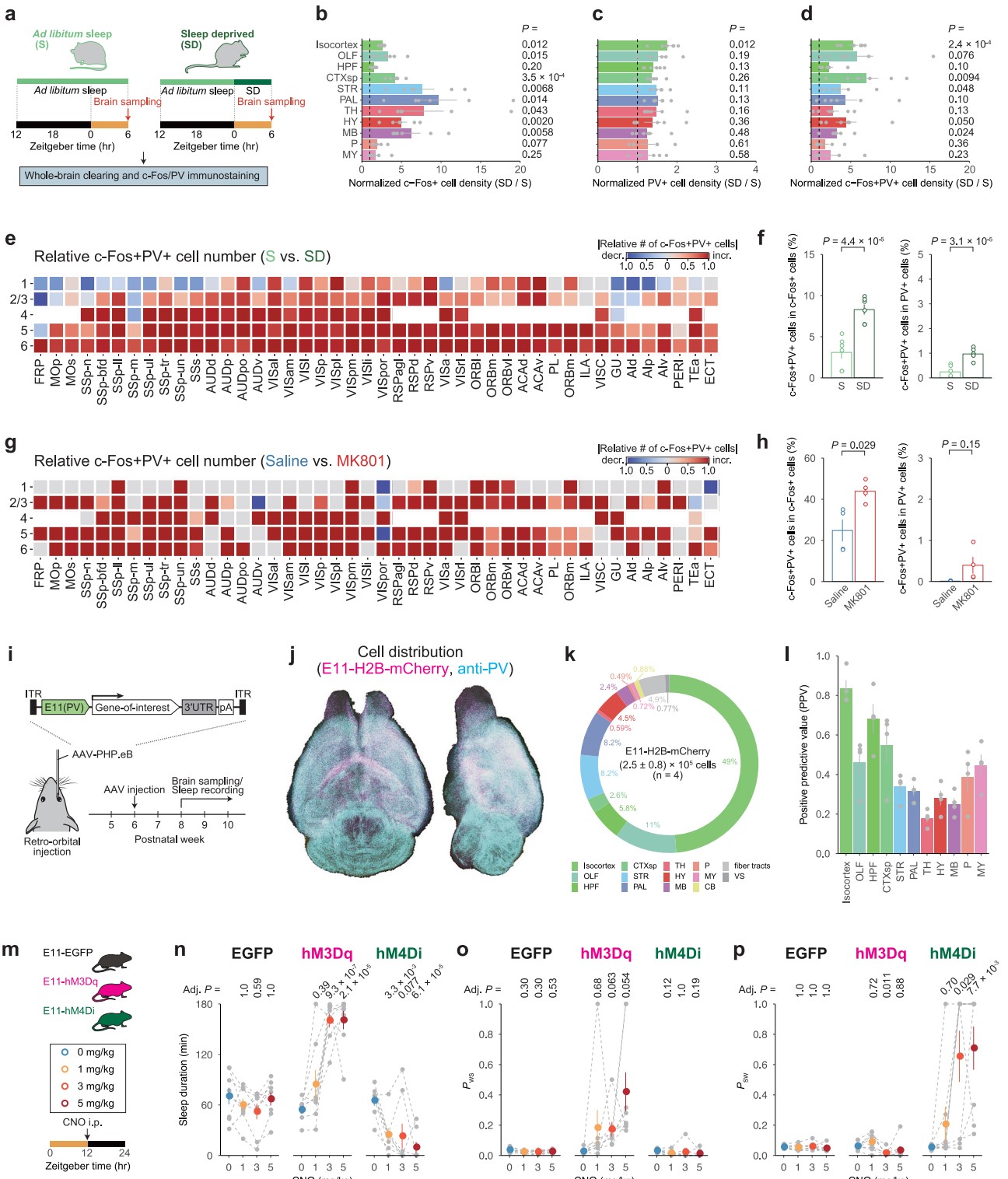

episodes in both the light and dark phases (Supplementary Fig. 5h).

We conducted chemogenetic manipulations of E11-neuron activity to investigate the effects of acute changes in E11-neuron activity on sleep-wake states (Fig. 3m). Mice expressing hM3Dq (E11-hM3Dq mice) had an increase in sleep duration after clozapine-N-oxide (CNO) administration (Fig. 3n and Supplementary Fig. 5i, j). In the E11-hM3Dq-induced sleep state, increased $P_{WS}$ and decreased $P_{SW}$ were observed (Fig. 3o, p and Supplementary Figs. 5k–n). This pattern is similar to the rebound phase after SD (Fig. 1g–p), indicating that E11 neuron excitation

caused the rebound-sleep-like state. Chemogenetic suppression, on the other hand, resulted in an acute reduction in sleep duration in mice expressing hM4Di (E11-hM4Di mice) (Fig. 3n and Supplementary Fig. 5i, j). The decreased sleep duration can be attributed to both wake consolidation (i.e., lower $P_{WS}$) and sleep fragmentation (i.e., higher $P_{SW}$) (Fig. 3o, p and Supplementary Fig. 5k–n), demonstrating that chemogenetic suppression of E11 neurons reduced sleep stability. Similar effect was observed in chemogenetic manipulation using another PV-neuron selective enhancer E22, which predominantly confined gene expression to cortical PV neurons[36] (Supplementary Fig. 6a–f).

**Fig. 3 | Activation of PV-enhancer-targeted (E11) neurons induces a rebound-sleep-like state. a** Whole-brain analysis of c-Fos+PV+ cell distribution in sleep-deprived (SD) condition. Control (S) and SD brains were collected at ZT6. **b–d** Normalized cell density (SD / S) of c-Fos+ (**b**), PV+ (**c**), and c-Fos+PV+ (**d**) cells in major brain regions (*n* = 6 for each group). Individual SD brains were normalized by the mean of S brains (black-dashed lines), and the colored bars represent the mean of SD brains in each brain region. Two-sided Welch's *t*-test was applied to the normalized value of S brains. **e, g** A region-wise heat map of relative c-Fos+PV+ cell number in the isocortex. SD brains were compared to S brains (**e**), and MK-801-injected brains were compared to saline-injected brains (**g**). **f, h** Rate of c-Fos+PV+ cells in c-Fos+ (left) or PV+ cells (right) in the isocortex under SD (*n* = 6 for each group) (**f**) or MK-801 administration (*n* = 4 for each group) (**h**). The groups were compared using two-sided Welch's *t*-test. **i** AAV-based gene expression under E11

enhancer. **j** Representative whole-brain views of dorsal and lateral E11-H2B-mCherry and anti-PV signals. **k** Whole-brain distribution of E11-H2B-mCherry signals (*n* = 4). The plot depicts the demographics of mCherry+ cells in major brain regions. **l** The positive predictive value (PPV) was calculated by dividing the number of mCherry +PV+ cells by the total number of mCherry+ cells (*n* = 4). **m** Each mouse received i.p. injection in turn: 0, 1, 3, or 5 mg/kg CNO injection at ZT12, with at least 2 days between injections. **n–p** Sleep duration (**n**), $P_{WS}$ (**o**), and $P_{SW}$ (**p**) on the days with CNO injection during ZT12–15 (*n* = 8 for each group). Colored dots represent the mean values in each condition. Individual's mouse dots are connected by gray-dashed lines. Within the groups, two-sided Welch's *t*-test with Bonferroni correction were run against the 0 mg/kg CNO condition. Bar and interval plots show mean ± SEM. Brain region acronyms follow the ontology defined by the Allen Brain Atlas. Source data are provided as a Source Data file.

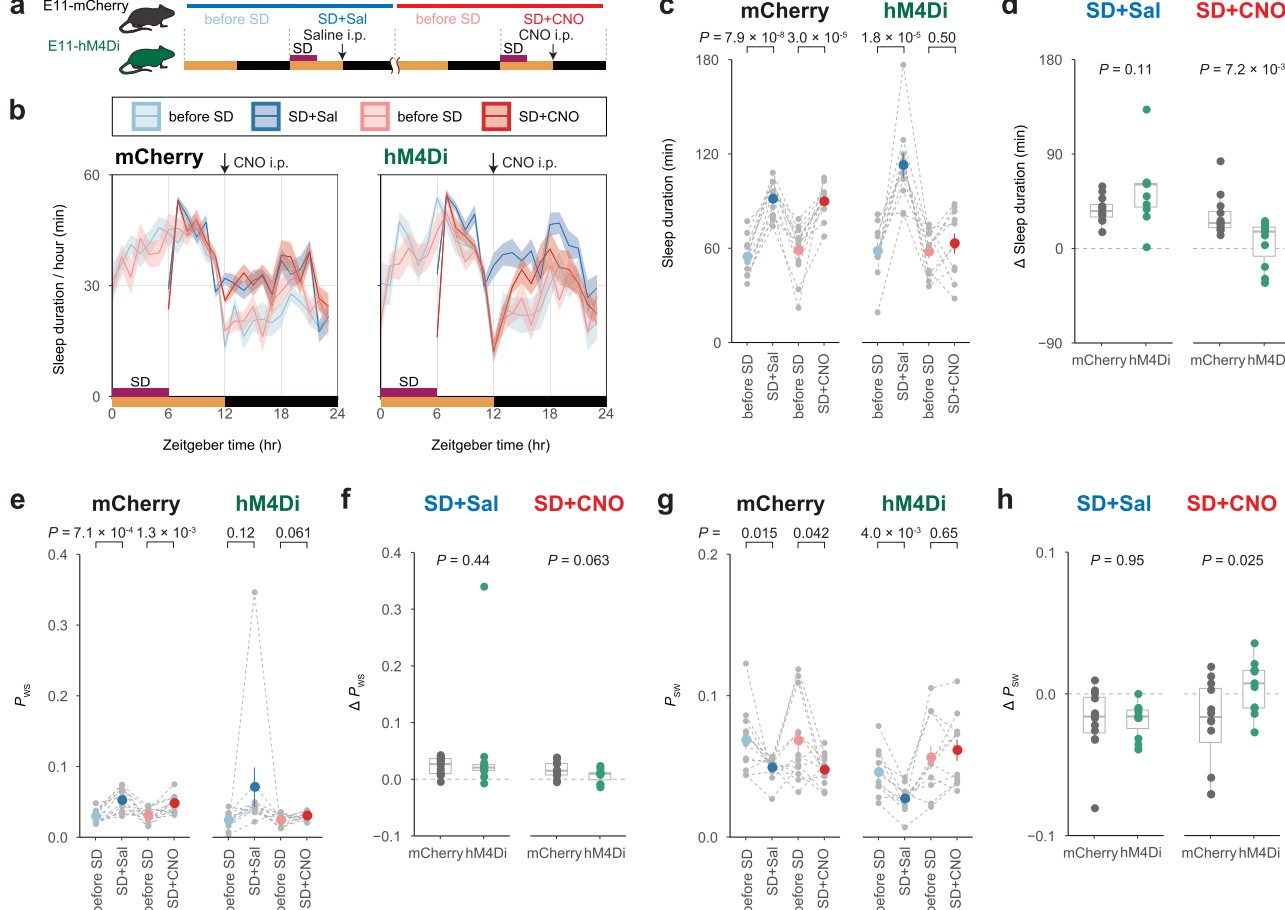

**Fig. 4 | Suppression of E11 neurons inhibits rebound sleep. a** Schematic diagram of CNO injection after sleep deprivation (SD). Every mouse received i.p. injection after a 6-hour SD during ZT0–6. At ZT12, 0 (SD + Sal) or 0.25 (SD + CNO) mg/kg CNO injection was given. One day before the scheduled day with SD and i.p. injections are labeled before SD. **b** Sleep duration over 24 h in E11-mCherry (*n* = 12) and E11-hM4Di (*n* = 11) mice on SD + Sal, SD + CNO, and before SD days. **c, e, g** Sleep duration (**c**), $P_{WS}$ (**e**), and $P_{SW}$ (**g**) on the days shown in the panel during ZT12–15 (**b**). Each gray dot represents an individual mouse of E11-mCherry (*n* = 12) and E11-hM4Di (*n* = 11) groups, while the colored dots represent the mean in each condition.

Individual mouse dots are connected by gray-dashed lines. Two-sided Welch's *t*-test was used to compare the day with SD to the day before SD. **d, f, h** The differences (Δ) in sleep duration (**d**), $P_{WS}$ (**f**), and $P_{SW}$ (**h**) between SD day during and the before SD day during ZT12–15. Two-sided Welch's *t*-test was used to compare the groups in the SD + Sal or SD + CNO conditions. Line and interval plots show mean ± SEM. In box plot, boxes show the median, 25th and 75th percentiles, and whiskers show minima to maxima excluding outliers. Source data are provided as a Source Data file.

## Suppression of cortical PV neurons inhibits rebound sleep

We hypothesized that cortical PV-neuron activity is required for rebound sleep because acute enhancement of the activity induced a rebound-sleep-like state (Fig. 3m–p and Supplementary Fig. 6a–f). To test this, we used chemogenetic inhibition of E11 neurons from the onset of dark period in rebound phase because an increase of sleep duration after SD is mainly shown during dark period (Fig. 4a). We used

a lower CNO dosage that had little effect on the normal sleep-wake cycle (Supplementary Fig. 7a–g). Even in this condition, E11-hM4Di mice had impaired homeostatic rebound in sleep duration after CNO administration (E11-hM4Di mice: SD + CNO), whereas normal rebound sleep was observed in the absence of CNO administration (E11-hM4Di mice: SD + Sal) (Fig. 4b, c). Mice only expressing mCherry (E11-mCherry mice) exhibited normal rebound sleep with or without CNO

administration (E11-mCherry mice: SD + Sal or SD + CNO) (Fig. 4b, c). The comparison of E11-mCherry and E11-hM4Di mice confirmed the impaired rebound sleep in the E11-hM4Di mice with CNO administration (Fig. 4d). Rebound responses in $P_{WS}$ and $P_{SW}$ were also reduced in E11-hM4Di mice after CNO administration (E11-hM4Di mice: SD + CNO) (Fig. 4e–h). These findings suggested that rebound sleep responses (i.e., increased sleep duration, increased $P_{WS}$, and decreased $P_{WS}$) require increased E11-neuron activity.

## CaMKII inhibition in cortical PV neurons impairs sleep homeostasis

We then sought to identify key molecules in cortical PV neurons that are involved in the regulation of sleep architecture and homeostasis. To that end, we concentrated on CaMKII within PV neurons because the phosphorylation states of CaMKII and its potential substrates in forebrain synapses change with sleep-wake states[32,38,39], and animals with CaMKIIα/β genetic modification had abnormal sleep architecture[29,30].

Using open single-cell RNA sequencing (scRNA-seq) data[40], we first examined the mRNA expression of CaMKII isoforms within cortical PV neurons. For this dataset, we considered cells assigned to the *Pvalb* subclass in the scRNA-seq data to be PV neurons. All CaMKII isoforms (*Camk2a*, *Camk2b*, *Camk2g*, and *Camk2d*) were detected at the mRNA level in cortical PV neurons (Supplementary Fig. 8a–d). Furthermore, we used other scRNA-seq data to investigate the developmental changes in the CaMKII expression pattern[41]. *Camk2a* expression level and positive-cell rate in PV neurons were higher in the young adults (P60) compared to juveniles (P21) (Supplementary Fig. 8e–h). CaMKII isoforms are expressed within cortical PV neurons, and the expression pattern of *Camk2a* is developmentally modulated.

To see if CaMKII within PV neurons regulates sleep architecture based on its kinase activity, we inhibited the kinase activity of endogenous CaMKII in E11 neurons with CN19o, an optimized CaMKII inhibitory peptide[42] (Fig. 5a). Mice expressing mCherry-fused CN19o under the E11 enhancer (E11-CN19o mice) showed no significant difference in sleep duration when compared to mice expressing mCherry-fused CN19scr (E11-CN19scr mice), the CN19o scrambled sequence (Supplementary Fig. 9a). However, the E11-CN19o mice had higher $P_{WS}$ and $P_{SW}$ than the E11-CN19scr mice (Supplementary Fig. 9b, c). These findings show that CaMKII inhibition caused sleep-wake state fragmentation in a case similar to E11-neuron chronic silencing (Supplementary Fig. 5c–e).

Following that, we performed a 6-h SD on E11-CN19o and E11-CN19scr mice to investigate the effects of CaMKII kinase activity in E11 neurons on the homeostatic regulation of sleep amount (Fig. 5a). Surprisingly, the E11-CN19o mice barely showed the homeostatic rebound in sleep duration, whereas the E11-CN19scr mice clearly showed rebound responses following SD (Fig. 5b–d and Supplementary Fig. 9d). Increased $P_{WS}$ (i.e., fragmented wakefulness) and decreased $P_{SW}$ (i.e., consolidated sleep), both of which are hallmarks of rebound sleep, were barely observed in E11-CN19o mice (Fig. 5e, f). After MK-801 administration, the E11-CN19o mice showed lower rebound responses than the E11-CN19scr mice, which is consistent with the case of SD (Fig. 5g–k and Supplementary Fig. 9e). These findings suggest that CaMKII kinase activity in E11 neurons is important for both basal sleep architecture and homeostatic rebound of sleep duration.

In order to conduct a detailed analysis of sleep architecture, we performed EEG/EMG recording in E11-CN19o mice. There was no significant difference in NREM sleep or REM sleep duration between the E11-CN19o and E11-CN19scr mice (Fig. 5l, m). The E11-CN19o mice, on the other hand, had higher transition probabilities between wake and NREM sleep states (i.e., increased W → N and N → W) and lower state stabilities (i.e., decreased W → W and N → N) (Fig. 5n). The level of delta (0.5–4.0 Hz) or slow oscillation (SO, 0.5–1.0 Hz) power (i.e., the SWA) during NREM sleep is correlated with the level of sleep depth[43]. The

delta power of the E11-CN19o mice decreased significantly during NREM sleep (Fig. 5o, p and Supplementary Fig. 9f–h). These findings show that decreased CaMKII activity in the E11 neurons reduces NREM sleep quality (i.e., its stability and delta power).

We next investigated whether CaMKII in cortical PV neurons is involved in the homeostatic regulation of the SWA as well as sleep duration. We performed a 6-hour SD on E11-CN19o and E11-CN19scr mice under EEG/EMG recording (Fig. 6a). The delta power during NREM sleep was significantly increased after SD and returned to baseline over 6 h in the E11-CN19scr mice (Fig. 6b, c). In contrast, the E11-CN19o mice presented no remarkable increase in delta power after SD, suggesting that the homeostatic rebound in delta power after SD was attenuated in the E11-CN19o mice (Fig. 6b, c). Accordingly, the delta power rebound was significantly lower in the E11-CN19o mice compared to the E11-CN19scr mice (Fig. 6d). These results indicate that CaMKII kinase activity in E11 neurons is crucial not only for homeostatic rebound of sleep amount but also for homeostatic rebound of the SWA.

To express CN19o in cortical PV neurons more specifically, we employed a confinement strategy using the E11 enhancer and *Pvalb* −2A-Cre mice (Fig. 6e). Cre-dependent expression of E11-CN19o resulted in a reduction in REM sleep duration, while it had no significant impact on total sleep or NREM sleep duration (Supplementary Fig. 9i–k). In contrast, a significant reduction in delta power during NREM sleep was observed (Supplementary Fig. 9l–n). The Cre-dependent E11-CN19o expression markedly attenuated the homeostatic rebound in delta power following SD (Fig. 6f–h), while its effect on the homeostatic rebound in sleep duration was less pronounced and exhibited high individual variation (Supplementary Fig. 9o–p).

## CaMKII activation in cortical PV neurons induces rebound sleep

We then investigated the links between CaMKII activation in E11 neurons and sleep architectures and sleep homeostasis. Because changes in the expression level of *Camk2a* during development are more visible than those of the other CaMKII isoforms (Supplementary Fig. 8g, h), we concentrated on the CaMKIIα isoform in E11 neurons. CaMKIIα is activated by $Ca^{2+}$/calmodulin (CaM) binding and autophosphorylates threonine (T) 286 to maintain kinase activity even when $Ca^{2+}$/CaM is not present[44] (Fig. 7a). As a result, the T286 phosphomimetic mutant (T286D), in which the threonine (T) is replaced with aspartic acid (D), becomes constitutively kinase-active form and phosphorylates physiological substrates including the T286 residue in endogenous CaMKIIα[44]. To see if CaMKIIα activation in E11 neurons affects sleep architecture, we expressed either wild-type (WT) or active-formed (T286D) CaMKIIα under the E11 enhancer. Mice expressing WT CaMKIIα (E11-CaMKIIα (WT) mice) showed no differences in all sleep parameters (sleep duration, $P_{WS}$, and $P_{SW}$) when compared to control (E11-EGFP) mice (Fig. 7b–d). In contrast, E11-CaMKIIα (T286D) mice had a significantly longer sleep duration than E11-EGFP and E11-CaMKIIα (WT) mice (Fig. 7b). The E11-CaMKIIα (T286D) mice also had higher $P_{WS}$ and lower $P_{SW}$ (Fig. 7c, d). These findings suggest that CaMKIIα activation in E11 neurons induces a sleep-predominant state similar to the rebound sleep (i.e., increased sleep duration, increased $P_{WS}$, and decreased $P_{WS}$). T286 non-phosphomimetic alanine (A) mutation (i.e., T286A) or kinase-dead mutation (K42R) adding to T286D (i.e., K42R:T286D) did not exhibit the changes of sleep parameters as observed in E11-CaMKIIα (T286D) mice (Supplementary Fig. 10a–c), confirming that the sleep-promoting effects require the constitutive kinase activity of expressed CaMKIIα. The expression of constitutively active CaMKIIβ (T287D) under the E11 enhancer also resulted in a rebound-sleep-like state as well as CaMKIIα activation (Supplementary Fig. 10d–f).

The next step was to see if the sleep-promoting effects of CaMKIIα activation in E11 neurons were dependent on the circadian core clock genes. CaMKIIα (WT or T286D) was expressed under the E11 enhancer

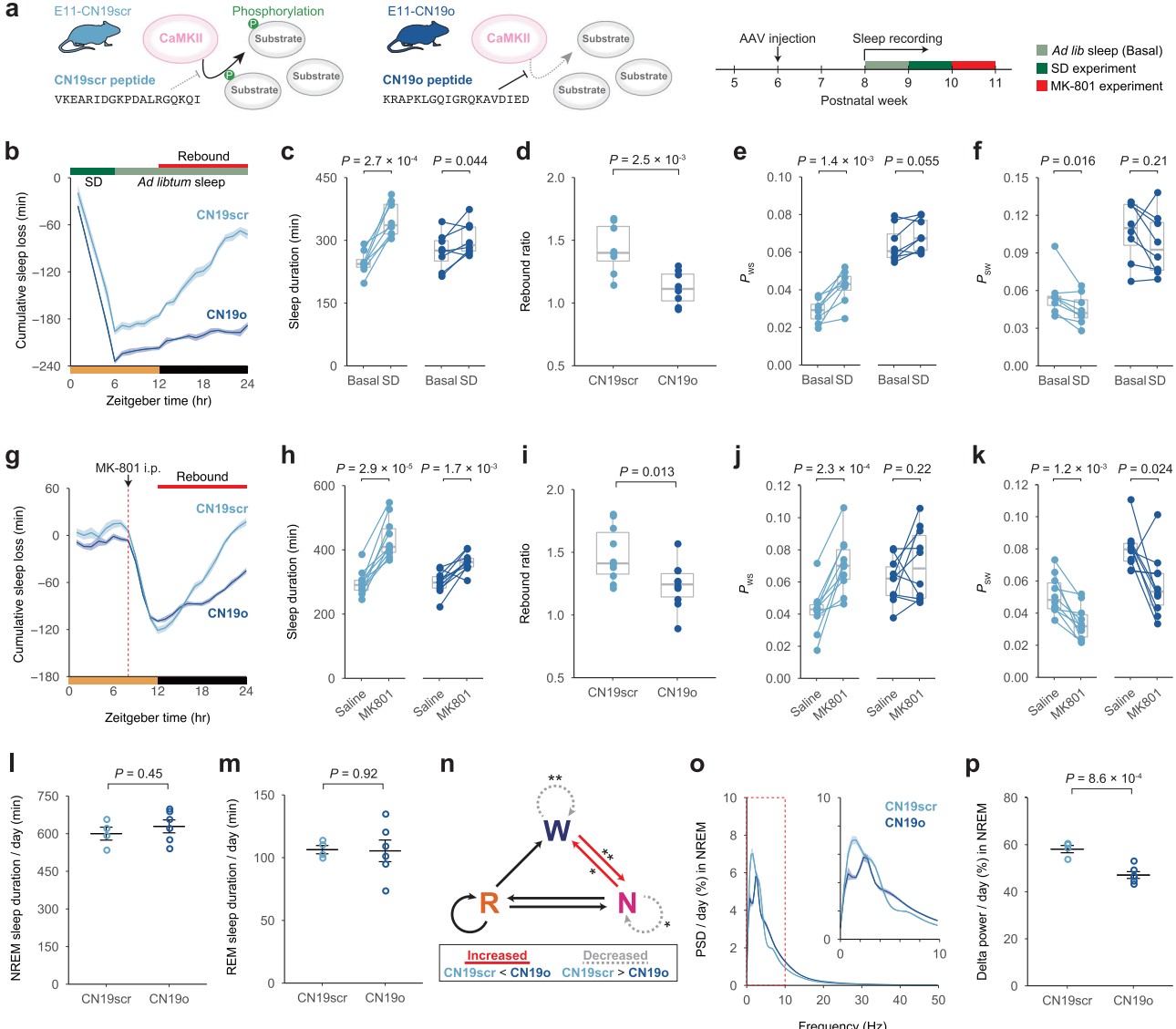

**Fig. 5 | Inhibition of CaMKII activity in E11 neurons inhibits rebound in sleep amount. a** Diagram of CaMKII inhibition by CN19o peptide (left) and experimental schedule (right). **b** Cumulative sleep loss (SD − Basal) in E11-CN19scr and E11-CN19o mice over 24 h (*n* = 8 for each group). The averaged parameters from the three days prior to SD are shown as Basal. **c**, **e**, **f** Sleep duration (**c**), $P_{WS}$ (**e**), and $P_{SW}$ (**f**) during ZT12−24 on Basal and SD (*n* = 8 for each group). Two-sided Student's paired *t*-test was performed within the groups. **d** Sleep rebound ratio (SD/Basal during ZT12-24; *n* = 8 for each group). Two-sided Welch's *t*-test was used. **g** Cumulative sleep loss (MK-801 − Saline) in E11-CN19scr and E11-CN19o mice over 24 h (*n* = 10 for each group). Each mouse received saline or 2 mg/kg MK-801 injection at ZT8, allowing at least 2 days between injections. **h**, **j**, **k** Sleep duration (**h**), $P_{WS}$ (**j**), and $P_{SW}$ (**k**) during ZT12−24 on days with saline or MK-801 injection (*n* = 10 for each group). Two-sided Student's paired *t*-test was used within the groups. **i** Sleep rebound ratio (MK-801 / Saline during ZT12−24, *n* = 10 for each group). Two-sided Welch's *t*-test was used. **l**, **m** Daily NREM sleep duration (**l**) and REM sleep duration (**m**) in E11-CN19scr (*n* = 4) and E11-CN19o (*n* = 6) mice. Two-sided Welch's *t*-test was used. **n** The differences in transition probabilities between wake, NREM, and REM sleep states. The solid-red and dotted-gray lines represent a significant increase and decrease, respectively. **o** Power spectral density (PSD) of the EEG during NREM sleep. The extracted diagram in the 0−10 Hz range is also shown (top-right). **p** Normalized delta power (0.5−4 Hz) during NREM sleep in E11-CN19scr (*n* = 4) and E11-CN19o (*n* = 6) mice. Two-sided Welch's *t*-test was used. Line and interval plots show mean ± SEM. In box plot, boxes show the median, 25th and 75th percentiles, and whiskers show minima to maxima excluding outliers. **P* < 0.05, ***P* < 0.01. Source data are provided as a Source Data file.

in *Per1*[-/-]:*Per2*[-/-] (*Per* dKO) or *Cry1*[-/-]:*Cry2*[-/-] (*Cry* dKO) mice (Fig. 7e). Under constant dark (DD) conditions, these dKO lines exhibit arrhythmic circadian behavior[45,46]. Regardless of circadian time (CT), E11-CaMKIIα (T286D) mice in both *Per* dKO and *Cry* dKO backgrounds kept under DD condition showed a significant increase in sleep duration compared to E11-CaMKIIα (WT) mice (Fig. 7f and Supplementary Fig. 10g, j). The increased sleep amount in E11-CaMKIIα (T286D) mice is explained by higher $P_{WS}$ and lower $P_{SW}$ (Fig. 7g, h and Supplementary Fig. 10h−i, k−l). These findings imply that CaMKII-mediated sleep facilitation is independent of behavioral circadian rhythmicity.

Given the rebound-sleep-like phenotype observed in E11-CaMKIIα (T286D) mice (Fig. 7b−d), it is reasonable to conclude that these mice have already acquired the saturated sleep need and that further SD does not effectively induce rebound sleep. To put this theory to the test, we performed a 6-hour SD on E11-CaMKIIα (T286D) mice. The E11-CaMKIIα (T286D) mice had rebound sleep characteristics even in basal sleep and only had marginal rebound sleep after the SD (Supplementary Fig. 10m−q). Similarly, after MK-801 administration, E11-CaMKIIα (T286D) mice showed lower rebound responses (Supplementary Fig. 10r−v). These findings support the notion that constitutive

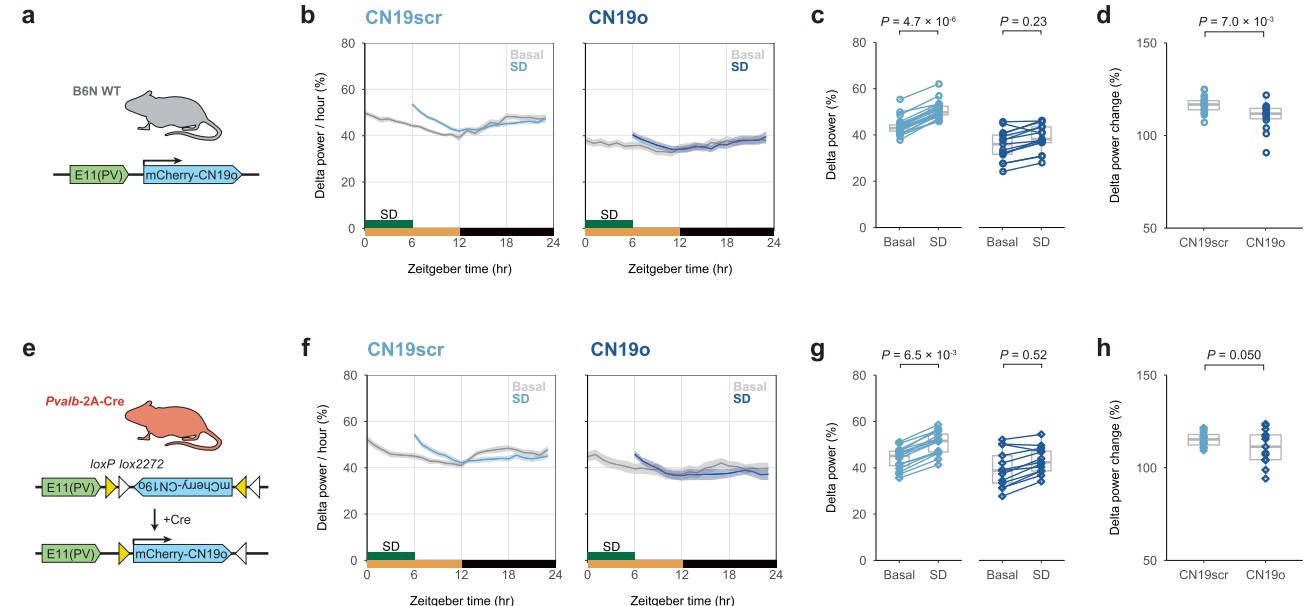

**Fig. 6 | Inhibition of CaMKII activity in E11 neurons inhibits rebound in slow-wave activity. a** Diagram of CN19o expression in E11 neurons for CaMKII inhibition. **b** Hourly normalized delta power (0.5–4 Hz) during NREM sleep in E11-CN19scr ($n = 22$) and E11-CN19o mice ($n = 15$) in SD experiments. Mice were allowed to behave freely for 1 day prior to SD, and the value is shown as Basal. SD was effectively performed during ZT0–6 with a high success rate (93.4 ± 5.3%). **c** Normalized delta power during NREM sleep on Basal and SD during ZT6–12 in E11-CN19scr ($n = 22$) and E11-CN19o mice ($n = 15$). Two-sided Welch's *t*-test was performed between Basal and SD within the groups. **d** The percentage of delta power change upon SD during ZT6–12 in E11-CN19scr ($n = 22$) and E11-CN19o mice ($n = 15$). The groups were compared using two-sided Welch's *t*-test. **e** Diagram of CN19o expression in cortical PV neurons with enhanced specificity using E11 enhancer and *Pvalb*–2A-Cre mice. **f** Hourly normalized delta power (0.5–4 Hz) during NREM sleep in *Pvalb*–2A-Cre:E11-DIO-CN19scr ($n = 16$) and *Pvalb*–2A-Cre:E11-DIO-CN19o mice ($n = 13$) in SD experiments. Mice were allowed to behave freely for 1 day prior to SD, and the value is shown as Basal. SD was effectively performed during ZT0–6 with a high success rate (95.0 ± 6.2%). **g** Normalized delta power during NREM sleep on Basal and SD during ZT6–12 in *Pvalb*–2A-Cre:E11-DIO-CN19scr ($n = 16$) and *Pvalb*–2A-Cre:E11-DIO-CN19o mice ($n = 13$). Two-sided Welch's *t*-test was performed between Basal and SD within the groups. **h** The percentage of delta power change upon SD during ZT6–12 in *Pvalb*–2A-Cre:E11-DIO-CN19scr ($n = 16$) and *Pvalb*–2A-Cre:E11-DIO-CN19o mice ($n = 13$). The groups were compared using two-sided Welch's *t*-test. Line plots show mean ± SEM. In box plot, boxes show the median, 25th and 75th percentiles, and whiskers show minima to maxima excluding outliers. Source data are provided as a Source Data file.

CaMKIIα activation in E11 neurons induces a rebound-sleep-like state, and that the mice already had a saturated level of sleep need.

Furthermore, we recorded EEG and EMG in E11-CaMKIIα (WT) and E11-CaMKIIα (T286D) mice. The increased sleep duration in the E11-CaMKIIα (T286D) mice is due to an increase in NREM sleep (Fig. 7i), whereas REM sleep did not differ between groups (Fig. 7j). Consistent with this, E11-CaMKIIα (T286D) mice had higher NREM sleep stability (i.e., increased N → N) (Fig. 7k). The SO power was increased in E11-CaMKIIα (T286D) mice (Fig. 7l, m and Supplementary Fig. 11a, b), but the delta power changes were not significant (Fig. 7l and Supplementary Fig. 11c). These findings show that constitutive CaMKIIα activation in E11 neurons improves both the quantity and quality of NREM sleep.

To confirm the involvement of cortical PV neurons, we employed the confinement strategy using the E11 enhancer and *Pvalb*–2A-Cre mice for specific CaMKIIα expression in cortical PV neurons (Fig. 7n). Similar to the direct CaMKIIα expression under the E11 enhancer, the rebound-sleep-like state was induced by Cre-dependent CaMKIIα (T286D) expression under the E11 enhancer (Fig. 7o–q). Sleep-promoting effects were also observed with Cre-dependent CaMKIIα (T286D) expression under pan-neural *hSyn* promoter (Supplementary Fig. 11d–g). These findings further support the hypothesis that CaMKII activation in cortical PV neurons induces rebound sleep.

**CaMKII activity in cortical PV neurons reflects sleep need**

CaMKII kinase activity in E11 neurons has similar effects on sleep homeostasis as E11-neuron activity manipulation. These findings support the hypothesis that CaMKII activation increases cortical PV-neuron activity, causing rebound sleep. To test this hypothesis, we used double immunostaining of PV and c-Fos proteins in brains

expressing CaMKIIα (T286D) in E11 neurons to look at the activity of cortical PV neurons. The isocortex contained more than three-quarters of the detected c-Fos+PV+ cells (Fig. 8a, b). When E11-CaMKIIα (T286D) brains were compared to E11-CaMKIIα (WT) brains, c-Fos+PV+ cells were increased in entire cortical regions (Fig. 8c). CaMKIIα (T286D) expression in E11 neurons selectively enhanced c-Fos expression in cortical PV neurons (Fig. 8d, e) but had little effect on the overall number of c-Fos+ or PV+ cells (Supplementary Fig. 12a, b). These findings imply that intracellular CaMKII activation increases PV-neuron activity. Given that prolonged wakefulness also increases PV-neuron activity (Fig. 3d–h), higher sleep needs appear to increase the activity, regardless of whether the preceding state was sleep or wakefulness.

We next evaluated whether enhancement of c-Fos expression upon CaMKII activation is characteristic of PV neurons. We expressed CaMKIIα and H2B-mCherry in excitatory neurons using *Vglut2*-ires-Cre mice to induce intracellular CaMKII activation in *Vglut2* neurons, not in PV neurons (Supplementary Fig. 13a, b). The sleep-promoting effects were observed with CaMKIIα (T286D) expression in *Vglut2* neurons (Supplementary Fig. 13c–e), consistent with sleep induction by CaMKIIβ activation in this population[30]. However, CaMKIIα (T286D) expression in *Vglut2* neurons did not alter c-Fos expression profile of cortical *Vglut2* neurons (Supplementary Fig. 13f, g), suggesting that intracellular CaMKII activation specifically enhances PV-neuron activity, rather than excitatory neuron activity, despite their similar sleep-promoting effects.

The next step was to see if the level of CaMKII T286 autophosphorylation in cortical PV neurons increases in response to the accumulation of sleep need. We developed a system for measuring CaMKIIα T286 autophosphorylation using liquid chromatography with

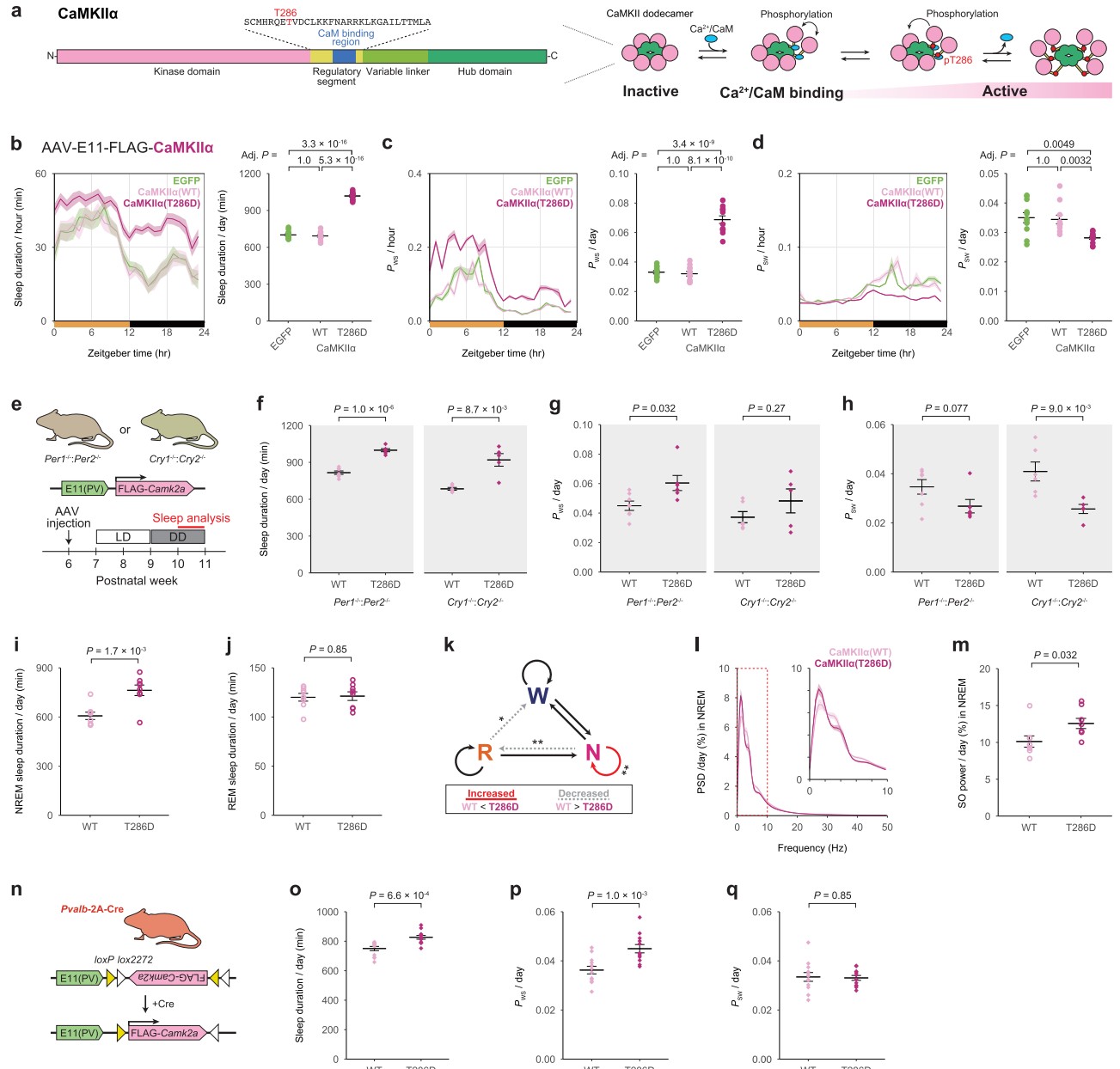

**Fig. 7 | Elevation of CaMKIIα activity in E11 neurons induces rebound-sleep-like state. a** Illustration of CaMKIIα protein sequences (left) and endogenous CaMKIIα activation mechanism (right). The phosphorylation of T286 residue in the regulatory segment is critical for CaMKIIα activity regulation. **b–d** Daily sleep duration (**b**), $P_{WS}$ (**c**), and $P_{SW}$ (**d**) in E11-EGFP ($n = 11$), E11-CaMKIIα (WT) ($n = 11$), and E11-CaMKIIα (T286D) ($n = 12$) mice. The groups were compared using two-sided Welch's $t$-test with Bonferroni correction. **e** Diagram of CaMKII expression and experimental schedule in *Per* or *Cry* dKO mice under constant dark conditions. **f–h** Daily sleep duration (**f**), $P_{WS}$ (**g**), and $P_{SW}$ (**h**) in the *Per* (left) or *Cry* (right) dKO mice expressing E11-CaMKIIα (WT) ($n = 7$ or 6) or E11-CaMKIIα (T286D) ($n = 6$ or 5). The groups were compared using two-sided Welch's $t$-test. **i, j** Daily NREM sleep duration (**i**) and REM sleep duration (**j**) in E11-CaMKIIα (WT) and E11-CaMKIIα (T286D) mice ($n = 8$ for each group). The groups were compared using two-sided Welch's $t$-test. **k** The difference in transition probabilities between wake, NREM, and

REM sleep states in E11-CaMKIIα (T286D) mice versus E11-CaMKIIα (WT) mice. The solid red and dotted gray lines represent a significant increase and decrease, respectively. **l** Power spectral density (PSD) of the EEG during NREM sleep. The extracted diagram in the 0–10 Hz range is also shown in the top-right corner of the panel. **m** Normalized slow-oscillation power (SO, 0.5–1 Hz) during NREM sleep in E11-CaMKIIα (WT) and E11-CaMKIIα (T286D) mice ($n = 8$ for each group). The groups were compared using two-sided Welch's $t$-test. **n** Diagram of CaMKIIα expression in cortical PV neurons with enhanced specificity using E11 enhancer and *Pvalb*–2A-Cre mice. **o–q** Daily sleep duration (**o**), $P_{WS}$ (**p**), and $P_{SW}$ (**q**) in *Pvalb*–2A-Cre:E11-DIO- CaMKIIα (WT), and *Pvalb*–2A-Cre:E11-DIO-CaMKIIα (T286D) mice ($n = 12$ for each group). The groups were compared using two-sided Welch's $t$-test. Line and interval plots show mean ± SEM. *$P < 0.05$, **$P < 0.01$. Source data are provided as a Source Data file.

tandem mass spectrometry (LC-MS/MS) in conjunction with AAV-mediated expression of watermark mutant CaMKIIα in E11 neurons (Fig. 8f). We created a watermark mutation near CaMKIIα T286 autophosphorylation residue by replacing glutamic acid (E) with aspartic acid (D) (i.e., E285D). The watermark mutant CaMKIIα (E285D) is distinguishable from all endogenous CaMKII isoforms

(Supplementary Fig. 14a), allowing us to selectively quantify the T286 autophosphorylation level of CaMKIIα (E285D) expressed in the brain (Fig. 8f). We confirmed that CaMKIIα (E285D) expression and kinase activity were nearly identical to WT CaMKIIα (Supplementary Fig. 14b, c). We also confirmed that no difference in any sleep parameters was observed between E11-CaMKIIα (WT) and E11-CaMKIIα

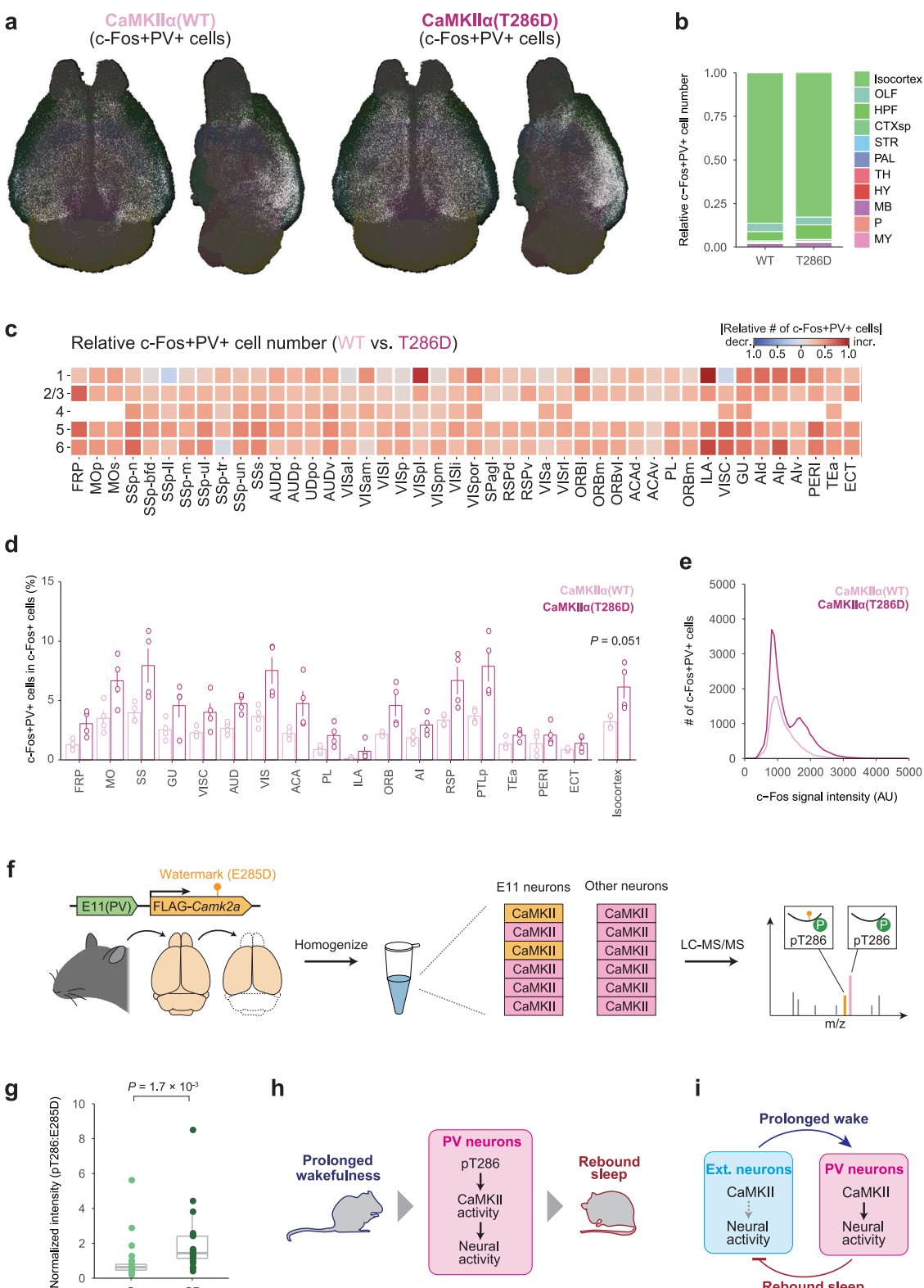

(E285D) mice under basal and sleep-deprived conditions (Supplementary Fig. 14d–n). These findings show that the E285D mutation has no effect on CaMKIIα physiological function in vitro or in vivo. We gave the E11-CaMKIIα (E285D) mice a 6-h SD and collected their cortical regions of brain immediately afterward (Fig. 8f). T286 phosphorylation (pT286) level derived from E11-CaMKIIα (E285D) increased in SD group compared to *ad libitum* sleep (S) group (Fig. 8g). Although pT286 level serves as a surrogate for CaMKIIα kinase activity, this result suggests

that CaMKII kinase activity in cortical PV neurons increases upon the accumulation of sleep need.

## Discussion

We identified cortical PV neurons as a neural population responsible for homeostatic sleep rebound. We also discovered that CaMKII kinase activity in cortical PV neurons is essential for rebound sleep. As a result, we propose a model in which CaMKII kinase activity increases cortical

**Fig. 8 | CaMKII activity in E11 neurons reflects sleep need. a** Representative whole-brain views of dorsal and lateral double-positive (c-Fos+PV+) cell distribution. The brains of E11-CaMKIIα (WT) (left) and E11-CaMKIIα (T286D) (right) mice were collected. **b** Stacked percentage bar plots of c-Fos+PV+ cells in the E11-CaMKIIα (WT) and E11-CaMKIIα (T286D) brains ($n = 4$ for each group). The plot depicts the demographics of c-Fos+PV+ cells in major brain regions. **c** A region-wise heat map of relative cell number showing changes in c-Fos+PV+ cell number in the isocortex. The E11-CaMKIIα (T286D) brains were compared to the E11-CaMKIIα (WT) brains, with red indicating an increase and blue indicating a decrease. **d** The rate of double-positive (c-Fos+PV+) cells in c-Fos+ cells in each cortical region (left) or entire isocortex (right). The groups were compared using two-sided Welch's $t$-test. **e** The cortical distribution of c-Fos signal intensity per c-Fos+PV+ cell in each group. **f** Experimental design for quantitative analysis of CaMKIIα T286

phosphorylation in E11 neurons. The cortical region of E11-CaMKIIα (T285D) mouse brain was dissected, and homogenates were used for LC-MS/MS analysis. **g** Signal intensities of peptides derived from CaMKIIα (T285D) with phosphorylated T286 residues ($n = 21$ for each group). The intensities were normalized to the total CaMKIIα level, which was calculated using all quantifiable non-phosphorylated peptides derived from WT CaMKIIα. Two-sided two-samples Wilcoxon test was used to compare the groups. **h** CaMKII-dependent activity regulation of cortical PV neurons for homeostasis regulation. **i** Cortical feedback circuit model for sleep homeostasis regulation. Bar plots show mean ± SEM. In box plot, boxes show the median, 25th and 75th percentiles, and whiskers show minima to maxima excluding outliers. Brain region acronyms follow the ontology defined by the Allen Brain Atlas. Source data are provided as a Source Data file.

PV-neuron activity, resulting in homeostatic sleep rebound (Fig. 8h). This model is supported by the following: (1) the inhibition/activation of cortical PV neurons suppresses/induces rebound sleep; (2) Prolonged wakefulness activates cortical PV neurons and CaMKII within the neurons; (3) CaMKII activation in cortical PV neurons evokes PV-neuron activity and a rebound-sleep-like state, while inhibition suppresses rebound sleep. These findings imply that cortical negative feedback circuits formed by PV neurons regulate sleep homeostasis.

Although previous research has shown that subcortical and/or brain-stem regions regulate sleep homeostasis across the brain, brain-wide regulatory systems are unable to fully account for several aspects of sleep homeostasis as demonstrated by local sleep, in which cortical activity becomes a sleep pattern locally. Recent research has revealed that silencing cortical excitatory (pyramidal) neurons in layer 5 results in impaired sleep rebound[47], demonstrating cortical regulation of sleep homeostasis. In this study, we discovered that increased activity of cortical PV neurons is required for rebound sleep. Our findings emphasize the importance of both cortical excitatory and inhibitory neurons in the regulation of sleep homeostasis. PV neurons establish negative feedback loops with pyramidal neurons, enabling the detection and synchronization of local circuit activity[48,49]. Given that synchronized cortical activity correlates with an expected level of sleep need[50], cortical PV neurons may regulate the synchronicity of local cortical microcircuits and generate cortical slow waves in collaboration with other inhibitory neurons[51,52]. The microcircuits, characterized by reciprocal connections between pyramidal and PV neurons, are widespread across cortical areas and species[49,53,54]. It would be interesting to see if the microcircuits of pyramidal-PV neurons are responsible for the local sleep observed in the mammalian cortex.

While kinase activity, including CaMKII and salt-inducible kinase 3 (SIK3), in excitatory neurons has been shown to influence baseline sleep amounts and depth[30,55], the specific kinases involved in homeostatic regulation of sleep within inhibitory neurons remain unidentified. Our findings suggest that CaMKII kinase activity in cortical PV neurons is one of the molecular factors reflecting sleepiness. CaMKII inhibition in excitatory neurons resulted in changes in the total amount of daily sleep[30]. In contrast, CaMKII inhibition in cortical PV neurons had minimal impact on daily sleep amount but impaired homeostatic rebound of sleep. Moreover, the intracellular CaMKII activation specifically enhances the activity of cortical PV neurons (Fig. 8c–e), not excitatory neurons (Supplementary Fig. 13f, g), even though both have similar sleep-promoting effects. These findings suggest that distinctive CaMKII signaling pathways exist in cortical PV neurons for the homeostatic regulation of sleep (Fig. 8i). The precise molecular mechanisms underlying this modulation remain unclear, but synaptic strength and/or intrinsic excitability in PV neurons may be altered. Indeed, sleep-wake states influence neural activity of cortical PV neurons and a balance between synaptic excitation and inhibition (E-I balance) in cortical microcircuits[13,14,56]. Although the downstream targets (i.e., substrates) of CaMKII remain unknown, the core circadian clock factors are not involved (Fig. 7e–h and Supplementary

Fig. 10g–l). The voltage-dependent $K^+$ channels $K_v3.1$ and $K_v3.3$ may be important candidates because double knockout of these channels changes the electrophysiological properties of PV neurons and impairs rebound sleep[57]. It will be important to investigate what physiological properties of PV neurons are modulated by CaMKII and what molecular mechanisms are responsible for this modulation through physiological recordings.

While studies in rats and mice contributed to understanding of the developmental aspects of sleep homeostasis, the results across these studies are not fully consistent. Some studies suggest that homeostatic rebound in sleep amount emerges even before weaning in rats[22,58], whereas others indicate that it matures during the post-weaning period in rats or mice[21,23]. Our longitudinal sleep recordings in developing mice found that rebound sleep appears late in developmental stage (Fig. 1g–p). Long-term EEG/EMG recording to assess the SWA remains a future challenge, as it could not be validated in this longitudinal sleep recording. However, it is noteworthy that the SWA is negatively correlated with the number of micro-arousals[43], suggesting that the decreased $P_{SW}$ could serve as a surrogate for the increased SWA. Consistent with the previous report that a significant increase in SWA after SD is observed only in mice older than 6 weeks old[21], a significant decrease in $P_{SW}$ after SD was observed at 8 and 12 weeks old, but not at 4 weeks old (Fig. 1k,p). Our findings align with previous reports suggesting that the homeostatic regulatory system of sleep is immature in juveniles and matures during the post-weaning period in mice.

The scRNA-seq analysis revealed that *Camk2a* expression in cortical PV neurons increases from juveniles to adulthood (Supplementary Fig. 8e–h). Given that CaMKIIα in cortical PV neurons is required for rebound sleep, the lack of rebound response in the juvenile phase may be due to premature CaMKII activation for rebound sleep induction. While it was conventionally believed that cortical PV neurons lack CaMKIIα expression at the protein level, a recent study identified CaMKIIα protein signals in a subpopulation of cortical PV neurons[59]. Another study demonstrated that CaMKIIγ in cortical and hippocampal PV neurons plays a crucial role in synaptic regulation and memory[60]. Investigating the developmental changes in protein expression of CaMKII isoforms within PV neurons and their specific functions in sleep homeostatic regulation remains crucial tasks for future research.

Various cortical and subcortical cell populations, including neurons and glial cells, contribute to homeostatic regulation of sleep[6,47,61,62]. Sleep substances such as adenosine and cytokines are also involved in this regulation[63]. Investigating the interplay of these factors and cortical PV neurons to regulate sleep homeostasis would be intriguing for future research. As a recent study revealed that distinct populations of excitatory neurons respectively regulate sleep duration and SWA through SIK3 signaling[55], it also may be crucial to identify subpopulation(s) of PV neurons involved in homeostatic rebound in sleep duration and SWA. Previous finding showed that activating cortical PV neurons in the M2 cortex increased NREM sleep duration but

decreased the SWA[15], whereas our results indicated that elevating E11-neuron activity through CaMKII activation increases both NREM sleep duration and SWA. These differences might be attributed to the heterogeneous nature of cortical PV neurons and the E11 enhancer's potential to selectively target specific subpopulation(s) within these neurons[36,40]. Furthermore, CaMKII inhibition in E11 neurons significantly suppressed homeostatic rebound in sleep duration and SWA, whereas combining the E11 enhancer and *Pvalb*−2A-Cre mice predominantly suppressed SWA rebound. Although this difference may result from lower efficiency of CaMKII inhibition in the Cre-dependent expression, there is a possibility that specific subpopulations within E11 neurons regulate the two aspects of homeostatic rebound of sleep with different efficiencies.

In conclusion, we propose that CaMKII is a key molecular factor in cortical PV neurons for encoding sleep need and inducing rebound sleep. The proposed mechanism may be responsible for local sleep homeostasis via the canonical negative feedback circuit in cortical inhibitory-excitatory neuron network (Fig. 8i). Such a regulatory motif for sleep homeostasis may be conserved across the animal kingdom: a distinct negative feedback circuit in flies is critical for sleep homeostasis regulation, with physiological properties modulated by sleep need[64,65]. The design principle of the sleep homeostatic machinery may be shared across species.

## Methods

### Mice
The Animal Care and Use Committee of the Graduate School of Medicine in the University of Tokyo or the Institutional Animal Care and Use Committee of RIKEN BDR approved all animal care and experimental procedures. Mice were housed in humidity- and temperature-controlled conditions on a 12-hour light/dark (LD) cycle, with standard chow and water available *ad libitum* in groups, unless otherwise specified. This study used C57BL/6NJcl (B6N WT) mice (CLEA Japan), *Pvalb*−2A-Cre mice (heterozygous B6.Cg-*Pvalb*$^{tm1.1(cre)Aibs}$/J mise; #012358, The Jackson Laboratory), *Vglut2*-ires-Cre mice (heterozygous *Slc17a6*$^{tm2(cre)Lowl}$/J mouse; #016963, The Jackson Laboratory), *Per1*$^{-/-}$:*Per2*$^{-/-}$ (*Per* dKO) mice[45], and *Cry1*$^{-/-}$:*Cry2*$^{-/-}$ (*Cry* dKO) mice[46]. *Pvalb*−2A-Cre, *Vglut2*-ires-Cre, *Per* dKO, and *Cry* dKO mice were derived from ES cells using 3i methods and used in experiments without crossing[66]. All mice used in this study were 3–12 weeks old male to avoid the influence of the estrus cycle in females.

### Plasmids
Except for CN19o expressing vectors without DIO sequences, pAAV plasmids were built using pAAV-hSyn-DIO-hM3Dq-mCherry (Addgene #44361, kindly provided by Dr. Bryan Roth). CN19o expressing vectors without DIO sequences were constructed from another pAAV backbone kindly provided by Dr. Hirokazu Hirai[30]. The promoter/enhancer, reporter/effector, WPRE, and polyA sequences were all present in all pAAV plasmids between ITRs (Fig. 3i). The enhancers were placed upstream of a minimal human β-globin promoter (Addgene #83900, kindly provided by Dr. Gordon Fishell). TeLC, hM3Dq or hM4Di sequence was cloned from pAAV-hSyn-FLEX-TeLC-P2A-EYFP (Addgene plasmid #135391, a gift from Dr. Fan Wang), pAAV-hSyn-DIO-hM3Dq-mCherry, or pAAV-hSyn-DIO-hM4Di-mCherry (Addgene plasmid #44362, a gift from Dr. Bryan Roth). CN19scr (VKEAR-IDGKPDALRGQKQI) or CN19o (KRAPKLGQIGRQKAVDIED) sequence was fused to the C-terminus of mCherry via a (GGGGS)x3 linker. *Camk2b* sequence was cloned from pAAV-hSyn-FLAG-CaMKIIβ used in our previous report[30]. *Camk2a* sequence (NM_177407) was amplified and cloned from C57BL/6 N mouse cDNA. The Infusion HD Cloning Kit (Takara Bio) was used to clone these sequences. Detailed vector information can be found in Supplementary Table 1.

The plasmids were then introduced into competent cells DH5α (TOYOBO) or NEB stable (New England Biolabs). The plasmids were purified using the Pure Yield Plasmid Midiprep System (Promega) and digested with *Xma*I (New England Biolabs) to ensure ITR integrity.

### Adeno-associated virus
In this study, we used an engineered AAV serotype, AAV-PHP.eB[37], and generated AAVs based on the previous reports[67]. AAVpro 293 T cells (632273, Takara Bio) were transfected using Polyethylenimine (PEI, MW 25,000, Polysciences) with three plasmids: pAAV (containing the gene(s) of interest), pUCmini-iCAP-PHP.eB (Addgene #103005, kindly provided by Dr. Viviana Gradinaru), and pHelper (Takara Bio). The medium from cultured AAVpro 293 T cells was collected three days after transfection, and the cells and medium were harvested at least five days later. Centrifugation (2000×*g*, 15 min, and 25 °C) was used to separate the harvested cells and medium. Freeze-thawing lysed the pellets, which were then treated with Benzonase Nuclease (Merck) or TurboNuclease (Accelagen) to digest non-packaged DNA. The lysates were mixed with 8% (wt/vol) polyethylene glycol (PEG MW 8,000, MP Biomedical). AAVs were isolated from the salting-out supernatant using ultracentrifugation (350,000 × *g*, 2 h 25 min, 18 °C; Optima XE-90, Beckman Coulter; Type 70Ti Fixed-Angle Titanium Rotor, Beckman Coulter) with iodixanol (OptiPrep, Sigma). Before use, an AAV solution in Dulbecco's phosphate-buffered saline (DPBS) was concentrated using a centrifuge (AmiconUltra-15, Merck) and stored at 4 °C.

The titration method was adapted from previous reports[67]. Using quantitative PCR (qPCR), AAV titers were calculated as the number of effective vector genomes (vg) in AAV solution. To avoid contamination of non-packaged DNA, a portion of the AAV solution was treated with nuclease (50 U/ml, 1 h, 37 °C; Benzonase Nuclease or TurboNuclease). Proteinase K (30 μg/ml, 1 h, 37 °C; *N*-lauroylsarcosine in 3 mM NaCl, Fujifilm) was then added to the AAV solution to digest the viral capsid and release the viral genome. A general phenol/chloroform extraction and ethanol precipitation method was used to purify the viral genome. AAV titers (vg) were determined using qPCR with SYBR Green Master Mix (Thermo Fisher Scientific). The following primer sequences were used to amplify the target site (WPRE sequence):

WPRE_qPCR_fwd: 5′-CTGTTGGGCACTGACAATTC-3′
WPRE_qPCR_rev: 5′-GAAGGGACGTAGCAGAAGGA-3′

AAV vectors were administered to 6-week-old male mice via retro-orbital injection (Fig. 3i). DPBS was used to adjust the injected AAV solution to the determined titer (0.5–15.0 × 10$^{11}$ vg/mouse, detailed titer information can be found in Supplementary Table 1), and 100–150 μl/mouse was injected. Mice were anesthetized in ambient air with 1%–5% isoflurane prior to the injection. Mice were housed in their home cage for at least >2 weeks prior to behavioral experiments or brain sampling, unless otherwise noted, to allow time for gene expression.

### Respiration-based sleep measurements by SSS
SSS was used to perform sleep phenotyping based on respiration[31]. The recordings were made in either WT male mice (3–12 weeks old) for the developmental analysis (details in below) or AAV-injected mice (8–10 weeks old) for other experiments. To measure respiratory flow, each experimental mouse was placed in the SSS chamber from its home cage. Unless otherwise specified, the recording condition was a 12-hour LD cycle with *ad libitum* food and water. Within two weeks, one set of recordings was completed. To avoid the first-night effect, data from the first day of the recording session were excluded from the analysis. The recordings of circadian KO mice were made under the DD condition for two weeks, and the data from the last six days was analyzed (Fig. 7e).

SSS data were divided into 8-second segments (epochs), with sleep/wake annotation performed after each epoch. The annotation rules and sleep parameters[31], such as sleep duration, $P_{WS}$, and $P_{SW}$, have previously been defined[31]. $P_{WS}$ and $P_{SW}$ are the probabilities in an

observed period from wake to sleep state (the tendency for sleep) and from sleep to wake state (the tendency for wakefulness) (Fig. 1b). Because $N_{XY}$ denotes the number of transitions from X to Y (X, Y ∈ {sleep (S), wake (W)}), the transition probabilities are independent of one another, and are calculated as follows:

$$P_{WS} = N_{WS}/(N_{WS} + N_{WW}) \qquad (1)$$

$$P_{SW} = N_{SW}/(N_{SW} + N_{SS}) \qquad (2)$$

$P_{WS}$ or $P_{SW}$ was treated as 1, when $(N_{WS} + N_{WW})$ or $(N_{SW} + N_{SS})$ was 0 (i.e., consecutive sleep or wake state in an observed period). The duration of a sleep or wake episode is calculated by taking the average of consecutive sleep or wake bouts, which is inversely related to $P_{SW}$ or $P_{WS}$, respectively.

Long-term recording with development lasted approximately 2 months in male mice, from P21 (just after weaning) to P84. During the recording, the SSS chambers were replaced with new ones every week, and sleep/wake annotation was performed on a weekly basis.

## Sleep measurements by EEG/EMG recording

EEG and EMG surgery was performed 4–5 days after AAV injection (6–7 weeks old). A combination anesthetic containing 0.9 mg/kg medetomidine hydrochloride (Domitol, ZENOAQ), 4.8 mg/kg midazolam (Sandos, Novartis), and 6.0 mg/kg butorphanol (Vetorphale, Meiji Seika Pharma) was used to anesthetize mice. Two EEG electrodes were placed in the skull over the cortex (1.5-mm right from midline, 1.5-mm anterior to the bregma; 1.5-mm right from the midline, 1.5-mm anterior to the lambda), and two EMG electrodes were placed in the neck muscle. Insulated wires connected the electrodes and were soldered to a pin header that was secured to the skull with dental cement. Mice were given 0.9 mg/kg atipamezole hydrochloride (Antisedan, ZENOAQ) after surgery and warmed to 37 °C using a warming plate to promote anesthesia recovery. The mice were given at least ten days to recover from surgery.

Amplified EEG/EMG signals from 8-week-old male mice were recorded in the EEG/EMG chamber with food and water using Vital Recorder software (KISSEI COMTEC). For both EEG and EMG, the sampling frequency was 128 Hz. One recording session lasted less than a week. Data analysis began the day after data was collected. For further analysis, the EEG/EMG data recorded in KCD format was converted to EDF format.

For sleep recording, an in-house improved version of the FASTER method was used to automatically annotate the stages (wakefulness, REM, and NREM sleep) from EEG/EMG data[30]. EEG/EMG data were divided into 8-second segments (epochs), and annotation was performed after each epoch. Following FASTER's auto-staging, each epoch was manually corrected using the following criteria. The low-amplitude EEG and high amplitude EMG were used to determine wakefulness. NREM sleep was distinguished by high amplitude EEG with a low frequency wave (0.5–4 Hz) and a low EMG signal. The low-amplitude EEG with predominantly theta wave (6–10 Hz) and EMG atonia was used to stage REM sleep.

As previously reported[30], transition probabilities between wakefulness, NREM sleep, and REM sleep were calculated in the same way as $P_{WS}$ and $P_{SW}$ (see "Methods" section for "Respiration-based sleep measurements by SSS"). The transition from wakefulness to REM sleep ($P_{WR}$) was not depicted because it occurred so briefly.

For each epoch, the power spectral density (PSD) was calculated using fast Fourier transformation and Welch's averaging method. The spectral analysis excluded epochs that contained movement artifacts. Each PSD bin was calculated by averaging within each behavioral state and was expressed as a percentage of the total power over all frequencies. Delta or SO power was calculated during NREM sleep as

percentages of delta (0.5–4 Hz) or SO (0.5–1 Hz) power over total EEG power per hour or day.

## Sleep measurements with SD

SD was performed during ZT0–6 (the first half of light phase) in SSS recordings using an automated orbital shaker (TAITEC). Every mouse was moved from an SSS chamber to new cage on the shaker at ZT0 on the fourth day of recording. The shaker was programmed to turn on for 30 s every 80 s (30 s on, 50 s off) at around maximum speed of around 120 rpm. Following the SD session (at ZT6), each mouse was returned to the same SSS chamber as before, and the recording was resumed. The basal recording was made three days before the SD session.

In the EEG/EMG recordings, mice underwent SD for 6 h from ZT0 to ZT6. SD was achieved by gently touching their bodies with a brush when they became immobile and assumed a sleeping posture. The day preceding the SD session corresponded to the basal recording.

## Sleep measurements with pharmacological administration

Natural saline was used to dissolve all chemicals used in pharmacological administration experiments (Otsuka pharma). Adjusting for individual mouse body weight, the single volume of injection was in the range of 150–250 µl. For MK-801 administration experiments, mice were given 0 or 2 mg/kg MK-801 maleate (Merck) i.p. injection at ZT8, with at least 1 day between injections.

For chemogenetic experiments, clozapine-*N*-oxide (CNO, Tocris) was used as the ligand of $G_q$-coupled DREADD hM3Dq and $G_i$-coupled DREADD hM4Di. For chemogenetic manipulation experiments under *ad libitum* sleep, mice expressing EGFP, hM3Dq, or hM4Di under the E11 enhancer were given i.p. injection of 0, 1, 3, or 5 mg/kg CNO at ZT12, with at least 2 days between injections. Mice expressing mCherry or hM4Di under the E11 enhancer were given i.p. injection of 0 or 0.25 mg/kg CNO at ZT12 for chemogenetic suppression experiments after SD using the above-mentioned method (see "Sleep measurements with SD" in "Methods" section).

## Whole-brain clearing, immunostaining, imaging, and analysis

An improved version of the CUBIC-HV methods was used to clear and stain mouse brains[34]. Mice were sedated with 100 mg/kg pentobarbital before being perfused transcardially with PBS and 4% paraformaldehyde (PFA). The brains were post-fixed in 4% PFA overnight at 4 °C and stored in PBS until the clearing processes began. CUBIC-L (10% [wt/wt] *N*-butyldiethanolamine, 10% [wt/wt] Triton X-100 in water) was used to dilapidate the brains. Nuclear staining and immunostaining were performed after delipidation using nuclear staining dye (SYTOX Green, Thermo Fisher Scientific) and fluorescence-dye-conjugated antibodies. CUBIC-R+ (45% [wt/wt] antipyrine, 30% [wt/wt] Nicotinamide, 0.5% [vol/vol] *N*-butyldiethanolamine in water) was used to process brains for refractive index matching, followed by embedding in 1.5% agarose gel in CUBIC-R +.

All whole-brain imaging was performed at International Research Center for Neurointelligence in the University of Tokyo. A custom-made light-sheet microscope was equipped with diode or DPSS lasers with wavelengths of 488, 532, 594, and 642 nm (Coherent). To ensure homogeneous resolution in XYZ axes, the microscope employed an axially-swept light sheet mechanism[68]. The microscope was fitted with a 0.63X macro-zoom objective lens (Olympus) and variable 0.63-6.3X intermediate zoom optics (Olympus). The fluorescence signal was efficiently captured using an sCMOS camera (PCO), paired with appropriate fluorescence filters. In all conducted experiments, the voxel size was (X, Y, Z) = (6.45, 6.45, 7.0) µm with 1.6X intermediate zoom optics, providing total 1.0X magnification. An effective resolution should consider the moderate tissue expansion (1.5X) induced by the CUBIC-R+ treatment.

Image analysis procedures on CUBIC-Cloud included cell detection, brain registration, and data quantification/visualization[33]. The ilastik software was used to detect cells in 3D brain image data. Manual annotations were prepared from at least two brains with identical labeling conditions in some parts of the 3D images using ITK-SNAP software, and the classifier was trained using the manual annotation data. The XYZ position, mean fluorescence intensity, and volume of single cells at the whole brain level were obtained in CSV format by applying the trained classifier to the whole brain 3D image. The symmetric image normalization (SyN) algorithm was used to register the 3D brain image. Individual brains in CUBIC-Cloud were registered to CUBIC-Atlas brains using nuclear staining images and single cells displayed regional information[69]. The CUBIC-Cloud platform (https://cubic-cloud.com/) was used to perform some data quantification and visualization.

### Whole-brain analysis of PV profile with development

B6N WT male brains (P21, P28, P35, P42, P84, and P270; $n = 6$ for each age) were cleared, stained, imaged, and analyzed (see "Whole-brain clearing, immunostaining, imaging, and analysis" in "Methods" section). PV antibody (1:50, PV235, Swant; with anti-mouse IgG1 secondary Fab fragment conjugated with A594 [1:150, 115-587-185, Jackson]) and SYTOX Green were used to stain the mouse brains.

### Whole-brain analysis of c-Fos profile with sleep-wake perturbation

B6N WT male mice (8 weeks old; $n = 6$ for S and SD groups) were individually housed in an SSS camber for 3 days prior to SD session for experiments. On the fourth day, mice in the SD group were sleep-deprived as described above (see "Sleep measurements with sleep deprivation" in "Methods" section), whereas mice in the S group were allowed to sleep *ad libitum* as much as they wanted. At ZT6 (just after the SD session), all mouse brains were sampled.

B6N WT male mice (8 weeks old; $n = 4$ for saline and MK-801 groups) were individually housed under the DD condition in an SSS camber for 3 days prior to i.p. administration of MK-801. On the fourth day, mice received i.p. injection of 0 or 2 mg/kg MK-801 maleate at CT2. Mouse brains were harvested two hours after i.p. injection (CT4). To prepare brains for CaMKII activation in cortical PV neurons, the AAV.PHP.eB vector transducing E11-CaMKIIα (WT) or E11-CaMKIIα (T286D) was injected via retro-orbital sinus into B6N WT male mice (6 weeks old; $n = 4$ for each group). The mice were individually housed in an SSS camber for 1 week when they were 8 weeks old, to monitor their sleep phenotype. Following sleep phenotype, the brains of mice were collected at ZT6. This procedure was employed to prepare the brains for CaMKII activation in *Vglut2* neurons. Two types of AAV.PHP.eB vectors, which carried H2B-mCherry and either CaMKIIα (WT or T286D) in a Cre-dependent manner, were injected into *Vglut2*-ires-Cre male mice (6 weeks old; $n = 3$ for each group). After 2 weeks, their sleep phenotype was recorded for one week using SSS. Subsequently, their brains were sampled at ZT6.

In all c-Fos analysis, collected mouse brains were cleared, stained, imaged, and analyzed as described above (see "Whole-brain clearing, immunostaining, imaging, and analysis" in "Methods" section). c-Fos antibody (1:26, 2250S, CST; with anti-rabbit IgG secondary Fab fragment conjugated with Alexa 647 [1:80, 111-607-008, Jackson]), PV antibody (1:20, PV235, Swant; with anti-mouse IgG1 secondary Fab fragment conjugated with A594 [1:80, 115-587-185, Jackson]), and SYTOX Green were used to stain the mouse brains. The c-Fos+PV+ cells were identified by a search for cells characterized by the presence of both c-Fos and PV signals (Supplementary Fig. 4a). We previously confirmed that the accuracy of the cell detection is more than 80% precision for both PV+ and c-Fos+ cells in most of brain regions[33]. The accuracy is relatively lower in the cerebellar regions so that we excluded the regions from the analysis. Cell counting was performed

independently for each channel, and the center of mass of the detected cells was determined. For each c-Fos+ cells, if a PV+ cell was found within a distance of 16 µm[69], it was considered a c-Fos+PV+ cell. Note that due to tissue expansion by a factor of ~1.5, this distance equated to approximately 10.5 µm in untreated tissue. This procedure was applied for the identification of PV+mCherry+ or c-Fos+mCherry+ cells.

To evaluate the colocalization analysis using the light-sheet microscope, selected cortical regions of the cleared brain were also imaged using a confocal microscope (BX61WI, Olympus) equipped with a 25X objective lens (NA: 1.00, WD: 8 mm, Olympus) and 559- and 635-nm lasers (NTT Electronics). The voxel size was (X, Y, Z) = (0.497, 0.497, 0.497) µm under the condition of no tissue expansion. Colocalization of c-Fos and PV signals was confirmed in images obtained from both types of microscopes in a three-dimensional manner (Supplementary Fig. 4b).

### In vitro assay for CaMKIIα expression and kinase activity

HEK293T cells (CRL-3216, ATCC) were transfected with pMU2-CMV-FLAG-CaMKIIα plasmids using PEI. After 24 h, the medium of cultured HEK293T cells was replaced with fresh culture medium. After 60 h, the cells were lysed with 1 ml of cell lysis buffer (50 mM HEPES-NaOH pH 7.6, 150 mM NaCl, 0.5 mM $CaCl_2$, 1 mM $MgCl_2$, and 0.25% [v/v] NP-40) supplemented with 1% phosphatase inhibitor cocktail (Nacalai) and homogenized using the homogenizer (Hielscher). Cell lysates were collected and kept at −80 °C.

Dot blot was used to estimate the relative expression levels of CaMKIIα in each cell lysate. Each cell lysate was diluted four times (1/4), then spotted on PVDF membrane (Merck). The membranes were reacted to anti-CaMKIIα antibody (1:3,000, SMC-124D, StressMarq) after blocking with Blocking One (Nacalai), followed by the reaction to the secondary antibody: anti-mouse IgG peroxidase (HRP)-conjugated antibody (1:3,000, W402B, Promega). HRP chemiluminescent reaction (Clarity Western ECL Substrate; ChemiDoc XRS+ system, Bio-Rad) was used to detect immunoreactivity of the blotted proteins. To ensure that the dot blot signals were within the linear range of detection, a cell lysate expressing the WT CaMKIIα was serially diluted (1, 1/2, 1/4, 1/8, 1/16, 1/32, 1/64, or 1/128) was spotted on the same membrane (data not shown).

The mobility shift assay was used to determine the CaMKII kinase activity in each cell lysate (LabChip EZ Reader II, PerkinElmer). Each diluted CaMKIIα cell lysate (1, 1/2, 1/4, 1/8, 1/16, 1/32, 1/64, or 1/128) was mixed with 0.25 mM ATP and 3.7 µM Fluorescently labeled peptide substrate 11 (PerkinElmer) in the presence or absence of 0.50 µM CaM (Merck). The mixture was incubated at 37 °C for 10 min, then at 98 °C for 10 min to stop the reaction. CaMKII kinase activity is calculated by dividing the phosphorylated peptide signal by the total substrate peptide signal.

### Liquid chromatography with tandem mass spectrometry

B6N WT male mice (6 weeks old) were injected with the AAV.PHP.eB vector transducing E11-CaMKIIα (T285D) via retro-orbital sinus. The E11-CaMKIIα (T285D) mice were individually housed in an SSS camber for 1 week after they were 8 weeks old to monitor their sleep phenotype. The mice in the SD group were then sleep-deprived using the method described above (see "Sleep measurements with SD" in "Methods" section), whereas mice in the S group were allowed to sleep *ad libitum* as much as they wanted. The mice were killed by cervical dislocation at ZT6 (immediately following the SD session), and their cortical regions were quickly dissected and frozen in liquid nitrogen ($n = 21$ for each group). The brain tissues were stored at −80 °C to preserve phosphorylation state.

The brain tissues were lysed and digested for LC-MS/MS analysis as previously described[30]. The frozen sample crusher (Tokken) was used to powder the brain tissues, and 5–10 mg of brain powder was homogenized in 0.5 ml of Solution B buffer (12 mM Sodium

deoxycholate, 12 mM *N*-lauroylsarcosine sodium salt, 50 mM Ammonium hydrogen carbonate) supplemented with 1% phosphatase inhibitor cocktail (Nacalai), preheated at 98 °C. Following a 15-minute incubation at 98 °C, the samples were reduced with 9 mM dithiothreitol (Fujifilm) at room temperature for 30 min before being alkylated with 45 mM iodoacetamide (Merck) at room temperature for 30 min. The samples were then digested overnight at 37 °C with 1 μg/ml of lysyl endopeptidase (Lys-C) (Fujifilm) and for 3–6 h with 0.8 μg/ml of trypsin (Roche). Following digestion, the sample was acidified with 0.5% TFA and an equal volume of ethyl acetate was added. The sample was thoroughly mixed before being centrifuged (1500×*g*, 30 min, 25 °C) to separate the peptide solution. Using a SpeedVac (Thermo Fisher Scientific), the peptide solution (an aqueous layer) was collected and dried.

The dried peptides were dissolved in a solution of 2% acetonitrile and 0.1% TFA. Each peptide solution was divided into two groups: one for individual samples, and the other for a mixture of each peptide solution as an internal control. Using a Sep-Pak C18 cartridge (Waters), the individual samples and internal control were trapped and desalted. The peptides on the cartridge were then dimethyl-labeled as previously described[70], with formaldehyde (CH$_2$O; Nacalai) added to the internal control (light label) or isotope-labeled formaldehyde (CD$_2$O; Cambridge Isotope Laboratories) added to the individual samples (medium label) together with NaBH$_3$CN (Merck). The dimethyl-labeled peptides were eluted with a solution of 80% acetonitrile and 0.1% TFA. After that, an equal amount of light-labeled internal control was added to each medium-labeled individual sample, allowing us to compare the relative amount of peptides in the individual samples using internal control as a standard.

A portion of the mixture (10%) was subjected to LC-MS/MS analysis to determine the amount of CaMKIIα and total proteins. The remaining mixture (90%) was used to enrich the phosphorylated peptides using the High-Select Fe-NTA Phosphopeptide Enrichment Kit (Thermo Fisher Scientific) according to the manufacturer's protocol. All samples were SpeedVac dried before being dissolved in 2% acetonitrile and 0.1% TFA.

The samples were analyzed using an LC-MS system with Orbitrap Exploris 480 (Thermo Fisher Scientific) equipped with a nano HPLC system (Advance UHPLC; Bruker Daltonics) and an HTC-PAL autosampler (CTC Analytics) with a trap column (0.3 × 5 mm, L-column, ODS, Chemicals Evaluation and Research Institute). A gradient was used to separate samples using mobile phases A (0.1% formic acid/H$_2$O) and B (0.1% formic acid and 100% acetonitrile) at a flow rate of 300 nl/ minute (4%–32% B for 190 min, 32%–95% B for 1 min, 95% B for 2 min, 95%–4% B for 1 min, and 4% B for 6 min) with a home-made capillary column (length of 200 mm and inner diameter of 100 μm) packed with 2-μm C18 resin (L-column2, Chemicals Evaluation and Research Institute). Several samples were separated using a different gradient (4%–36% B for 55 min, 36%–95% B for 1 min, 95% B for 5 min, 95%–4% B for 1 min, and 4% B for 8 min) without the phosphorylated peptide enrichment process. The eluted peptides were electrosprayed (2.1 kV) and subjected to positive ion mode analysis. Full MS spectra were obtained with a scan range of 350–1800 m/z and resolution of 60,000 FWHM at 200 m/z. MS2 spectra with 7,500 FWHM resolution at 200 m/z and a defined first mass at 120 m/z were obtained. Without the phosphorylated peptide enrichment process, samples were analyzed by data-dependent MS/MS acquisition, in which precursor ions (excluding isotopes of a cluster) above 5.0e3 intensity threshold with charge state ranging from 2+ to 7+ were selected for the MS/MS acquisition at 2.0 m/z isolation window during the 3 s interval between every full MS spectra acquisition. A dynamic exclusion of 20 s was used. Samples from the phosphorylated peptide enrichment process were analyzed by targeted MS/MS acquisition, where

precursor ions matched with the dimethyl-labeled peptide, which included the phosphorylated T286 residue derived from the CaM-KIIα (WT) and CaMKIIα (E285D) (i.e., light-WT 564.754 m/z, medium-WT 568.779 m/z, light-E285D 557.748, medium-E285D 561.773; z = 2 + ).

The raw data obtained were subjected to database searches (UniProt, reviewed mouse database as of October 5th, 2022) using the Sequest HT/Percolator algorithm running on Proteome Discoverer 2.5 (Thermo Fisher Scientific). Peptide cleavage was set to trypsin; missed cleavage sites were allowed to be up to two residues; peptide lengths were set to 6–144 amino acids; and mass tolerances were set to 40 ppm for precursor ions and 0.08 Da for fragment ions. Fixed modification included carbamidomethylation at cysteine and dimethylation [H (4) C (2), or 2H (4) C (2)] at lysine and the peptide N-terminus. Methionine oxidation was designated as a variable modification. A significance level of $q < 0.01$ was used. Precursor ion abundances for the phosphorylated T286 residue were manually checked and quantified using Qual browser function of the Xcalibur 4.4 software (Thermo Fisher Scientific). Proteome Discoverer 2.5 was used to calculate the abundances of the other precursor ions (Thermo Fisher Scientific).

### Statistical analysis

Microsoft Excel for Mac version 16.49, R version 3.6.1., and Python version 3.8.2 were used for statistical analyses. For all tests, the significance level was set at <0.05 (*$P < 0.05$, **$P < 0.01$, ***$P < 0.001$, n.s. = non-significance). The Fig. legends describe the statistical method and results for each comparison. The statistical analysis procedures were as follows.

To assess normality and homoscedasticity in two samples, the Shapiro test and F-test with a significance level of 0.05 were used. When two groups had normal distributions with equal variance, the Student's *t*-test was used for independent samples; and the Student's paired *t*-test was used for paired samples. When two groups had normal distributions with unequal variance, Welch's *t*-test was used; otherwise, two-samples Wilcoxon test was used.

To evaluate normality and homoscedasticity in more than three samples, the Kolmogorov–Smirnov test (KS-test) and Bartlett's test at the significance level of 0.05 were used. When all groups were normally distributed with equal variance, the Student's *t*-test with Bonferroni correction was used; when two groups had normal distributions but not equal variance, the Welch's *t*-test with Bonferroni correction was used; otherwise, the Steel test was used.

### Reporting summary

Further information on research design is available in the Nature Portfolio Reporting Summary linked to this article.

## Data availability

Protein mass spectrometry raw data have been deposited in MassIVE repository (MSV000095087). Due to the large size of raw data, other data used in this study is available from the corresponding author upon request. Source data are provided with this paper.

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

## Acknowledgements

We would like to express our gratitude to S. Tomita, A. Shimokawa, and H. Ono for their assistance with the SSS and EEG/EMG recordings; A. Shimokawa, K. Yamashita, R. Narumi, K. Yamamoto, M.N.B. Roig, and F. Kinoshita for their help in SD experiments; T. Miyawaki, S. Sato, and K. Shimizu for assisting with AAV production; Y. Saito for assisting with brain sample preparation; M. Kuroda for LSFM imaging technical assistance; D.R. Weaver and W. Nakamura for providing *Per* dKO mice; G.T.J. van der Horst and J.H.J. Hoeijmakers for providing *Cry* dKO mice; and T. Jin for supporting with NMR analysis. We would like to thank Enago (www.enago.jp) for the English language review. This work was supported by a Grant-in-Aid for JSPS Fellows (JSPS KAKENHI, 19J22074 and 23KJ0234, to K.K.); a Grant-in-Aid for Scientific Research (C) (JSPS KAKENHI, 20K06576 and 23K05738, to K.L.O.); a Grant-in-Aid for Early-Career Scientists (JSPS KAKENHI, 20K15766, to Y.S.); a Grant-in-Aid for Scientific Research (B) (JSPS KAKENHI, 22H02824, to E.A.S.); AMED-PRIME (JP20gm6210027, to E.A.S.); a Grants-in-Aid from the Takeda Science Foundation, Nakatani foundation for advancement of measuring technologies in biomedical engineering, and Uehara Memorial Foundation (to E.A.S.); a Grant-in-Aid for Scientific Research (S) (JSPS KAKENHI, 18H05270, to H.R.U.); Human Frontier Science Program (HFSP) Research Grant Program (RGP0019/2018, to H.R.U.); Exploratory Research for Advanced Technology (ERATO) (JST, JPMJER2001, to H.R.U.); Quantum Leap Flagship Program (Q-LEAP MEXT, JPMXS0120330644, to H.R.U.); Brain Mapping by Integrated Neuro-technologies for Disease Studies (Brain/MINDS) (AMED, JP21wm0425003, to E.A.S.; JP21dm0207049, to H.R.U.); Innovative Drug Discovery and Development (AMED, JP19am0401011, to H.R.U.); and an intra-mural Grant-in-aid (RIKEN BDR, to H.R.U.).

## Author contributions

K.K., K.L.O., and H.R.U. designed the study; S.Shi, and E.A.S. helped with the study design; K.K., H.F., D.T., and R.G.Y. performed the SSS recording; K.K., H.F., R.R.T., S.Shiono, K.Matsuzawa, and R.G.Y. performed the EEG/EMG recording; K.K., K.L.O., and R.G.Y. analyzed the sleep data; K.K., K.L.O., D.T., and S.Y. designed and constructed the plasmids; K.K., R.R.T., and C.S. produced AAVs; K.K., T.M., R.R.T., C.S., and S.Shiono performed the brain sample preparation, LSFM imaging, and data analysis; S.Y.Y., R.R.T., and S.Shiono performed the confocal imaging; K.K., K.L.O., and R.R.T. performed biochemical experiments and analyzed the data; Y.S. synthesized CNO for preliminary chemogenetic experiments; R.G.Y. established the improved FASTER methods for EEG/EMG sleep staging; E.A.S. developed the improved version of the CUBIC-HV methods for whole-brain immunostaining; D.T., J.Y.G., M.K., K.Miyamichi., K.S., and H.K. established the ES-cell-derived mice; K.K. prepared the figures and draft manuscript; K.K., K.L.O., and H.R.U. wrote the manuscript with input from all co-authors.

## Competing interests

R.G.Y. and E.A.S. are employees of CUBICStars, Inc. H.R.U. is a founder and CTO of CUBICStars, Inc., which provides and maintains CUBIC-Cloud web service. The remaining authors declare no competing interests.

## Additional information

[1]Department of Systems Pharmacology, Graduate School of Medicine, The University of Tokyo, Bunkyo-ku, Tokyo, Japan. [2]Laboratory for Synthetic Biology, RIKEN Center for Biosystems Dynamics Research (BDR), Suita, Osaka, Japan. [3]Department of Information Physics and Computing, Graduate School of Information Science and Technology, The University of Tokyo, Bunkyo-ku, Tokyo, Japan. [4]Department of Systems Biology, Institute of Life Science, Kurume University, Kurume, Fukuoka, Japan. [5]Laboratory for Animal Resources and Genetic Engineering, RIKEN Center for Biosystems Dynamics Research (BDR), Chuou-ku, Kobe, Hyogo, Japan. [6]Laboratory for Comparative Connectomics, RIKEN Center for Biosystems Dynamics Research (BDR), Chuou-ku, Kobe, Hyogo, Japan. [7]Laboratory for Mouse Genetic Engineering, RIKEN Center for Biosystems Dynamics Research (BDR), Suita, Osaka, Japan. [8]Present address: Kennedy Krieger Institute, Solomon H. Snyder Department of Neuroscience, Johns Hopkins University School of Medicine, Baltimore, MD, USA. [9]Present address: Computational Neuroethology Unit, Okinawa Institute of Science and Technology, Onna, Okinawa, Japan. [10]Present address: International Institute for Integrative Sleep Medicine (IIIS), University of Tsukuba, Tsukuba, Ibaraki, Japan. [11]Present address: Department of Molecular Embryology, Research Institute, Osaka Women's and Children's Hospital, Izumi, Osaka, Japan. [12]Present address: Division of Molecular Psychoimmunology, Institute for Genetic Medicine, Graduate School of Medicine, Hokkaido University, Sapporo, Japan. [13]Present address: Laboratory of Animal Genetics and Breeding, Graduate School of Bioagricultural Sciences, Nagoya University, Chikusa-ku, Nagoya, Japan. [14]Present address: Department of Biochemistry and Systems Biomedicine, Juntendo University Graduate School of Medicine, Bunkyo-ku, Tokyo, Japan. ✉e-mail: uedah-tky@umin.ac.jp

