## [Peer Review File · Nature Communications]

Cortical parvalbumin neurons are responsible for homeostatic sleep rebound through CaMKII activationREVIEWER COMMENTS

Reviewer #1 (Remarks to the Author):

Kon and colleagues present a large set of experiments from which they conclude that the activity of parvalbumin inhibitory neurons in cortex plays an important role in the homeostatic regulation of sleep, during development and in adulthood. The paper includes many experiments in young and adult animals. Unfortunately, in most experiments the tools used to support the main claim, and the experimental design, make the interpretation of the results difficult at best. Many experiments are very difficult to interpret, and their number cannot compensate for lack of rigorous tools and design to address the question of interest. In general, these results are not put into the larger context of the many studies that have already examined the homeostatic regulation of sleep during early sensitive periods, and the general role of PV cells in sleep. Sleep-dependent effects on the balance excitation/inhibition in the cortex, including in PV cells, are well established, but are not cited and discussed here. Calcium imaging and EGG/LFP/unit recordings show that PV cells are more active in sleep, especially REM sleep, and their chemogenetic block decreases SWA activity. This evidence is either not cited, or cited but not discussed.

The first part of the study focuses on the developmental period between early adolescence to adulthood. The authors describe changes in some sleep/wake patterns and response to sleep deprivation, and they ascribe these changes to changes in PV cells. However, two major technical issues discussed in detail below (the use of Snappy Sleep Stager to distinguish sleep and wake stages; the lack of any method to assess the efficacy of the sleep deprivation procedure) seriously limit the significance of the results. Also worrisome is the fact that the authors assume that a change in PV cell number will lead to changes in the inhibitory control of PV over pyramidal neurons, without using any functional measure of inhibition. In other words, it is not obvious why fewer PV cells would lead to a change in inhibition in cortex. The first part of the study is weak, unless the experiments are repeated after addressing critical technical and functional issues. The second part of the paper focuses on SD experiments in adults and suffers from the same technical and design limitations, including the apparent choice of focusing on a late phase of the rebound after SD, and the lack of EEG-based evidence, as discussed below.

Major issues

The Snappy Sleep Stager is unconventional and needs to be described and validated here. As stated in the cited paper, this system cannot distinguish NREM sleep from REM sleep. Obviously, it also cannot measure “depth” of sleep (SWA). In the current study, which claims to analyze the development of sleep homeostasis, it is puzzling that the authors rely almost exclusively on SSS for their experiments. Many

previous studies have found that in developing rodents, it is not the total duration of sleep but NREM vs REM amounts that change. Moreover, previous studies have found that sleep rebound is mainly/only in terms of sleep duration in the least mature animal (~P20-25), while later it mainly consists of changes in SWA. Given the complexity of these very dynamical changes, any claim about developmental changes in sleep homeostasis need to be supported by EEG/LFP sleep/wake recordings. The readout of “sleep fragmentation” using the probability w_s or w_w only based on SSS, which is used for all experiments (e.g. silencing of E11 cells) is not enough.

The second major issue is that sleep deprivation was done in animals that were not recorded (no SSS, no EEG), thus we do not know whether the procedure was equally effective in very young or more mature animals. It is very possible, in fact likely, that this was not the case. This evidence needs to be provided, even more so because the method used 30sec stim, 50 sec off; this method mainly fragments sleep, often without actually changing the total time spent asleep. In short, the smaller sleep rebound at 4 weeks may be due to less effective sleep deprivation at that age. This possibility needs to be formally tested.

The rationale to link PV cells density to changes in sleep homeostasis is weak at best. The underlying assumption is that the extent to which pyramidal neurons are under inhibitory control is changing, but cell number is not a proxy for the efficacy of inhibition. Thus, I do not find this experiment to be informative.

The authors use the AAV-mediated approach via E11 to target PV neurons. They should clearly state why they did not use cre-dependent mouse lines in which cortical PV cells are labeled. Wouldn't this approach target more cells than the AAV-mediated approach? Please clarify the rationale. The effects of the silencing of PV cells via TeLI experiments are measured only via the SSS method (w_s and w_w transitions) and therefore are very difficult to interpret for the reasons explained above. Moreover, I may have missed it, but I did not see any direct evidence that synaptic transmission is actually blocked in PV cells using this AAV approach.

The design of the chemogenetic experiment (E11-M3 or M4 mice) is difficult to follow: as shown in figure 4, why is CNO given only 6 hours after the end of SD? Extensive previous evidence (based on EEG recordings) showed that most of the rebound in terms of duration and depth of sleep has dissipated after 6 hours. To make the case that PV neurons can interfere with the rebound, they should be excited/inhibited while the rebound is occurring, i.e. immediately after SD. Similar issue for figure 3: why are PV cells only chemogenetically activated or inhibited in the dark, while mice mainly sleep during the day?

A similar issue applies to the experiments to block CamKII activity in PV cells. What is the evidence that the activity is indeed blocked? Is this effect assumed to be continuous? Again, the authors find an effect

on the rebound after SD, but they rely only on the SSS method. When the authors finally measure EEG in these mice, they find signs of sleep fragmentation, but apparently do not repeat the SD experiment to support their claim that sleep rebound is reduced after blocking CamKII activity. It is difficult to understand why the SD experiment was not done. Moreover, Figure 6b shows that for the first 6 hours after SD the recovery of sleep occurs to the same extent. How do the authors explain this?

Similar major issues apply to the experiment with presumed increase in CamKII activity (not shown). First, the first claim about longer sleep in T286D mice during baseline is validated with EEG recordings, but the authors again do not study the response to sleep deprivation with EEG recording, which is puzzling. The assumption is that increased CamKII activity makes PV cells more active, resulting in a “sleep rebound” state even without sleep deprivation. However, a previous study showed that chemogenetic PV cell activation increases NREM duration but at the same time decreases delta activity, which is not consistent with a physiological sleep rebound.

The Fos experiment in figure 7 finds that the presumed increase in CamKII activity leads to more PV cells labeled with Fos. Is this Fos expression present both after sleep and after waking?

The rationale for focusing on CamKII in PV is not straightforward and should be clarified. The cited studies focus on CamKII on glutamatergic synapses. In the final experiment (Fig. 7g), T286 phospho levels are measured after SD relative to control in the whole brain. While suggestive, this experiment is not direct proof that CamKII activity is increased, as the authors acknowledge.

Minor:

lines 60-62: please fix the sentence

line 78: may be responsible

line 92: what is “sleep stabilization”? please clarify

Reviewer #2 (Remarks to the Author):

The manuscript Cortical parvalbumin neurons are responsible for homeostatic sleep rebound through CaMKII activation by Kon and colleagues examines the role of parvalbumin (PV) expressing neurons in

the maintenance of homeostatic regulation of sleep rebound. The authors propose as a model that CaMKII kinase modulates the activity of cortical PV neurons to regulate sleep rebound.

Overall, I found this manuscript enjoyable to read with an intriguing narrative. The manuscript is well written and comprised of a series of interesting and logical experiments. The data analysis is well performed and robust throughout. Despite this overall positive impression, I have some concerns, outlined below, about the reliability of some of the experiments.

Major concerns

1. My major concern with the manuscript relates to the whole brain immunolabeling/lightsheet imaging. Although this is a flashy technique and potentially one that might aid with high throughput analysis, it is one that clearly compromises the reliability of the data output. The imaging criteria are not well documented in the methods (this should be improved/expanded) but the authors state the use of a 0.63X objective lens. This magnification, especially combined with the inherent shortcomings of tissue clearing and lightsheet microscopy, would be inappropriate for co-localization analysis.

2. Related to the above; in extended data figure 4, there are data points suggesting 100% of c-fos+ cells are PV+. i.e. No neurons other than PV cells are active. This alone seems nearly if not completely impossible. Coupled with other data points in the same group showing 0% of c-fos+ cells are PV+ demonstrates that there is something highly unreliable about the co-localization data. My suspicion is that this relates to the tissue clearing and low magnification imaging as described above. Adequate demonstration of fos/pv co-localization will require the authors to perform some standard tissue sectioning/staining and high magnification high resolution imaging.

3. Also related to the above concerns, the authors don't include any real imaging to show colocalization and confirm that these techniques are capable of reliably giving colocalization data and that the labeling was specific.

4. Validation of the ilastik cell counts. Can the authors confirm the accuracy of the cell counts with this method? How reliable is it in each of the brain regions the authors are examining? This must be confirmed to be reliable. It seems likely that superficial imaging would be more reliable than deeper imaging would be.

5. The normalized c-fos counts (SD/SLP groups) included in figure 3 are fine as a data reduction for analysis but the non-normalized data should also be included in the extended data set for transparency.

6. The authors show very high expression of CaMKII in PV+ interneurons using scRNA-seq. It's important to not here that this is mRNA. Can the authors demonstrate with immunolabeling what Percentage of these cells actually express CamKII protein?

7. Is c-fos a reliable marker for activity in experiments where CaMKII is manipulated? It has been previously demonstrated that inhibition of CaMKII impairs fos expression as it is part of the c-fos induction cascade (e.g. <https://doi.org/10.1074/jbc.M412680200>). Therefore, there may be changes in c-fos simply as a result of disrupting this pathway without having a direct effect of neuronal activity per se. A CaMKII independent activity marker would be more appropriate.

Reviewer #3 (Remarks to the Author):

The homeostatic regulation of sleep is characterized by rebound sleep after prolonged wakefulness, but the molecular and cellular mechanisms underlying this regulation are still unknown. Kon et al., showed that CaMKII-dependent regulation of PV-expressing cortical neurons is involved in sleep homeostasis regulation. Prolonged wakefulness enhances cortical PV-neuron activity. Chemogenetic suppression or activation of cortical PV neurons inhibits or induces rebound sleep, implying that rebound sleep is dependent on increased activity of cortical PV neurons. CaMKII kinase activity boosts the activity of cortical PV neurons and is important for homeostatic sleep rebound. The authors propose that CaMKII-dependent PV neuron activity represents negative feedback inhibition of cortical neural excitability, which serves as the distributive cortical circuits for homeostatic sleep regulation.

Main comments:

This is an important topic for sleep research. Overall, the experiments are carefully designed and executed, and the results are interesting and generally supportive of their conclusions. However, the authors need to address the following concerns.

1. The authors argued that cortical PV neurons are responsible for homeostatic sleep regulation, but their experiments mainly used the E11 enhancer for PV-specific genetic and chemogenetic manipulations. However, the E11 promoter not only drive expression in cortical PV neurons, but also PV and non-PV neurons in other brain regions. Thus, I think there are alternative explanations that non-PV neurons or other brain regions might be responsible for the observed sleep phenotypes. I suggest that the authors try to use either PV-Cre mice or the E22 promoter, which drives more restricted expression

in the cortex than E11 promoter (Vormstein-Schneider et al., PMID: 32807948), for manipulations to confirm that cortical PV neurons are specifically required for homeostatic sleep response.

2. NREMS delta power is an important aspect of homeostatic sleep regulation. Since NREMS delta power increases in proportion to the duration of prior wakefulness, it is considered a good index of sleep need. However, the authors seldomly used EEG/EMG recording to measure dynamic changes of NREMS delta power during the 24-h cycle or after sleep deprivation. I think analysis of NREMS delta power needs to be improved in the revision, particularly in Fig 4 and Fig 5o-p, using a hourly plot of NREMS delta power will indicate dynamic changes of delta power after sleep deprivation.

3. The authors argued that CaMKII activity in E11 neurons encoded sleep need based on observation that sleep deprivation resulted in increase of CaMKIIa T286 phosphorylation. I think this conclusion is an over-interpretation of the data because much more are required to claim that something encodes sleep need. At the minimum, the degree of increase/decrease of CaMKII activity should be in proportion to the increase/decrease of sleep need, as measured by either NREMS amount and/or NREMS delta power, during 24-h cycle or after sleep-deprivation.

4. The authors used "Slp" to refer to normal slept mice as opposed to sleep-deprived (SD) mice. I suggest that the authors change Slp because it was previously used as abbreviation for Sleepy mutant mice.

Response to Reviewers' Concerns for Kon et al., "*Cortical parvalbumin neurons are responsible for homeostatic sleep rebound through CaMKII activation*".

We thank the reviewers for their constructive and insightful comments. We are excited that the reviewers provided positive comments about the finding of rebound sleep control driven by the CaMKII activity in the cortical PV/E11 neurons, for example, "The manuscript is well written and comprised of a series of interesting and logical experiments. The data analysis is well performed and robust throughout. (Reviewer #2)", and "Overall, the experiments are carefully designed and executed, and the results are interesting and generally supportive of their conclusions. (Reviewer #3)".

We also acknowledge the reviewers for their fair and constructive concerns especially about the PV-neuron specificity (Reviewers #1 and #3), needs for the in-depth EEG/EMG analysis (Reviewers #1 and #3), and qualification of imaging data (Reviewer #2). We took these concerns seriously and substantially revised the manuscript by incorporating new experiments and data analysis including 1) analysis of the slow-wave activity (SWA) after sleep deprivation (SD) using E11 enhancer and PV-Cre dependent expression of CaMKII inhibitory peptide CN19o, and 2) evaluation of whole-brain imaging data to verify the quality of co-localization analysis. These data further support the conclusion that CaMKII activity in the PV/E11 neurons is involved in the rebound response in terms of both sleep duration and depth.

REVIEWER COMMENTS

Reviewer #1 (Remarks to the Author):

Kon and colleagues present a large set of experiments from which they conclude that the activity of parvalbumin inhibitory neurons in cortex plays an important role in the homeostatic regulation of sleep, during development and in adulthood. The paper includes many experiments in young and adult animals. Unfortunately, in most experiments the tools used to support the main claim, and the experimental design, make the interpretation of the results difficult at best. Many experiments are very difficult to interpret, and their number cannot compensate for lack of rigorous tools and design to address the question of interest. In general, these results are not put into the larger context of the many studies that have already examined the homeostatic regulation of sleep during early sensitive periods, and the general role of PV cells in sleep. Sleep-dependent effects on the balance excitation/inhibition in the cortex, including in PV cells, are well established, but are not cited and discussed here.

Calcium imaging and EGG/LFP/unit recordings show that PV cells are more active in sleep, especially REM sleep, and their chemogenetic block decreases SWA activity. This evidence is either not cited, or cited but not discussed.

The first part of the study focuses on the developmental period between early adolescence to adulthood. The authors describe changes in some sleep/wake patterns and response to sleep deprivation, and they ascribe these changes to changes in PV cells. However, two major technical issues discussed in detail below (the use of Snappy Sleep Stager to distinguish sleep and wake stages; the lack of any method to assess the efficacy of the sleep deprivation procedure) seriously limit the significance of the results. Also worrisome is the fact that the authors assume that a change in PV cell number will lead to changes in the inhibitory control of PV over pyramidal neurons, without using any functional measure of inhibition. In other words, it is not obvious why fewer PV cells would lead to a change in inhibition in cortex. The first part of the study is weak, unless the experiments are repeated after addressing critical technical and functional issues. The second part of the paper focuses on SD experiments in adults and suffers from the same technical and design limitations, including the apparent choice of focusing on a late phase of the rebound after SD, and the lack of EEG-based evidence, as discussed below.

We thank the reviewer for the careful evaluation of our methodologies and pointing out the limitation of this study. We conducted new experiments and analyses to address remaining concerns. We are also thankful for this reviewer for valuable comments about the previous studies. We revised the Introduction and Discussion sections to clearly address the possible relationship between our finding and previous studies about E-I balance and Ca²⁺ imaging of PV neurons. We are confident that the revised manuscript is now more solid and easy to follow.

line 90: Furthermore, cortical PV-neuron activity is associated with sleep-wake states. Ca²⁺ activity in PV neurons is increased during both rapid eye movement (REM) and non-REM (NREM) sleep in the somatosensory cortex, whereas the activity is constantly higher in REM sleep and lower but incremental during NREM sleep in the motor cortex, compared to wakefulness. Chemogenetic activation of PV neurons in the secondary motor cortex leads an increase in NREM sleep duration, but a decrease in the SWA. Nonetheless, the role of cortical PV neurons in the homeostatic rebound in sleep duration and the SWA is poorly understood.

line 528: Indeed, sleep-wake states influence neural activity of cortical PV neurons and a balance between synaptic excitation and inhibition (E-I balance) in cortical microcircuits.

Other point-by-point revisions are explained below:

MAJOR POINTS:

(1)

The Snappy Sleep Stager is unconventional and needs to be described and validated here. As stated in the cited paper, this system cannot distinguish NREM sleep from REM sleep. Obviously, it also cannot measure “depth” of sleep (SWA). In the current study, which claims to analyze the development of sleep homeostasis, it is puzzling that the authors rely almost exclusively on SSS for their experiments. Many previous studies have found that in developing rodents, it is not the total duration of sleep but NREM vs REM amounts that change. Moreover, previous studies have found that sleep rebound is mainly/only in terms of sleep duration in the least mature animal (~P20-25), while later it mainly consists of changes in SWA. Given the complexity of these very dynamical changes, any claim about developmental changes in sleep homeostasis need to be supported by EEG/LFP sleep/wake recordings. The readout of “sleep fragmentation” using the probability w_s or w_w only based on SSS, which is used for all experiments (e.g. silencing of E11 cells) is not enough.

We thank the reviewer for the comments. As the reviewer mentioned, it is important to assess sleep homeostasis in terms of sleep duration and sleep depth (i.e., SWA). We added new data of EEG/EMG recordings under sleep deprivation (SD) in key parts of this study, confirming that CaMKII activity in the cortical PV/E11-neurons is important for the homeostatic rebound in the SWA after SD (**Figs. 6a–h**, please refer to our response to **Comments (2), (4), and (6)** below). Furthermore, in the revised manuscript, we modified our terminology of “homeostasis” and “rebound” to specify whether sleep duration (mainly measured by SSS) or the SWA (measured by EEG/EMG) was described.

line 132: Consistent with previous reports, the longer sleep after SD mainly emerged in the dark period (zeitgeber time [ZT] 12–24); thus, we calculated the sleep parameters during the dark period to assess rebound responses. The homeostatic rebound in sleep duration was clearly observed at 8 and 12 weeks old, but less so at 4 weeks old (**Figs. 1g–i and Supplementary Fig. 2a**), indicating that the homeostatic regulatory system in sleep amount is immature at 4 weeks old.

We also agree that EEG/EMG recording is a conventional method to assess sleep/wake states in mammals. We also want to point out that it is still difficult, albeit may not impossible, to conduct the long-term EEG/EMG recording (over 2 months) due to its invasiveness and mechanical vulnerability. Longitudinal analysis offers a more accurate characterization of sleep development compared to cross-sectional analysis (Campbell and Feinberg, 2009; Frank *et al.*, 2017; Rensing *et al.*, 2018). Our longitudinal sleep recording, including SD experiments, could be performed in individual developing mice because the SSS system is less invasive for mice (**Figs. 1a,f**). We believe that our longitudinal sleep analysis provides valuable insights into the developmental changes in sleep architecture and sleep homeostasis, complementing other cross-sectional sleep analyses by EEG/EMG recording (e.g., Nelson *et al.*, 2013). We also want to emphasize the reliability and accuracy of the SSS system for sleep/wake staging. In the revised manuscript, we explicitly added the statistical evaluation of SSS method compared with the conventional EEG/EMG recording.

line 119: The SSS system demonstrates high accuracy ($95.3 \pm 0.4\%$) when sleep/wake staging based on EEG/electromyography (EMG) recordings is used as a reference.

We also agree with the reviewer's comment for the limitation of the SSS system and amended our manuscript to indicate that longitudinal evaluation of the SWA based on EEG/EMG recordings during the development is future perspectives and clearly described the rationale/limitation to use P_{WS} and P_{SW} as a surrogate of the SWA dynamics.

line 543: Long-term EEG/EMG recording to assess the SWA remains a future challenge, as it could not be validated in this longitudinal sleep recording. However, it is noteworthy that the SWA is negatively correlated with the number of micro-arousals, suggesting that the decreased P_{SW} could serve as a surrogate for the increased SWA. Consistent with the previous report that a significant increase in SWA after SD is observed only in mice older than 6 weeks old, a significant decrease in P_{SW} after SD was observed at 8 and 12 weeks old, but not at 4 weeks old (**Figs. 1k,p**).

(2)

The second major issue is that sleep deprivation was done in animals that were not recorded (no SSS, no EEG), thus we do not know whether the procedure was equally effective in very young or more mature animals. It is very possible, in fact likely, that this was not the case. This evidence needs to be provided, even more so because the method used 30sec stim, 50

sec off; this method mainly fragments sleep, often without actually changing the total time spent asleep. In short, the smaller sleep rebound at 4 weeks may be due to less effective sleep deprivation at that age. This possibility needs to be formally tested.

We thank the reviewer for pointing out this issue. The primary rationale behind employing an orbital shaker for SD was to prevent any potential enduring effects of acute stress during the developmental period on subsequent sleep. This approach has been demonstrated to result in a lower elevation of blood corticosterone levels compared to other SD methods, thereby minimizing the induction of acute stress (Bian *et al.*, 2022; Zamore and Veasey, 2022). Furthermore, although, as pointed out by the reviewer, this method is more commonly associated with chronic sleep fragmentation studies, it has shown effectiveness in reducing sleep duration even in milder condition, particularly during the initial period of its application (Sinton, Kovakkattu and Friese, 2009; Bian *et al.*, 2022). It was observed that intermittent shaking appeared to enhance the SD effects compared to continuous shaking based on our preliminary experiments; hence, we chose this intermittent approach.

However, we acknowledge the reviewer's concern that our SD method may not have had equal effectiveness across different ages. To address this concern, we conducted MK-801 administration experiments, enabling us to measure the actual sleep loss induced by the MK-801 administration (**Fig. 1I**). We confirmed a weaker rebound response in sleep duration at 4 weeks old, even though sleep duration decreased after MK-801 administration at all ages. We included this rationale in the revised manuscript for better readability.

line 143: This approach allowed us to measure the actual sleep loss induced by the MK-801 administration (**Fig. 1I**). The homeostatic rebound in sleep duration was clearer at 8 and 12 weeks old than at 4 weeks old after MK-801 administration (**Figs. 1I–n** and **Supplementary Fig. 2b**), as observed in the SD. Similar to SD, increased P_{ws} and decreased P_{sw} were observed during the rebound phase at 8 and 12 weeks old (**Figs. 1o,p**). Together, long-term sleep phenotyping in developing mice revealed that the daily sleep-wake pattern and its homeostatic regulatory system mature with post-weaning development in mice.

Furthermore, we want to emphasize that we now include key additional SD experiments using EEG/EMG recordings. In these experiments, we calculated the sleep-deprived rate during the SD session to ensure its effectiveness and found that approximately 95% of sleep was effectively deprived. We have included this information in the Figure legends of the respective experiments (**Figs. 6b,f**). Please also refer to our response for **Comments (4)** and **(6)** below.

line 1339: Hourly normalized delta power (0.5–4 Hz) during NREM sleep in E11-CN19scr ($n = 22$) and E11-CN19o mice ($n = 15$) in SD experiments. Mice were allowed to behave freely for 1 day prior to SD, and the value is shown as “Basal.” SD was effectively performed during ZT0–6 with a high success rate ($93.4 \pm 5.3\%$).

line 1350: Hourly normalized delta power (0.5–4 Hz) during NREM sleep in *Pvalb-2A-Cre:E11-DIO-CN19scr* ($n = 16$) and *Pvalb-2A-Cre:E11-DIO-CN19o* mice ($n = 13$) in SD experiments. Mice were allowed to behave freely for 1 day prior to SD, and the value is shown as “Basal.” SD was effectively performed during ZT0–6 with a high success rate ($95.0 \pm 6.2\%$).

(3)

The rationale to link PV cells density to changes in sleep homeostasis is weak at best. The underlying assumption is that the extent to which pyramidal neurons are under inhibitory control is changing, but cell number is not a proxy for the efficacy of inhibition. Thus, I do not find this experiment to be informative.

We appreciate the reviewer's comments. We acknowledge that the change of PV+ cell density may not be the direct evidence of their involvement in developmental sleep changes. In the revised manuscript, we rephrased the sentence to maintain a more neutral tone for better readability.

line 181: In summary, the decline of PV+ cell density was observed across the entire brain during development when the daily sleep-wake pattern and its homeostatic regulatory system matured.

Although the PV+ cell number may not directly reflect their inhibitory function, previous studies indicated that a lower number of PV neurons is associated with PV-neuron hypoactivity and reduced inhibition onto excitatory neurons in animal models of autism spectrum disorder (Filice *et al.*, 2020; Contractor, Ethell and Portera-Cailliau, 2021). Our results demonstrated that sleep stability was decreased by inhibiting PV-neuron activity, resembling the condition as a lower PV+ cell density (i.e., in the late development period). Future research is needed to explore the developmental changes in physiological PV-neuron activity and their inhibitory functions within the cortical circuits. We revised the manuscript to include this limitation and interpretation as follows.

line 186: Although the PV+ cell density may not directly reflect their inhibitory function, previous studies indicated that a lower number of PV neurons is associated with PV-neuron hypoactivity and reduced inhibition onto excitatory neurons in ASD models, whose sleep homeostasis is impaired. Thus, we next investigated the potential link between PV-neuron activity and homeostatic regulation of sleep.

(4)

The authors use the AAV-mediated approach via E11 to target PV neurons. They should clearly state why they did not use cre-dependent mouse lines in which cortical PV cells are labeled. Wouldn't this approach target more cells than the AAV-mediated approach? Please clarify the rationale. The effects of the silencing of PV cells via TeLI experiments are measured only via the SSS method (ws and sw transitions) and therefore are very difficult to interpret for the reasons explained above. Moreover, I may have missed it, but I did not see any direct evidence that synaptic transmission is actually blocked in PV cells using this AAV approach.

We would like to express our gratitude to the reviewer for their careful evaluation of our AAV-mediated approach for targeting cortical PV neurons. Our primary rationale for employing this approach was to selectively label cortical PV neurons, which exhibit the most pronounced response to increased sleep need (**Figs. 3a–h**), rather than targeting PV neurons throughout the entire brain. We acknowledge the future challenge of generating a Cre mouse line which could selectively target cortical PV neurons (rather than entire PV neurons). The E11 enhancer was recently identified as a valuable tool for the preferential targeting of cortical PV neurons (Vormstein-Schneider *et al.*, 2020). We also confirmed the preferential targeting of cortical PV neurons by the E11 enhancer (**Figs. 3j–l**).

We were also aware of the constraints associated with the AAV-mediated targeting approach and the requirement to confirm the involvement of cortical PV neurons and their intracellular CaMKII in sleep homeostatic regulation through alternative methods. In the revised manuscript, we further employed the E22 enhancer for chemogenetic inhibition/activation of cortical PV neurons (**Supplementary Figs. 6a–f**), as it is known to preferentially target cortical PV neurons (Vormstein-Schneider *et al.*, 2020). We confirmed that chemogenetic manipulation using both E11 and E22 enhancers resulted in similar effects.

line 247: Similar effect was observed in chemogenetic manipulation using another PV-neuron selective enhancer E22, which predominantly confined gene expression to cortical PV neurons (**Supplementary Figs. 6a–f**).

Furthermore, following to the reviewer's comment, we employed a confinement strategy involving PV-Cre mice and the E11 enhancer in CaMKII inhibition/activation within cortical PV neurons (**Figs. 6e–h, Figs. 7n–q**). These additional findings have been integrated into the revised manuscript to provide robust evidence of the requirement of cortical PV neurons and their intracellular CaMKII activation for homeostatic sleep rebound.

line 333: To express CN19o in cortical PV neurons more specifically, we employed a confinement strategy using the E11 enhancer and *Pvalb-2A-Cre* mice (**Fig. 6e**). Cre-dependent expression of E11-CN19o resulted in a reduction in REM sleep duration, while it had no significant impact on total sleep or NREM sleep duration (**Supplementary Figs. 9i–k**). In contrast, a significant reduction in delta power during NREM sleep was observed (**Supplementary Figs. 9l–n**). Consistent with the results observed with the direct CN19o expression under the E11 enhancer (**Figs. 6b–d**), the Cre-dependent E11-CN19o expression markedly attenuated the homeostatic rebound in delta power following SD (**Figs. 6f–h**). Taken together, these findings demonstrate that CaMKII inhibition in cortical PV neurons leads to impairment of sleep homeostasis.

line 400: To confirm the involvement of cortical PV neurons, we employed the confinement strategy using the E11 enhancer and *Pvalb-2A-Cre* mice for specific CaMKII α expression in cortical PV neurons (**Figs. 7n**). Similar to the direct CaMKII α expression under the E11 enhancer, the rebound-sleep-like state was induced by Cre-dependent CaMKII α (T286D) expression under the E11 enhancer (**Figs. 7o–q**). Sleep-promoting effects were also observed with Cre-dependent CaMKII α (T286D) expression under pan-neural *hSyn* promoter (**Supplementary Figs. 11d–g**). These findings further support the hypothesis that CaMKII activation in cortical PV neurons induces rebound sleep.

We acknowledge the importance of tool validation for tetanus-toxin light-chain (TeLC) experiment, although we did not conduct specific experiments to examine the efficiency of TeLC expression in blocking synaptic transmission. It is important to note that expression of TeLC protein, including AAV-mediated expression, has been a well-established method to block synaptic transmission and has been used in numerous studies including our and other researchers' groups (e.g., Niwa *et al.*, 2018; Takatoh *et al.*, 2022; Jung *et al.*, 2023).

Therefore, while tool validation is crucial, we believe that it might not be essential to include these specific experiments in this study, where we observe the phenotypical changes in TelLC-expressed mice (**Supplementary Figs. 5c–g**).

(5)

The design of the chemogenetic experiment (E11-M3 or M4 mice) is difficult to follow: as shown in figure 4, why is CNO given only 6 hours after the end of SD? Extensive previous evidence (based on EEG recordings) showed that most of the rebound in terms of duration and depth of sleep has dissipated after 6 hours. To make the case that PV neurons can interfere with the rebound, they should be excited/inhibited while the rebound is occurring, i.e. immediately after SD. Similar issue for figure 3: why are PV cells only chemogenetically activated or inhibited in the dark, while mice mainly sleep during the day?

We appreciate the reviewer's insightful comments. The experiment involving the chemogenetic suppression of E11 neurons using a lower CNO dosage after SD aims to determine the necessity of E11 neurons for the homeostatic rebound in sleep duration. As mentioned earlier, while the homeostatic rebound in the SWA is immediately induced after SD, the rebound in sleep duration is predominantly shown during the dark period of recovery phase (Franken, Malafosse and Tafti, 1999), consistent with our findings (**Figs. 1g–p**). To align with this temporal pattern, CNO was administered at ZT12, just before the expected rebound phase (**Figs. 4a**). Indeed, significant rebound in sleep duration was observed during the dark period under control conditions (**Figs. 4b,c**). A similar experimental design was applied to the chemogenetic activation of E11 neurons, enabling a comparison with characteristics of the physiological rebound. We elaborated on the rationale for the timing of CNO administration in the revised manuscript.

line 255: To test this, we used chemogenetic inhibition of E11 neurons from the onset of dark period in rebound phase because an increase of sleep duration after SD is mainly shown during dark period (**Fig. 4a**).

(6)

A similar issue applies to the experiments to block CamKII activity in PV cells. What is the evidence that the activity is indeed blocked? Is this effect assumed to be continuous? Again, the authors find an effect on the rebound after SD, but they rely only on the SSS method. When the authors finally measure EEG in these mice, they find signs of sleep fragmentation, but apparently do not repeat the SD experiment to support their claim that sleep rebound is

reduced after blocking CamKII activity. It is difficult to understand why the SD experiment was not done. Moreover, Figure 6b shows that for the first 6 hours after SD the recovery of sleep occurs to the same extent. How do the authors explain this?

We thank the reviewer for the comments. In the case of control mice (i.e., E11-CN19scr mice), we observed the marked rebound in sleep duration during the dark period rather than the light period (**Figs. 5b–k**), in line with previous reports and our own findings. We believe this observation provides a reasonable explanation for the similar rebound during the light period between the E11-CN19scr and E11-CN19o mice. We revised our manuscript to emphasize that the paragraph focuses on homeostatic regulation in terms of sleep duration or amount.

line 297: Following that, we performed a 6-hour SD on E11-CN19o and E11-CN19scr mice to investigate the effects of CaMKII kinase activity in E11 neurons on the homeostatic regulation of sleep amount (**Fig. 5a**). Surprisingly, the E11-CN19o mice barely showed the homeostatic rebound in sleep duration, whereas the E11-CN19scr mice clearly showed rebound responses following SD (**Figs. 5b–d** and **Supplementary Fig. 9d**).

We also acknowledge the importance of evaluating sleep homeostasis in terms of the SWA and determining whether CaMKII activity in E11 neurons is necessary for the homeostatic rebound in the SWA. To address this question, we included new data from EEG/EMG recordings under SD with CaMKII inhibition in E11 neurons (**Figs. 6a–h**). Indeed, the results indicate that rebound increase of NREM delta power after SD is attenuated in the mice expressing CaMKII inhibitor in E11 neurons. Note that we also showed similar results using the combination of PV-Cre mouse and E11 enhancer, as mentioned in our response to **Comment (4)** above.

line 322: We next investigated whether CaMKII in cortical PV neurons is involved in the homeostatic regulation of the SWA as well as sleep duration. We performed a 6-hour SD on E11-CN19o and E11-CN19scr mice under EEG/EMG recording (**Fig. 6a**). The delta power during NREM sleep was significantly increased after SD and returned to baseline over 6 hours in the E11-CN19scr mice (**Figs. 6b,c**). In contrast, the E11-CN19o mice presented no remarkable increase in delta power after SD, suggesting that the homeostatic rebound in delta power after SD was attenuated in the E11-CN19o mice (**Figs. 6b,c**). Accordingly, the delta power rebound was significantly lower in the E11-CN19o mice compared to the E11-CN19scr mice (**Fig. 6d**). These results indicate that CaMKII kinase activity in E11 neurons is

crucial not only for homeostatic rebound of sleep amount but also for homeostatic rebound of the SWA.

line 333: To express CN19o in cortical PV neurons more specifically, we employed a confinement strategy using the E11 enhancer and *Pvalb-2A-Cre* mice (**Fig. 6e**). Cre-dependent expression of E11-CN19o resulted in a reduction in REM sleep duration, while it had no significant impact on total sleep or NREM sleep duration (**Supplementary Figs. 9i–k**). In contrast, a significant reduction in delta power during NREM sleep was observed (**Supplementary Figs. 9l–n**). Consistent with the results observed with the direct CN19o expression under the E11 enhancer (**Figs. 6b–d**), the Cre-dependent E11-CN19o expression markedly attenuated the homeostatic rebound in delta power following SD (**Figs. 6f–h**). Taken together, these findings demonstrate that CaMKII inhibition in cortical PV neurons leads to impairment of sleep homeostasis.

Regarding the potency of CN19o as a CaMKII inhibitor, it is important to note that CN19o has been established as a highly selective and potent peptide inhibitor in comparison to other conventional CaMKII inhibitors including AIP2 (Ishida *et al.*, 1998; Coultrap and Bayer, 2011), which has been used for *in vivo* CaMKII inhibition both within our and other researchers' groups (e.g., Murakoshi *et al.*, 2017; Iino *et al.*, 2020; Tone *et al.*, 2022). Additionally, in our experiments, we included a control condition that utilized the scrambled sequence of CN19o (CN19scr). Thus, we believe that it is most plausible to interpret the results as indicative of the effects of CaMKII inhibition by CN19o.

Similar major issues apply to the experiment with presumed increase in CamKII activity (not shown). First, the first claim about longer sleep in T286D mice during baseline is validated with EEG recordings, but the authors again do not study the response to sleep deprivation with EEG recording, which is puzzling. The assumption is that increased CamKII activity makes PV cells more active, resulting in a “sleep rebound” state even without sleep deprivation. However, a previous study showed that chemogenetic PV cell activation increases NREM duration but at the same time decreases delta activity, which is not consistent with a physiological sleep rebound.

We appreciate the reviewer's comments. As the reviewer noted, a previous study indicated that chemogenetic activation of PV neurons in the secondary motor cortex (M2) induces unphysiological NREM sleep. In other words, behaviorally/polygraphically-defined NREM sleep duration was increased, whereas the SWA in M2 region was decreased by the

chemogenetic activation (Funk *et al.*, 2017). In contrast to this previous report, our results showed that CaMKII activation in PV/E11 neurons induces an increase in both sleep duration and the SWA (**Figs. 7b–q**).

It is important to note that these different phenotypes could be attributed to differences in the target cells. Cortical PV neurons are a heterogeneous population with distinct transcriptomic, morphological, and electrophysiological characteristics (Gouwens *et al.*, 2020). Given that the sensitivity of E11-mediated targeting for PV+ cells is approximately 50% (**Supplementary Fig. 5b**), it is possible that E11 is targeting a specific subpopulation within PV neurons. Investigating whether E11 targets a specific cell type and, if so, which cell type, is a task for future research to better understand the involvement of cortical PV neurons in sleep homeostatic regulation. In the revised manuscript, we added a comparison with previous reports and a hypothesis to explain the different phenotypes in the Discussion section.

line 512: In contrast to a previous finding in which activating cortical PV neurons increased NREM sleep duration but decreased the SWA, elevating E11-neuron activity through CaMKII activation increases both NREM sleep duration and SWA. These differences might be attributed to the heterogeneous nature of cortical PV neurons and the E11 enhancer's potential to selectively target specific subpopulation(s) within these neurons.

We appreciate the reviewer's concern about the lack of direct measurement of CaMKII kinase activity in PV neurons. However, we did not observe significant changes in any sleep parameters in the kinase-dead version of T286D (i.e., K42R:T286D) as well as other control conditions (**Supplementary Figs. 10a–c**); thus, it is plausible that at least the constitutive kinase activity of expressed CaMKII α is required to induce a rebound-sleep-like state. Furthermore, we demonstrated that CaMKII inhibition in PV neurons attenuates rebound responses both in sleep duration and the SWA after SD (**Figs. 5b–k, Figs. 6a–h**). We believe that these results collectively well support our conclusion that the CaMKII kinase activity in PV neurons plays a crucial role in homeostatic sleep rebound.

(7)

The Fos experiment in figure 7 finds that the presumed increase in CamKII activity leads to more PV cells labeled with Fos. Is this Fos expression present both after sleep and after waking?

We thank the reviewer for the insightful comments. The brains of E11-CaMKII α (WT) and E11-CaMKII α (T286D) mice were collected at approximately ZT6, a time when the E11-CaMKII α (T286D) mice predominantly exhibited sleep characterized by higher SWA. On the other hand, in the SD or MK-801 administration experiments, we collected the mouse brains after prolonged wakefulness (immediately following SD or 2 hours after MK-801 injection). Collectively, these findings suggest that c-Fos expression in cortical PV neurons is likely to increase when animals have high sleep need, regardless of whether the preceding state was sleep or wakefulness. We revised the manuscript to include this interpretation in the Results section and the timing of brain collection in the Methods section.

line 422: Given that prolonged wakefulness also increases PV-neuron activity (**Figs. 3d–h**), higher sleep needs appear to increase the activity, regardless of whether the preceding state was sleep or wakefulness.

line 814: Following sleep phenotype, the brains of mice were collected at ZT6.

(8)

The rationale for focusing on CamKII in PV is not straightforward and should be clarified. The cited studies focus on CamKII on glutamatergic synapses. In the final experiment (Fig. 7g), T286 phospho levels are measured after SD relative to control in the whole brain. While suggestive, this experiment is not direct proof that CamKII activity is increased, as the authors acknowledge.

We appreciate the reviewer's comments. As the reviewer noted, previous studies indicated associations between the phosphorylation states of CaMKII in synaptosomes or post-synaptic density (excitatory/glutamatergic synapse fractions) and sleep need (Diering *et al.*, 2017; Wang *et al.*, 2018; Brüning *et al.*, 2019). We concentrated on the CaMKII action in cortical PV neurons because these neurons possess excitatory synapses as well as excitatory neurons (Sancho and Bloodgood, 2018; Melander *et al.*, 2021). Of note, our present study does not negate the potential involvement of CaMKII in excitatory neurons in sleep homeostatic regulation. Understanding the cell type-specific function of CaMKII in sleep homeostatic regulation is an important future challenge. Thus, we revised our manuscript to include potential research directions in the Discussion section as follows.

line 558: Investigating the developmental changes in protein expression of CaMKII isoforms within PV neurons and their specific functions in sleep homeostatic regulation remains crucial tasks for future research.

In the LC-MS/MS analysis (**Fig. 8g**; Fig. 7g in the initial submission), we specifically collected the cortical regions, rather than the entire brain, immediately after SD. Furthermore, our quantification in this analysis is specific to CaMKII α expressed under the E11 enhancer, as the watermark mutation (E285D) allows us to distinguish E11-CaMKII α from all endogenous CaMKII isoforms (please refer to **Fig. 8f**). Therefore, our LC-MS/MS analysis measures the T286 phosphorylation (pT286) level of CaMKII α in the cortical PV neurons targeted by the E11 enhancer. We revised our manuscript to clearly describe that cortical regions were selectively collected, and the LC-MS/MS analysis targeted the pT286 level of E11-CaMKII α (E285D). Additionally, we revised the conclusion sentence to include the limitation of pT286 quantification as a surrogate for the direct measurement of kinase activity.

line 452: We gave the E11-CaMKII α (E285D) mice a 6-hour SD and collected their cortical regions of brain immediately afterward (**Fig. 8f**). T286 phosphorylation (pT286) level derived from E11-CaMKII α (E285D) increased in SD group compared to *ad libitum* sleep (S) group (**Fig. 8g**). Although pT286 level serves as a surrogate for CaMKII α kinase activity, this result suggests that CaMKII kinase activity in cortical PV neurons increases upon the accumulation of sleep need.

MINOR POINTS:

(1)

lines 60-62: please fix the sentence

We appreciate the reviewer's careful review. We amended the grammatical error.

line 67: However, the “flip-flop” switch between wake- and sleep-promoting neurons, resulting in an overall positive feedback system, is less suitable for homeostatic regulation.

(2)

line 78: may be responsible

Again, we appreciate the reviewer's careful review. We added “be”.

line 84: Cortical PV neurodevelopmental dysfunctions may be responsible for the pathogenesis of autism spectrum disorder (ASD), and abnormal sleep symptoms are observed in many ASD patients and animal models of ASD.

(3)

line 92: what is "sleep stabilization"? please clarify

We thank the reviewer for pointing out our inconsistent terminology. In the revised manuscript, we maintained consistency by using the same wording (sleep stability) as in other parts of the manuscript.

line 102: The chemogenetic manipulation of cortical PV-neuron activity revealed that the activity is important for sleep stability.

Reviewer #2 (Remarks to the Author):

The manuscript Cortical parvalbumin neurons are responsible for homeostatic sleep rebound through CaMKII activation by Kon and colleagues examines the role of parvalbumin (PV) expressing neurons in the maintenance of homeostatic regulation of sleep rebound. The authors propose as a model that CaMKII kinase modulates the activity of cortical PV neurons to regulate sleep rebound.

Overall, I found this manuscript enjoyable to read with an intriguing narrative. The manuscript is well written and comprised of a series of interesting and logical experiments. The data analysis is well performed and robust throughout. Despite this overall positive impression, I have some concerns, outlined below, about the reliability of some of the experiments.

We are grateful to the reviewer for constructive and insightful comments. We are also pleased to hear that the reviewer found our manuscript well-written and the series of experiments, along with the data analysis, are robust and logically structured. We conducted new experiments and analyses, enhancing the reliability and solidity of our revised manuscript.

Point-by-point revisions are explained below:

MAJOR POINTS:

(1)

My major concern with the manuscript relates to the whole brain immunolabeling/lightsheet imaging. Although this is a flashy technique and potentially one that might aid with high throughput analysis, it is one that clearly compromises the reliability of the data output. The imaging criteria are not well documented in the methods (this should be improved/expanded) but the authors state the use of a 0.63X objective lens. This magnification, especially combined with the inherent shortcomings of tissue clearing and lightsheet microscopy, would be inappropriate for co-localization analysis.

We thank the reviewer for pointing out this issue. We revised our manuscript to include the detailed imaging criteria and cell detection/co-localization methods. First, to improve the clarity and reliability of our manuscript, we provided a more detailed description of the light-sheet imaging conditions and equipment in the Methods section as follows.

line 768: The microscope was equipped with diode or DPSS lasers with wavelengths of 488, 532, 594, and 642 nm (Coherent). To ensure homogeneous resolution in XYZ axes, the microscope employed an axially-swept light sheet mechanism. The microscope was fitted with a 0.63X macro-zoom objective lens (Olympus) and variable 0.63-6.3X intermediate zoom optics (Olympus). The fluorescence signal was efficiently captured using an sCMOS camera (PCO), paired with appropriate fluorescence filters. In all conducted experiments, the voxel size was (X, Y, Z) = (6.45, 6.45, 7.0) μm with 1.6X intermediate zoom optics, providing total 1.0X magnification. An effective resolution should consider the moderate tissue expansion (1.5X) induced by the CUBIC-R+ treatment.

In this condition, we previously validated the accuracy of cell detection for both PV+ and c-Fos+ cells (Mano *et al.*, 2021). Our validation indicated that the precision exceeded 80% in most brain regions, except for cerebellar regions—because of this limitation, we excluded cerebellar regions for our co-localization analysis. Importantly, precision in the cortical regions exceeded 90%, which is likely attributed to the relatively sparse distribution of cells within these areas. We included this accuracy about the cell detection in the Methods section. We also included the quantitative information regarding on the co-localization analysis. As intercellular distances in the unexpanded mouse brain were ranging from 16 to 20 μm in almost all brain regions (Murakami *et al.*, 2018), we set the threshold of intercellular distances at 16 μm to consider the co-localization of c-Fos+ PV+ signals. This threshold is larger than the voxel size and furthermore, due to tissue expansion by a factor of ~ 1.5 because of CUBIC-R+ treatment (Tainaka *et al.*, 2018), this distance equated to approximately 10.5 μm in untreated tissue. We believe our method is adequately reliable for co-localization analysis.

line 826: The c-Fos+PV+ cells were identified by a search for cells characterized by the presence of both c-Fos and PV signals (**Supplementary Fig. 4a**). We previously confirmed that the accuracy of the cell detection is more than 80% precision for both PV+ and c-Fos+ cells in most of brain regions. The accuracy is relatively lower in the cerebellar regions so that we excluded the regions from the analysis. We performed cell identification as previously reported. Cell counting was performed independently for each channel, and the center of mass of the detected cells was determined. For each c-Fos+ cells, if a PV+ cell was found within a distance of 16 μm , it was considered a c-Fos+PV+ cell. Note that due to tissue expansion by a factor of ~ 1.5 , this distance equated to approximately 10.5 μm in untreated tissue. This procedure was applied for the identification of PV+mCherry+ or c-Fos+mCherry+ cells.

(2)

Related to the above; in extended data figure 4, there are data points suggesting 100% of c-fos+ cells are PV+. i.e. No neurons other than PV cells are active. This alone seems nearly if not completely impossible. Coupled with other data points in the same group showing 0% of c-fos+ cells are PV+ demonstrates that there is something highly unreliable about the co-localization data. My suspicion is that this relates to the tissue clearing and low magnification imaging as described above. Adequate demonstration of fos/pv co-localization will require the authors to perform some standard tissue sectioning/staining and high magnification high resolution imaging.

We appreciate the reviewer's thorough review. As the reviewer noted, there were data points indicating that 100% of c-Fos+ cells were PV+ cells in certain subregions in the isocortex. Upon a careful re-evaluation, we realized that this outcome was due to the extremely low number of c-Fos+ cells, likely attributed to the mice being kept in constant dark conditions to minimize background noise. Recognizing that a ratio analysis based on such limited c-Fos+ cell numbers may not provide a sufficiently reliable assessment, we decided to remove the figures related to the quantification of cortical subregions in the revised manuscript. Our primary findings are based on the quantification of the entire isocortex, and thus, this removal does not impact the overall conclusion. We deeply thank the reviewer for catching this issue.

line 201: The same effect was seen with MK-801 administration; cortical c-Fos+PV+ cells were found to be increased in MK-801-treated brains (**Figs. 3g,h and Extended Data Figs. 4e-g**).

(3)

Also related to the above concerns, the authors don't include any real imaging to show colocalization and confirm that these techniques are capable of reliably giving colocalization data and that the labeling was specific.

We appreciate the reviewer's constructive comments. We now included representative images to illustrate the effectiveness of our methods for detecting co-localized cells (**Supplementary Figs. 4a**). We confirmed that our method is capable to specifically detect co-localized cells.

line 826: The c-Fos+PV+ cells were identified by a search for cells characterized by the presence of both c-Fos and PV signals (**Supplementary Fig. 4a**).

(4)

Validation of the ilastik cell counts. Can the authors confirm the accuracy of the cell counts with this method? How reliable is it in each of the brain regions the authors are examining? This must be confirmed to be reliable. It seems likely that superficial imaging would be more reliable than deeper imaging would be.

We thank the reviewer for the valuable comments. As mentioned in our response to **Comment (1)** above, we previously validated the accuracy of cell detection using ilastik and confirmed its reliability (Mano *et al.*, 2021). Based on our previous analysis, it seems that the accuracy is more related to the cell density in the region rather than the depth of imaging. We explicitly stated in the revised manuscript that we excluded the cerebellar regions from the co-localization analysis due to their relatively lower accuracy (please also refer to our response to **Comment (1)** above). We believe that this clarification enhances the readability and transparency of our methods.

line 828: We previously confirmed that the accuracy of the cell detection is more than 80% precision for both PV+ and c-Fos+ cells in most of brain regions. The accuracy is relatively lower in the cerebellar regions so that we excluded the regions from the analysis.

(5)

The normalized c-fos counts (SD/SLP groups) included in figure 3 are fine as a data reduction for analysis but the non-normalized data should also be included in the extended data set for transparency.

We appreciate the valuable suggestion from the reviewer. To enhance transparency, we included the non-normalized data indicating the cell density in the revised manuscript (**Supplementary Figs. 4b–d**). Non-normalized data further supports our primary findings from the normalized data.

line 192: A comparison of control (*ad libitum* sleep: S) and sleep-deprived (SD) brains revealed that SD increased c-Fos-positive (c-Fos+) cell density in many brain regions, consistent with previous reports (**Fig. 3b** and **Supplementary Figs. 4a,b**). We also discovered that the SD increased the density of PV+ cell in the isocortex to some extent (**Fig.**

3c and Supplementary Figs. 4a,c). Furthermore, there was an increase in c-Fos⁺ cell density among the PV⁺ cell population (i.e., c-Fos⁺PV⁺ cells), particularly in the isocortex (**Fig. 3d and Supplementary Figs. 4a,d**).

(6)

The authors show very high expression of CaMKII in PV+ interneurons using scRNA-seq. It's important to not here that this is mRNA. Can the authors demonstrate with immunolabeling what Percentage of these cells actually express CamKII protein?

We appreciate the reviewer's comments and apologize for any confusion. Our analysis indicated the presence of mRNA for CaMKII isoforms within cortical PV neurons but did not suggest exceptionally high expression levels. In fact, mRNA expression of CaMKII isoforms is generally higher in excitatory neurons than in inhibitory neurons, including PV neurons (Tasic *et al.*, 2018). The sentence was rephrased for better clarity in the revised manuscript.

line 280: All CaMKII isoforms (*Camk2a*, *Camk2b*, *Camk2g*, and *Camk2d*) were detected at the mRNA level in cortical PV neurons (**Supplementary Figs. 8a–d**).

We also appreciate the reviewer's comment and agree on the importance of examining CaMKII expression at the protein level through immunostaining. However, we think that it would be difficult to evaluate the percentage of PV cells expressing CaMKII, especially CaMKII α , by immunostaining approaches. This is partly because excitatory neurons express the high level of CaMKII α compared with PV neurons. Thus, it has been traditionally assumed that PV neurons do not express CaMKII α because immunostained CaMKII α signals are barely detectable (Liu and Jones, 1996; Sık *et al.*, 1998). Recent study detected immunostained CaMKII α signals in a subpopulation of PV neurons through the in-depth combination of microRNA-guided cell targeting, single-cell RNA sequencing, and electrophysiological characterization (Keaveney *et al.*, 2020). Related to the protein level expression of CaMKII, recent study demonstrated that CaMKII γ is present in cortical and hippocampal PV neurons at the protein level, where it plays a significant role in synaptic regulation and memory (He *et al.*, 2021). We agree that investigating the protein expression of CaMKII isoforms within PV neurons and their specific role in sleep homeostatic regulation remains crucial tasks for future research and added these related studies in the Discussion section as follows.

line 553: While it was conventionally believed that cortical PV neurons lack CaMKII α expression at the protein level, recent study identified CaMKII α protein signals in a subpopulation of cortical PV neurons. Another study demonstrated that CaMKII γ in cortical and hippocampal PV neurons plays a crucial role in synaptic regulation and memory. Investigating the developmental changes in protein expression of CaMKII isoforms within PV neurons and their specific functions in sleep homeostatic regulation remains crucial tasks for future research.

(7)

Is c-fos a reliable marker for activity in experiments where CaMKII is manipulated? It has been previously demonstrated that inhibition of CaMKII impairs fos expression as it is part of the c-fos induction cascade (e.g. <https://doi.org/10.1074/jbc.M412680200>). Therefore, there may be changes in c-fos simply as a result of disrupting this pathway without having a direct effect of neuronal activity per se. A CaMKII independent activity marker would be more appropriate.

We appreciate the insightful comment from the reviewer. As pointed out by the reviewer, there is a concern regarding the potential nonspecific (i.e., independent of neuronal activity) elevation of c-Fos expression due to CaMKII activation. To address this concern, we conducted additional experiments to analyze c-Fos expression profile upon CaMKII activation in *Vglut2*-expressing excitatory neurons (*Vglut2* neurons). As expected by our previous study (Tone et al. 2022), CaMKII activation in *Vglut2* neurons increased sleep amount (**Supplementary Figs. 13c–e**). However, the CaMKII activation did not result in significant changes in c-Fos expression profile in cortical *Vglut2* neurons (**Supplementary Figs. 13f,g**), suggesting that the marked increase in c-Fos expression observed in PV neurons was, at the very least, not likely due to nonspecific and cell-type independent effects of intracellular CaMKII activation.

line 426: We next evaluated whether enhancement of c-Fos expression upon CaMKII activation is characteristic of PV neurons. We expressed CaMKII α and H2B-mCherry in excitatory neurons using *Vglut2*-ires-Cre mice to induce intracellular CaMKII activation in *Vglut2* neurons, not in PV neurons (**Supplementary Figs. 13a,b**). The sleep-promoting effects were observed with CaMKII α (T286D) expression in *Vglut2* neurons (**Supplementary Figs. 13c–e**), consistent with sleep induction by CaMKII β activation in this population. However, CaMKII α (T286D) expression in *Vglut2* neurons did not alter c-Fos expression profile of cortical *Vglut2* neurons (**Supplementary Figs. 13f,g**), suggesting that intracellular

CaMKII activation specifically enhances PV-neuron activity, rather than excitatory neuron activity, despite their similar sleep-promoting effects.

We agree that it is important to assess PV-neuron activity through CaMKII-independent activity markers or more direct methods such as Ca^{2+} imaging and electrophysiology for future research. Thus, we revised our manuscript to include potential research directions in the Discussion section as follows.

line 537: It will be important to investigate what physiological properties of PV neurons are modulated by CaMKII and what molecular mechanisms are responsible for this modulation through physiological recordings.

Reviewer #3 (Remarks to the Author):

The homeostatic regulation of sleep is characterized by rebound sleep after prolonged wakefulness, but the molecular and cellular mechanisms underlying this regulation are still unknown. Kon et al., showed that CaMKII-dependent regulation of PV-expressing cortical neurons is involved in sleep homeostasis regulation. Prolonged wakefulness enhances cortical PV-neuron activity. Chemogenetic suppression or activation of cortical PV neurons inhibits or induces rebound sleep, implying that rebound sleep is dependent on increased activity of cortical PV neurons. CaMKII kinase activity boosts the activity of cortical PV neurons and is important for homeostatic sleep rebound. The authors propose that CaMKII-dependent PV neuron activity represents negative feedback inhibition of cortical neural excitability, which serves as the distributive cortical circuits for homeostatic sleep regulation.

This is an important topic for sleep research. Overall, the experiments are carefully designed and executed, and the results are interesting and generally supportive of their conclusions. However, the authors need to address the following concerns.

We thank the reviewer for providing constructive comments and valuable suggestions. We are pleased that the reviewer evaluated our work is carefully designed and executed, and the results are interesting and generally supportive of our conclusions. The new experiments, as suggested by the reviewer, have significantly increased the reliability and solidity of our work.

Point-by-point revisions are explained below:

MAJOR POINTS:

(1)

The authors argued that cortical PV neurons are responsible for homeostatic sleep regulation, but their experiments mainly used the E11 enhancer for PV-specific genetic and chemogenetic manipulations. However, the E11 promoter not only drive expression in cortical PV neurons, but also PV and non-PV neurons in other brain regions. Thus, I think there are alternative explanations that non-PV neurons or other brain regions might be responsible for the observed sleep phenotypes. I suggest that the authors try to use either PV-Cre mice or the E22 promoter, which drives more restricted expression in the cortex than

E11 promoter (Vormstein-Schneider et al., PMID: 32807948), for manipulations to confirm that cortical PV neurons are specifically required for homeostatic sleep response.

We thank the reviewer for valuable suggestions. As the reviewer suggested, we performed additional experiments using the E22 enhancer for targeting cortical PV neurons in chemogenetic manipulation. The chemogenetic activation increased sleep, while the chemogenetic suppression tended to decrease sleep. Although the effects were milder, E22-mediated manipulation phenotypes were consistent with the results of chemogenetic manipulation using the E11 enhancer, supporting the idea that activation of cortical PV neurons induces a rebound-sleep-like state. This finding was included in the revised manuscript (**Supplementary Figs. 6a–f**).

line 247: Similar effect was observed in chemogenetic manipulation using another PV-neuron selective enhancer E22, which predominantly confined gene expression to cortical PV neurons (**Supplementary Figs. 6a–f**).

Moreover, we pursued another precise approach by using PV-Cre mice combining with the E11 enhancer to inhibit or activate CaMKII exclusively within cortical PV neurons (**Figs. 6e–h, Figs. 7n–q**). The results recapitulate the sleep phenotypes obtained by using the E11 enhancer without PV-Cre mice, including attenuation of increased NREM delta power upon SD by CaMKII inhibition (**Figs. 6a–h**). These additional findings were incorporated into the revised manuscript to further underscore the pivotal role of CaMKII activity in cortical PV neurons in homeostatic sleep regulation.

line 333: To express CN19o in cortical PV neurons more specifically, we employed a confinement strategy using the E11 enhancer and *Pvalb-2A-Cre* mice (**Fig. 6e**). Cre-dependent expression of E11-CN19o resulted in a reduction in REM sleep duration, while it had no significant impact on total sleep or NREM sleep duration (**Supplementary Figs. 9i–k**). In contrast, a significant reduction in delta power during NREM sleep was observed (**Supplementary Figs. 9l–n**). Consistent with the results observed with the direct CN19o expression under the E11 enhancer (**Figs. 6b–d**), the Cre-dependent E11-CN19o expression markedly attenuated the homeostatic rebound in delta power following SD (**Figs. 6f–h**). Taken together, these findings demonstrate that CaMKII inhibition in cortical PV neurons leads to impairment of sleep homeostasis.

line 400: To confirm the involvement of cortical PV neurons, we employed the confinement strategy using the E11 enhancer and *Pvalb-2A-Cre* mice for specific CaMKII α expression in cortical PV neurons (**Figs. 7n**). Similar to the direct CaMKII α expression under the E11 enhancer, the rebound-sleep-like state was induced by Cre-dependent CaMKII α (T286D) expression under the E11 enhancer (**Figs. 7o–q**). Sleep-promoting effects were also observed with Cre-dependent CaMKII α (T286D) expression under pan-neural *hSyn* promoter (**Supplementary Figs. 11d–g**). These findings further support the hypothesis that CaMKII activation in cortical PV neurons induces rebound sleep.

(2)

NREMS delta power is an important aspect of homeostatic sleep regulation. Since NREMS delta power increases in proportion to the duration of prior wakefulness, it is considered a good index of sleep need. However, the authors seldomly used EEG/EMG recording to measure dynamic changes of NREMS delta power during the 24-h cycle or after sleep deprivation. I think analysis of NREMS delta power needs to be improved in the revision, particularly in Fig 4 and Fig 5o-p, using a hourly plot of NREMS delta power will indicate dynamic changes of delta power after sleep deprivation.

We appreciate the reviewer's constructive comments. As the reviewer pointed out, we agree that it is crucial to examine delta power dynamics after SD in addition to sleep duration to comprehensively understand homeostatic sleep regulation. To address this, we integrated new data depicting NREM delta power dynamics before and after SD with CaMKII inhibition in E11 neurons into the revised manuscript (**Figs. 6a–h**). The results indicate that inhibition of CaMKII activity in PV/E11 neurons attenuates the accumulation of sleep need measured as NREM delta power upon SD, supporting our proposed model (please also refer to our response to **Comment (1)** above).

line 322: We next investigated whether CaMKII in cortical PV neurons is involved in the homeostatic regulation of the SWA as well as sleep duration. We performed a 6-hour SD on E11-CN19o and E11-CN19scr mice under EEG/EMG recording (**Fig. 6a**). The delta power during NREM sleep was significantly increased after SD and returned to baseline over 6 hours in the E11-CN19scr mice (**Figs. 6b,c**). In contrast, the E11-CN19o mice presented no remarkable increase in delta power after SD, suggesting that the homeostatic rebound in delta power after SD was attenuated in the E11-CN19o mice (**Figs. 6b,c**). Accordingly, the delta power rebound was significantly lower in the E11-CN19o mice compared to the E11-CN19scr mice (**Fig. 6d**). These results indicate that CaMKII kinase activity in E11 neurons is

crucial not only for homeostatic rebound of sleep amount but also for homeostatic rebound of the SWA.

line 333: To express CN19o in cortical PV neurons more specifically, we employed a confinement strategy using the E11 enhancer and *Pvalb-2A-Cre* mice (**Fig. 6e**). Cre-dependent expression of E11-CN19o resulted in a reduction in REM sleep duration, while it had no significant impact on total sleep or NREM sleep duration (**Supplementary Figs. 9i–k**). In contrast, a significant reduction in delta power during NREM sleep was observed (**Supplementary Figs. 9l–n**). Consistent with the results observed with the direct CN19o expression under the E11 enhancer (**Figs. 6b–d**), the Cre-dependent E11-CN19o expression markedly attenuated the homeostatic rebound in delta power following SD (**Figs. 6f–h**). Taken together, these findings demonstrate that CaMKII inhibition in cortical PV neurons leads to impairment of sleep homeostasis.

(3)

The authors argued that CaMKII activity in E11 neurons encoded sleep need based on observation that sleep deprivation resulted in increase of CaMKIIα T286 phosphorylation. I think this conclusion is an over-interpretation of the data because much more are required to claim that something encodes sleep need. At the minimum, the degree of increase/decrease of CaMKII activity should be in proportion to the increase/decrease of sleep need, as measured by either NREMS amount and/or NREMS delta power, during 24-h cycle or after sleep-deprivation.

We appreciate the reviewer's thorough review. We have replaced the term “encode” and substituted it with the sentences that directly describe the increased CaMKIIα T286 phosphorylation upon SD.

line 437: The next step was to see if the level of CaMKII T286 autophosphorylation in cortical PV neurons increases in response to the accumulation of sleep need.

line 454: T286 phosphorylation (pT286) level derived from E11-CaMKIIα (E285D) increased in SD group compared to *ad libitum* sleep (S) group (**Fig. 8g**). Although pT286 level serves as a surrogate for CaMKIIα kinase activity, this result suggests that CaMKII kinase activity in cortical PV neurons increases upon the accumulation of sleep need.

line 502: Our findings suggest that CaMKII kinase activity in cortical PV neurons is one of the candidates that reflect sleepiness.

(4)

The authors used “Slp” to refer to normal slept mice as opposed to sleep-deprived (SD) mice. I suggest that the authors change Slp because it was previously used as abbreviation for Sleepy mutant mice.

We agree with the reviewer's comment. We changed the abbreviation of *ad libitum* sleep condition from “Slp” to “S”.

References for Response to Reviewers' Concerns

- Bian, W.-J. *et al.* (2022) "Adolescent sleep shapes social novelty preference in mice," *Nature neuroscience*. Nature Publishing Group, pp. 1–12.
- Brüning, F. *et al.* (2019) "Sleep-wake cycles drive daily dynamics of synaptic phosphorylation," *Science*, 366(6462), p. eaav3617.
- Campbell, I. G. and Feinberg, I. (2009) "Longitudinal trajectories of non-rapid eye movement delta and theta EEG as indicators of adolescent brain maturation," *Proceedings of the National Academy of Sciences of the United States of America*, 106(13), pp. 5177–5180.
- Cirelli, C., Pompeiano, M. and Tononi, G. (1995) "Sleep deprivation and c-fos expression in the rat brain," *Journal of sleep research*, 4(2), pp. 92–106.
- Contractor, A., Ethell, I. M. and Portera-Cailliau, C. (2021) "Cortical interneurons in autism," *Nature neuroscience*. Nature Publishing Group, 24(12), pp. 1648–1659.
- Coultrap, S. J. and Bayer, K. U. (2011) "Improving a natural CaMKII inhibitor by random and rational design," *PloS one*, 6(10), p. e25245.
- Diering, G. H. *et al.* (2017) "Homer1a drives homeostatic scaling-down of excitatory synapses during sleep," *Science*, 355(6324), pp. 511–515.
- Filice, F. *et al.* (2020) "The Parvalbumin Hypothesis of Autism Spectrum Disorder," *Frontiers in cellular neuroscience*, 14, p. 577525.
- Frank, M. G. *et al.* (2017) "Development of Circadian Sleep Regulation in the Rat: A Longitudinal Study Under Constant Conditions," *Sleep*, 40(3). doi: 10.1093/sleep/zsw077.
- Franken, P., Malafosse, A. and Tafti, M. (1999) "Genetic determinants of sleep regulation in inbred mice," *Sleep*, 22(2), pp. 155–169.
- Funk, C. M. *et al.* (2017) "Role of Somatostatin-Positive Cortical Interneurons in the Generation of Sleep Slow Waves," *The Journal of neuroscience: the official journal of the Society for Neuroscience*, 37(38), pp. 9132–9148.
- Gouwens, N. W. *et al.* (2020) "Integrated Morphoelectric and Transcriptomic Classification of Cortical GABAergic Cells," *Cell*, 183(4), pp. 935-953.e19.

He, X. *et al.* (2021) "Gating of hippocampal rhythms and memory by synaptic plasticity in inhibitory interneurons," *Neuron*, 109(6), pp. 1013-1028.e9.

Iino, Y. *et al.* (2020) "Dopamine D2 receptors in discrimination learning and spine enlargement," *Nature*, 579(7800), pp. 555–560.

Ishida, A. *et al.* (1998) "Critical amino acid residues of AIP, a highly specific inhibitory peptide of calmodulin-dependent protein kinase II," *FEBS letters*, 427(1), pp. 115–118.

Jung, K. *et al.* (2023) "An adaptive behavioral control motif mediated by cortical axo-axonic inhibition," *Nature neuroscience*, 26(8), pp. 1379–1393.

Keaveney, M. K. *et al.* (2020) "CaMKII α -Positive Interneurons Identified via a microRNA-Based Viral Gene Targeting Strategy," *The Journal of neuroscience: the official journal of the Society for Neuroscience*, 40(50), pp. 9576–9588.

Liu, X. B. and Jones, E. G. (1996) "Localization of alpha type II calcium calmodulin-dependent protein kinase at glutamatergic but not gamma-aminobutyric acid (GABAergic) synapses in thalamus and cerebral cortex," *Proceedings of the National Academy of Sciences of the United States of America*, 93(14), pp. 7332–7336.

Mano, T. *et al.* (2021) "CUBIC-Cloud provides an integrative computational framework toward community-driven whole-mouse-brain mapping," *Cell Reports Methods*, 1(2), p. 100038.

Melander, J. B. *et al.* (2021) "Distinct in vivo dynamics of excitatory synapses onto cortical pyramidal neurons and parvalbumin-positive interneurons," *Cell reports*, 37(6), p. 109972.

Murakami, T. C. *et al.* (2018) "A three-dimensional single-cell-resolution whole-brain atlas using CUBIC-X expansion microscopy and tissue clearing," *Nature neuroscience*, 21(4), pp. 625–637.

Murakoshi, H. *et al.* (2017) "Kinetics of Endogenous CaMKII Required for Synaptic Plasticity Revealed by Optogenetic Kinase Inhibitor," *Neuron*, 94(1), pp. 37-47.e5.

Nelson, A. B. *et al.* (2013) "Sleep patterns and homeostatic mechanisms in adolescent mice," *Brain sciences*, 3(1), pp. 318–343.

Niwa, Y. *et al.* (2018) "Muscarinic Acetylcholine Receptors Chrm1 and Chrm3 Are Essential for REM Sleep," *Cell reports*, 24(9), pp. 2231-2247.e7.

Rensing, N. *et al.* (2018) "Longitudinal analysis of developmental changes in electroencephalography patterns and sleep-wake states of the neonatal mouse," *PloS one*, 13(11), p. e0207031.

Sancho, L. and Bloodgood, B. L. (2018) "Functional Distinctions between Spine and Dendritic Synapses Made onto Parvalbumin-Positive Interneurons in Mouse Cortex," *Cell reports*, 24(8), pp. 2075–2087.

Sik, A. *et al.* (1998) "The absence of a major Ca²⁺ signaling pathway in GABAergic neurons of the hippocampus," *Proceedings of the National Academy of Sciences of the United States of America*, 95(6), pp. 3245–3250.

Sinton, C. M., Kovakkattu, D. and Friese, R. S. (2009) "Validation of a novel method to interrupt sleep in the mouse," *Journal of neuroscience methods*, 184(1), pp. 71–78.

Tainaka, K. *et al.* (2018) "Chemical Landscape for Tissue Clearing Based on Hydrophilic Reagents," *Cell reports*, 24(8), pp. 2196-2210.e9.

Takato, J. *et al.* (2022) "The whisking oscillator circuit," *Nature*, 609(7927), pp. 560–568.

Tasic, B. *et al.* (2018) "Shared and distinct transcriptomic cell types across neocortical areas," *Nature*, 563(7729), pp. 72–78.

Terao, A. *et al.* (2003) "Region-specific changes in immediate early gene expression in response to sleep deprivation and recovery sleep in the mouse brain," *Neuroscience*, 120(4), pp. 1115–1124.

Tone, D. *et al.* (2022) "Distinct phosphorylation states of mammalian CaMKII β control the induction and maintenance of sleep," *PLoS biology*, 20(10), p. e3001813.

Vormstein-Schneider, D. *et al.* (2020) "Viral manipulation of functionally distinct interneurons in mice, non-human primates and humans," *Nature neuroscience*, 23(12), pp. 1629–1636.

Wang, Z. *et al.* (2018) "Quantitative phosphoproteomic analysis of the molecular substrates of sleep need," *Nature*, 558(7710), pp. 435–439.

Zamore, Z. and Veasey, S. C. (2022) "Neural consequences of chronic sleep disruption," *Trends in neurosciences*, 45(9), pp. 678–691.

REVIEWER COMMENTS

Reviewer #1 (Remarks to the Author):

I appreciate the efforts by the authors to conduct new experiments, which I think have improved the paper significantly. There are two outstanding issues, however, one related to the results presented in figure 1 (and 2), and the other related to the discussion. In short, I believe the results in figure 1 weaken the overall paper. The reasons for my concerns are listed below. The second general point is about the discussion, which could be streamlined and should include a more balanced discussion of the many factors known to affect sleep rebound. I am sure that the authors do not believe that the PV population (and their activation as reflected by CamKII signaling) is the only factor affecting sleep rebound, but this should be stated more clearly. Changes in neuromodulatory levels, adenosine signaling, astrocytic signaling, and pyramidal neurons activity, all affect sleep rebound. In fact, in some cases (SIK3 mutations in neurons and CamKII signaling in excitatory neurons are good examples), it is also the baseline amount of sleep that is affected. Since sleep is regulated also in baseline conditions, not just after sleep deprivation, these examples show that, not surprisingly, many factors related to synaptic activity and neuronal/glial activity affect the need for sleep.

As shown in Figure 1g, SD resulted in different degrees of sleep loss across the 3 groups, which is not surprising; the authors are measuring the rebound arbitrarily only during the night, ignoring the first 6 hours after SD; they reference “previous reports”(line 133); in fact, only one paper is cited, and that paper showed (in adult mice) that “In the first 6 hours, compensation for PS time and SWS intensity seemed to have priority over SWS time, whereas in the dark period a consistent increase in SWS time was observed in the absence of increases of SWS delta power”. So, there is no justification, even using that paper, to only focus on the night and not include the full 18 hours post-SD (if not a longer period) to measure “rebound”. Even more crucially, a very relevant and seminal study (not cited) in neonatal/young rats showed that young rats have strong and early rebound in NREM sleep duration (starting immediately after SD); Frank et al., 01 JUL 1998<https://doi.org/10.1152/ajpregu.1998.275.1.R148>

In the SD experiment, as shown in figure 1g, the cumulative sleep loss (first 18 hours after SD) is largest at 4w, lowest at 8w, and intermediate at 12 weeks, which is not so straightforward to interpret as a maturational change, as the authors want to believe (also in relation to the changes in PV cell density shown in figure 2); the new MK-801 experiment complicates things further: as shown in figure 1l, the cumulative sleep loss at the end of the night is the same for the 4w and the 12w mice, and the 8w mice have a positive rebound; thus, the conclusion by the authors that 4w mice have weaker rebound (figure 1m) is not supported by the profile shown in figure 1l, and is due to the focus on the night hours; in both experiments (SD and the drug) 4w mice start recovering sleep earlier than older mice.

Reviewer #2 (Remarks to the Author):

The manuscript Cortical parvalbumin neurons are responsible for homeostatic sleep rebound through CaMKII activation by Kon and colleagues examines the role of parvalbumin (PV) expressing neurons in the maintenance of homeostatic regulation of sleep rebound. The authors propose as a model that CaMKII kinase modulates the activity of cortical PV neurons to regulate sleep rebound.

The authors have made numerous modifications to the manuscript in response to reviewer feedback. Numerous areas have been clarified through revised text. Most of my previous concerns have been addressed with additional experimentation or at least discussion of potential issues and future work that should be completed.

I found the added vglut cfos experiment to be a bit surprising but convincing that the effects are not nonspecific.

I agree with the authors that it may be difficult to obtain meaningful CaMKII protein analysis using IHC. There are other, more sensitive approaches to this that could be performed, eg. Proximity ligation assays. However, perhaps it isn't necessary here given the scope of work that has been completed.

My only remaining concern is still with the PV/fofos colocalization data from the lightsheet/tissue clearing experiments. I appreciate that some methodological detail has been added and some images showing potential colocalization of a small number of neurons. This doesn't however really address the accuracy of the technique. The authors refer to prior work with validation, however in that paper I still do not see proper validation, comparing their result using lightsheet imaging with gold standard approach of confocal microscopy with XYZ reconstruction to confirm colocalization rather than touching or stacked cells. Images can be misleading even from high magnification images until viewed in all three planes. Proper confirmation of the changes in PV fofos expression need to be confirmed here in this manuscript.

Reviewer #3 (Remarks to the Author):

My major concern is whether the authors can claim cortical PV neurons are responsible for the homeostatic rebound sleep after sleep deprivation, based on experiments using the E11 promoter alone. In revision, the authors did attempt to alleviate my concern by restricting gene expression in cortical PV neurons by the more specific E22 promoter and by combining PV-Cre mice and the E11 promoter. While I appreciate the authors' efforts, I disagree with their interpretation of the new results that they obtained (see below). The authors have addressed my other concerns.

The effects of these more specific manipulations of cortical PV neurons are generally much milder than manipulations of the E11 promoter-marked neurons (Figs. 6e-h; Figs. 7n-q). In particular, "the E11-CN19o mice barely showed the homeostatic rebound in sleep duration, whereas the E11-CN19scr mice clearly showed rebound responses following SD (Figs. 5b-d and Sup Fig. 9d)...Additionally, the delta power rebound was significantly lower in the E11-CN19o mice compared to the E11-CN19scr mice (Fig. 6d)." By contrast, "the PV-Cre-dependent E11-CN19o expression markedly attenuated the homeostatic rebound in delta power after SD", but did not affect the homeostatic rebound in sleep duration (Fig. 6f-h). The combination of these results suggest to me that cortical PV neurons are involved in the delta power rebound, while other neuron populations are responsible for the rebound in sleep duration after SD. The rebound of sleep amount and delta power are two key aspects of the homeostatic response following sleep deprivation. I think the authors should be cautious and use precise language to interpret their experimental results.

Response to Reviewers' Concerns for Kon et al., "Cortical parvalbumin neurons are responsible for homeostatic sleep rebound through CaMKII activation".

We thank the reviewers for their constructive and insightful comments. We have further revised the manuscript, added new data analyses, and included new measurements. Specifically, addressing reviewer #1's concern, we have added a new analysis considering the different initial timing of sleep rebound for each mouse, further strengthening our claim that changes in rebound sleep amount are less prominent in the early stage of post-weaning development. Regarding on reviewer #2's suggestion, we have conducted verification of our light-sheet imaging using confocal microscopy, supporting the validity of our co-localization analysis using the light-sheet microscopy. Following reviewer #3's comment, we have added analyses of changes in rebound sleep amount under the condition of PV-Cre dependent CN19o expression and provided a more rigorous and fair interpretation of the contribution of PV neurons to homeostatic rebound in sleep amount and delta power. Additionally, we have streamlined the discussion section, adopting a more concise style by eliminating redundant content. The discussion section now includes several references suggested by the reviewers.

REVIEWER COMMENTS

Reviewer #1 (Remarks to the Author):

I appreciate the efforts by the authors to conduct new experiments, which I think have improved the paper significantly. There are two outstanding issues, however, one related to the results presented in figure 1 (and 2), and the other related to the discussion. In short, I believe the results in figure 1 weaken the overall paper. The reasons for my concerns are listed below. The second general point is about the discussion, which could be streamlined and should include a more balanced discussion of the many factors known to affect sleep rebound. I am sure that the authors do not believe that the PV population (and their activation as reflected by CamKII signaling) is the only factor affecting sleep rebound, but this should be stated more clearly. Changes in neuromodulatory levels, adenosine signaling, astrocytic signaling, and pyramidal neurons activity, all affect sleep rebound. In fact, in some cases (SIK3 mutations in neurons and CamKII signaling in excitatory neurons are good examples), it is also the baseline amount of sleep that is affected. Since sleep is regulated also in baseline conditions, not just after sleep deprivation, these examples show that, not surprisingly, many factors related to synaptic activity and neuronal/glia activity affect the need for sleep.

We thank the reviewers for their fair and constructive comments. As the reviewer mentioned, various factors, including neuromodulators level, astrocyte activity, and pyramidal neuron activity, play a significant role in regulating sleep homeostasis. Furthermore, it is also recognized that CaMKII and SIK3 signaling in excitatory neurons have an impact on baseline sleep amounts and depth, underscoring the intricate nature of sleep regulation. In the revised manuscript, we included an explicit statement that our study does not imply that CaMKII signaling in the PV neurons are the only factor regulating sleep homeostasis. Instead, we highlight their specific role in the sleep rebound rather than baseline sleep amounts. Additionally, we also streamlined the overall discussion for better readability and included a brief overview on the broader context of sleep regulation, emphasizing the intricate interplay of various factors.

line 488: While kinase activity, including CaMKII and salt-inducible kinase 3 (SIK3), in excitatory neurons has been shown to influence baseline sleep amounts and depth, the specific kinases involved in homeostatic regulation of sleep within inhibitory neurons remain unidentified. Our findings suggest that CaMKII kinase activity in cortical PV neurons is one of the molecular factors reflecting sleepiness. CaMKII inhibition in excitatory neurons resulted in changes in the total amount of daily sleep. In contrast, CaMKII inhibition in cortical PV neurons had minimal impact on daily sleep amount but impaired homeostatic rebound of sleep. Moreover, the intracellular CaMKII activation specifically enhances the activity of cortical PV neurons (**Figs. 8c–e**), not excitatory neurons (**Supplementary Figs. 13f,g**), even though both have similar sleep-promoting effects. These findings suggest that distinctive CaMKII signaling pathways exist in cortical PV neurons for the homeostatic regulation of sleep (**Fig. 8i**).

line 537: Various cortical and subcortical cell populations, including neurons and glial cells, contribute to homeostatic regulation of sleep. Sleep substances such as adenosine and cytokines are also involved in this regulation. Investigating the interplay of these factors and cortical PV neurons to regulate sleep homeostasis would be intriguing for future research.

As shown in Figure 1g, SD resulted in different degrees of sleep loss across the 3 groups, which is not surprising; the authors are measuring the rebound arbitrarily only during the night, ignoring the first 6 hours after SD; they reference “previous reports”(line 133); in fact, only one paper is cited, and that paper showed (in adult mice) that “In the first 6 hours, compensation for PS time and SWS intensity seemed to have priority over SWS time, whereas in the dark period a consistent increase in SWS time was observed in the absence

of increases of SWS delta power". So, there is no justification, even using that paper, to only focus on the night and not include the full 18 hours post-SD (if not a longer period) to measure "rebound". Even more crucially, a very relevant and seminal study (not cited) in neonatal/young rats showed that young rats have strong and early rebound in NREM sleep duration (starting immediately after SD); Frank et al., 01 JUL 1998.

In the SD experiment, as shown in figure 1g, the cumulative sleep loss (first 18 hours after SD) is largest at 4w, lowest at 8w, and intermediate at 12 weeks, which is not so straightforward to interpret as a maturational change, as the authors want to believe (also in relation to the changes in PV cell density shown in figure 2); the new MK-801 experiment complicates things further: as shown in figure 1l, the cumulative sleep loss at the end of the night is the same for the 4w and the 12w mice, and the 8w mice have a positive rebound; thus, the conclusion by the authors that 4w mice have weaker rebound (figure 1m) is not supported by the profile shown in figure 1l, and is due to the focus on the night hours; in both experiments (SD and the drug) 4w mice start recovering sleep earlier than older mice.

We thank the reviewer for their constructive and insightful comments. To address the concern, we conducted additional analyses to precisely validate rebound sleep without limiting to the dark phase. In the new analyses, we first detected the "rebound onset" in individual mice as the point when hourly sleep duration after SD or MK-801 administration exceeded the corresponding value in the control condition (**Supplementary Figs. 2b,e**). Subsequently, we computed the sleep rebound ratio from the rebound onset to the end of the day (ZT24) (**Supplementary Figs. 2c,f**), allowing us to include the light phase in the calculation if the onset occurred within that period. Notably, the rebound onset at 4 weeks old is earlier than at 8 or 12 weeks old in MK-801 administration (**Supplementary Fig. 2e**), as pointed out by the reviewer, while no significant difference between ages was observed in the onset after SD (**Supplementary Fig. 2c**). In both SD and MK-801 administration, the rebound ratio is significantly lower at 4 weeks old than at 8 or 12 weeks old when calculated from individual rebound onset (**Supplementary Figs. 2c,f**), supporting that, at least, a part of homeostatic regulatory system of sleep matures between 4 and 8 weeks old when cortical PV neuron density changes. Although there is a slight difference in rebound response at 8 and 12 weeks old, it is notably less than the difference from 4 weeks old. The contribution of PV neurons and CaMKII within these neurons to these differences remains to be explored in future studies. These additional analyses were included in the revised manuscript (**Supplementary Figs. 2b–c, d–f**).

line 146: The homeostatic rebound in sleep duration was less pronounced at 4 weeks old, even when rebound sleep was calculated from individual rebound onset without limiting to the dark phase (**Supplementary Figs. 2a–f**).

We also appreciate the reviewer's profound knowledge and insights. While acknowledging the existence of various studies, including the one cited by the reviewer, that contribute to our understanding of the developmental aspects of sleep homeostasis, we recognize that the results across these studies are not fully consistent among species and recording methods. Hence, additional investigations, including our present study, are essential for a comprehensive understanding of the development of sleep homeostasis.

For instance, as the reviewer mentioned, developing analysis in rats (P12, P16, P20, and P24) showed that the increased SWA after SD is observed only in older rats (P24), whereas SD induces NREM sleep rebound in younger rats (P12 and P16) (Frank, Morrissette and Heller, 1998). However, a different longitudinal study in rats (P21-22 and P29-30) reported contrasting results, with increased SWA observed in both P22 and P30, and NREM sleep rebound detected only in P30, not P22 (Gvilia *et al.*, 2006). Another study reported pronounced rebound responses, including increased sleep duration and arousal threshold after SD, even in early infant rats (P2 and P8) (Todd *et al.*, 2010). In developing mice (P19-111), consistent up-regulation of SWA after SD was observed in mice older than P42 (Nelson *et al.*, 2013). Furthermore, recent findings highlighted that adult (P90-100) mice exhibited a significant sleep rebound and could recover from sleep loss within the day, whereas juvenile (P21-28) or adolescent (P42-45) mice displayed an absent or blunted response to SD (Gay *et al.*, 2023), aligning with our findings. At least in mice, it appears that the homeostatic regulatory system of sleep may mature during the post-weaning period. We believe that our present study provides valuable insights into the developmental changes in sleep homeostasis, complementing other studies. The revised manuscript includes more detailed information about developmental changes in sleep homeostasis, although we couldn't cite all relevant studies due to reference limitations.

line 90: Moreover, cortical PV neurons mature throughout postnatal development in terms of distribution, electrophysiological properties, and gene expression, coinciding with developmental changes in sleep architecture and homeostatic sleep responses. Developmental dysfunctions in cortical PV neurons are associated with autism spectrum disorder (ASD), where abnormal sleep symptoms are observed in both patients and animal models, highlighting the potential connection between PV neuron maturation and sleep-wake

regulation. However, the role of cortical PV neurons in the regulation of sleep architecture and homeostatic sleep rebound remains poorly understood.

line 511: While studies in rats and mice contributed to understanding of the developmental aspects of sleep homeostasis, the results across these studies are not fully consistent. Some studies suggest that homeostatic rebound in sleep amount emerges even before weaning in rats (Frank, Morrissette and Heller, 1998; Todd *et al.*, 2010), whereas others indicate that it matures during the post-weaning period in rats or mice (Gvilia *et al.*, 2006; Nelson *et al.*, 2013). Our longitudinal sleep recordings in developing mice found that rebound sleep appears late in developmental stage (**Figs. 1g–p**).

line 523: Our findings align with previous reports suggesting that the homeostatic regulatory system of sleep is immature in juveniles and matures during the post-weaning period in mice.

Reviewer #2 (Remarks to the Author):

The manuscript Cortical parvalbumin neurons are responsible for homeostatic sleep rebound through CaMKII activation by Kon and colleagues examines the role of parvalbumin (PV) expressing neurons in the maintenance of homeostatic regulation of sleep rebound. The authors propose as a model that CaMKII kinase modulates the activity of cortical PV neurons to regulate sleep rebound.

The authors have made numerous modifications to the manuscript in response to reviewer feedback. Numerous areas have been clarified through revised text. Most of my previous concerns have been addressed with additional experimentation or at least discussion of potential issues and future work that should be completed.

I found the added vglut cfos experiment to be a bit surprising but convincing that the effects are not nonspecific.

I agree with the authors that it may be difficult to obtain meaningful CaMKII protein analysis using IHC. There are other, more sensitive approaches to this that could be performed, eg. Proximity ligation assays. However, perhaps it isn't necessary here given the scope of work that has been completed.

My only remaining concern is still with the PV/fofos colocalization data from the lightsheet/tissue clearing experiments. I appreciate that some methodological detail has been added and some images showing potential colocalization of a small number of neurons. This doesn't however really address the accuracy of the technique. The authors refer to prior work with validation, however in that paper I still do not see proper validation, comparing their result using lightsheet imaging with gold standard approach of confocal microscopy with XYZ reconstruction to confirm colocalization rather than touching or stacked cells. Images can be misleading even from high magnification images until viewed in all three planes. Proper confirmation of the changes in PV fos expression need to be confirmed here in this manuscript.

We would like to express our gratitude to the reviewer for their careful evaluation of our methodologies. We agree on the importance of validating colocalization in multiple colors with higher resolution imaging. To address this concern, we performed confocal imaging with higher XYZ resolution on the same brains previously imaged with the light-sheet microscope. We carefully selected several regions from the light-sheet images and conducted the confocal imaging within the same regions so that we were able to directly compare the results between light-sheet and confocal microscopy. Following the imaging, we digitally reconstructed coronal (z-x) and sagittal (y-z) sections to confirm colocalization of c-Fos and

PV signals in three dimensions. Our confocal imaging results confirmed the colocalization of c-Fos and PV signals without overlap to adjacent cells in a three-dimensional manner (**Supplementary Figs. 4b**), thereby supporting our findings from the light-sheet microscopy. We have included representative images for direct comparison of the same cortical region between light-sheet and confocal images in three planes in the revised manuscript (**Supplementary Figs. 4b**).

line 821: To evaluate the colocalization analysis using the light-sheet microscope, selected cortical regions of the cleared brain were also imaged using a confocal microscope (BX61WI, Olympus) equipped with a 25X objective lens (NA: 1.00, WD: 8 mm, Olympus) and 559- and 635-nm lasers (NTT Electronics). The voxel size was (X, Y, Z) = (0.497, 0.497, 0.497) μm under the condition of no tissue expansion. Colocalization of c-Fos and PV signals was confirmed in images obtained from both types of microscopes in a three-dimensional manner (**Supplementary Fig. 4b**).

Reviewer #3 (Remarks to the Author):

My major concern is whether the authors can claim cortical PV neurons are responsible for the homeostatic rebound sleep after sleep deprivation, based on experiments using the E11 promoter alone. In revision, the authors did attempt to alleviate my concern by restricting gene expression in cortical PV neurons by the more specific E22 promoter and by combining PV-Cre mice and the E11 promoter. While I appreciate the authors' efforts, I disagree with their interpretation of the new results that they obtained (see below). The authors have addressed my other concerns.

The effects of these more specific manipulations of cortical PV neurons are generally much milder than manipulations of the E11 promoter-marked neurons (Figs. 6e-h; Figs. 7n-q). In particular, "the E11-CN19o mice barely showed the homeostatic rebound in sleep duration, whereas the E11-CN19scr mice clearly showed rebound responses following SD (Figs. 5b-d and Sup Fig. 9d)...Additionally, the delta power rebound was significantly lower in the E11-CN19o mice compared to the E11-CN19scr mice (Fig. 6d)." By contrast, "the PV-Cre-dependent E11-CN19o expression markedly attenuated the homeostatic rebound in delta power after SD", but did not affect the homeostatic rebound in sleep duration (Fig. 6f-h). The combination of these results suggest to me that cortical PV neurons are involved in the delta power rebound, while other neuron populations are responsible for the rebound in sleep duration after SD. The rebound of sleep amount and delta power are two key aspects of the homeostatic response following sleep deprivation. I think the authors should be cautious and use precise language to interpret their experimental results.

We appreciate the reviewer's careful and valuable comments. To comprehensively validate the effects of Cre-dependent E11-CN19o expression on sleep homeostasis, we added new analysis about homeostatic rebound in sleep duration. While some CN19o-expressed mice did not show clear rebound in sleep duration, there was no significant difference between CN19scr- and CN19o-expressed mice (**Supplementary Figs. 9o-p**). Thus, we concluded that the impact of Cre-dependent E11-CN19o expression on sleep duration rebound was less pronounced compared to the effect on delta power rebound. The additional analysis was included in the revised manuscript (**Supplementary Figs. 9o-p**).

line 339: The Cre-dependent E11-CN19o expression markedly attenuated the homeostatic rebound in delta power following SD (**Figs. 6f-h**), while its effect on the homeostatic rebound in sleep duration was less pronounced and exhibited high individual variation (**Supplementary Figs. 9o-p**).

Considering these results, as the reviewer highlighted, the effects of more specific manipulations by the combination of E11 enhancer and *Pvalb-2A-Cre* mice appear to be generally milder than manipulations solely by the E11 enhancer in our experimental condition (**Figs. 6e–h; Figs. 7n–q**). As a recent study demonstrating SIK3 signaling in hypothalamic and cortical excitatory neurons respectively regulates sleep duration and SWA (Kim *et al.*, 2022), we agree with the reviewer's perspective suggesting that cortical PV neurons selectively regulate homeostatic rebound of SWA. However, it remains challenging to conclude from our results that cortical PV neurons selectively regulate SWA because sleep duration was also affected in Cre-dependent E11-CaMKII activation (**Figs. 7n–q**). It would be an interesting direction for future studies to examine whether specific subpopulations within cortical PV neurons selectively or biasedly regulate homeostatic rebound of sleep duration and SWA, rather than attributing the milder effects to lower efficiency of CaMKII manipulation. We included this possibility and the limitations of this study in the Discussion section of the revised manuscript.

line 550: Furthermore, CaMKII inhibition in E11 neurons significantly suppressed homeostatic rebound in sleep duration and SWA, whereas combining the E11 enhancer and *Pvalb-2A-Cre* mice predominantly suppressed SWA rebound. Although this difference may result from lower efficiency of CaMKII inhibition in the Cre-dependent expression, there is a possibility that specific subpopulations within E11 neurons regulate the two aspects of homeostatic rebound of sleep with different efficiencies.

References for Response to Reviewers' Concerns

Frank, M. G., Morrissette, R. and Heller, H. C. (1998) "Effects of sleep deprivation in neonatal rats," *The American journal of physiology*, 275(1), pp. R148-157.

Gay, S. M. *et al.* (2023) "Synapses are uniquely vulnerable to sleep loss during brain development while maturation confers resilience," *bioRxiv*. doi: 10.1101/2023.11.06.565853.

Gvilia, I. *et al.* (2006) "Homeostatic regulation of sleep: a role for preoptic area neurons," *The Journal of neuroscience: the official journal of the Society for Neuroscience*, 26(37), pp. 9426–9433.

Kim, S. J. *et al.* (2022) "Kinase signalling in excitatory neurons regulates sleep quantity and depth," *Nature*. Nature Publishing Group, pp. 1–7.

Nelson, A. B. *et al.* (2013) "Sleep patterns and homeostatic mechanisms in adolescent mice," *Brain sciences*, 3(1), pp. 318–343.

Todd, W. D. *et al.* (2010) "Brainstem and hypothalamic regulation of sleep pressure and rebound in newborn rats," *Behavioral neuroscience*, 124(1), pp. 69–78.

REVIEWERS' COMMENTS

Reviewer #1 (Remarks to the Author):

I thank the authors for addressing all my major concerns. I have no further comments.

Reviewer #2 (Remarks to the Author):

The authors have addressed my remaining comments. I appreciate their efforts and have no further concerns.

Reviewer #3 (Remarks to the Author):

The revised manuscript has addressed my remaining concern and is now acceptable for publication in Nature Communications.

Response to Reviewers' Concerns for Kon et al., "*Cortical parvalbumin neurons are responsible for homeostatic sleep rebound through CaMKII activation*".

We thank the reviewers for their efforts. This is the final version of our revised manuscript and there were no further comments from the reviewers.

REVIEWER COMMENTS

Reviewer #1 (Remarks to the Author):

I thank the authors for addressing all my major concerns. I have no further comments.

Reviewer #2 (Remarks to the Author):

The authors have addressed my remaining comments. I appreciate their efforts and have no further concerns.

Reviewer #3 (Remarks to the Author):

The revised manuscript has addressed my remaining concern and is now acceptable for publication in Nature Communications.